

**Carbon-nitrogen coupling under three schemes of model representation:**
**Traceability analysis**
Zhenggang Du[1,2], Ensheng Weng[3], Jianyang Xia[1,2*], Lifen Jiang[4], Yiqi Luo[4,5], Xuhui Zhou[1,2,6*]
[1]*Center for Global Change and Ecological Forecasting, School of Ecological and*
*Environmental Sciences, East China Normal University, Shanghai 200062, China*
[2]*Tiantong National Field Observation Station for Forest Ecosystem, School of Ecological and*
*Environmental Sciences, East China Normal University, Shanghai 200062, China*
[3]*Department of Ecology & Evolutionary Biology, Princeton University, Princeton, NJ, USA*
[4]*Center for Ecosystem Science and Society, Northern Arizona University, AZ, USA*
[5]*Department for Earth System Science, Tsinghua University, Beijing 100084, China*
[6]*Shanghai Institute of Pollution Control and Ecological Security, 1515 North Zhongshan Rd,*
*Shanghai 200437, China*
**\*For correspondence:**
Xuhui Zhou
*School of Ecological and Environmental Sciences*
*East China Normal University*
*500 Dongchuan Road, Shanghai 200062, China*
**Email**: xhzhou@des.ecnu.edu.cn
**Tel/Fax:** +86 21 54341275
Jianyang Xia
*School of Ecological and Environmental Sciences*



*East China Normal University*
*500 Dongchuan Road, Shanghai 200062, China*
**Email**: jyxia@des.ecnu.edu.cn

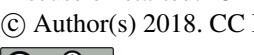



**Abstract** The interaction between terrestrial carbon (C) and nitrogen (N) cycles has been
incorporated into more and more land surface models. However, the scheme of C-N coupling
differs greatly among models, and how these diverse representations of C-N interactions will
affect C-cycle modeling remains unclear. In this study, we explored how the simulated
ecosystem C storage capacity in the terrestrial ecosystem (TECO) model varies with three
different commonly-used schemes of C-N coupling. The three schemes (SM1, SM2, and SM3)
have been used in three different coupled C-N models (i.e., TECO-CN 2.0, CLM 4.5, and O-CN,
respectively). They differ mainly in the stoichiometry of C and N in vegetation and soils, plant N
uptake strategies, pathways of N import, and the competition between plants and microbes for
soil mineral N. We incorporated them into the C-only version of TECO model, and evaluated
their impacts on the C cycle with a traceability framework. Our results showed that all of the
three C-N schemes resulted in significant reductions in steady-state C storage capacity compared
with the C-only version, but the magnitude varied with -23%, -30% and -54% for SM1, SM2,
SM3, respectively. The reduced C storage capacity is the combination of decreases in net
primary productivity (NPP) by -29%, -15% and -45% with changes of mean C residence time
(MRT) by 9%, -17% and -17% for SM1, SM2, and SM3, respectively. The divergent NPP are
mainly attributed to the different assumptions on plant N uptake, plant tissue C:N ratio, down-
regulation photosynthesis, and biological N fixation. In comparison, the alternative
representations of the plant and microbe competition strategy and the plant N uptake, combining
with the flexible C:N ratio in vegetation and soils, led to a notable spread MRT. These results
highlight that the diverse assumptions on N process representation among different C-N coupled
models could cause additional uncertainty to land surface models. Understanding their difference
can help us to improve the capability of models to predict future biogeochemical cycles on land.
**Keywords:** carbon-nitrogen coupling, traceability analysis, carbon storage capacity, nitrogen
limitation, carbon residence time



**1. Introduction**
The terrestrial ecosystem carbon (C) storage is jointly determined by ecosystem C input (i.e., net
primary productivity, NPP) and mean residence time (MRT), which are modulated by the
availability of nitrogen (N) for plant and microbial growth (Vitousek et al., 1991; Wieder et al.,
2015; Luo et al., 2017). N is an essential component of enzymes, proteins, and secondary
metabolites (van Oijen and Levy, 2004). Plant production require N to meet the stoichiometric
demands (Cleveland et al., 2013). Although there is abundant N in the atmosphere, it is difficult
to make it available for biological systems (Houlton et al., 2008). As a consequence, the
biological N availability, which strongly affects C storage in ecosystems, is often highly
correlated with key metabolic rates such as photosynthesis (Field and Mooney, 1986; Du et al.,
2017) and respiration (Sprugel et al., 1996). N thus plays an important role in governing the C
balance and turnover of terrestrial ecosystem (García-Palacios et al., 2013; Shi et al., 2015).
Given the importance of N availability on C sink projections (Wang and Houlton 2009,
Zaehle et al., 2015, Wieder et al., 2015), N processes are increasingly incorporated into
biogeochemical models. The representation of N cycling and their feedback to C cycling in
models reflects what has been established in the ecosystem research community. Early C-N
coupled models demonstrated that the N availability limits C storage capacity and can lead to
growth enhancement when N mineralization increases in many terrestrial ecosystems (i.e.,
progressively increasing N limitation) (Melillo et al., 1993; Luo et al., 2004). Evidences from
more recently studies have largely confirmed these results and have generated multiple
hypotheses for improving C-N coupling models (Zhou et al., 2013; Zaehle et al., 2014; Thomas
et al., 2015). They include the plant down-regulation productivity based on N required for cell
construction or N availability for plant absorption (Thornton et al., 2009; Gerber et al., 2010),
constant or flexible stoichiometry for allocation and tissue (Wang et al., 2001; Shevliakova et al.,
2009; Zaehle et al., 2010), competition between plants and microbes for soil nutrients (Zhu et al.,
2017), Evapotranspiration- or NPP-driven empirical functions to generate spatial estimates of
biological N fixation (BNF) (Wieder et al., 2015), and respiration of excess C to obtain N from
environment and/or to prevent the accumulation of C beyond the storage capacity (Zaehle et al.,
2010). These knowledge have significantly helped improve our understanding of the terrestrial
C-N coupling and are an important basis to develop comprehensive terrestrial process-based



models (Thornton et al., 2007; Thomas et al., 2013). However, simulated results of the terrestrial
C cycle illustrated considerable spread among models, and much of uncertainty arose from
predictions of N effects on C dynamic. The contradictory results were largely from different
representations of fundamental N processes (e.g., the degree of flexibility of C:N ratio in
vegetation and soils, plant N uptake strategies, pathways of N export, decomposition, and the
representations of the competition between plants and microbes for mineral N). Furthermore, the
methodology used to derive the C-N coupling schemes among models varies largely, which may
be invalid for the model intercomparisons to provide insight into the underlying mechanism of N
status for terrestrial C cycle projection.
In the past decades, terrestrial models integrated more and more processes to improve model
performance. While the more processes incorporated, the more difficult it becomes to understand
or evaluate model behavior (Luo et al., 2015). Xia et al (2013) developed a traceability analysis
framework that helped improve the comparability of models and data, evaluated impacts of
additional model components, facilitated benchmark analyses, model intercomparisons, and data-
model fusion, and improved model predictive power. Based on the traceability analysis
framework, key traceable elements, including fundamental properties of the terrestrial C cycle
and their representation in shared structures among existing models, can be identified and
characterized under different sources of variation (e.g., external forcing and uncertainty in
processes) compared to the achieved predictive ability. The traceability analysis framework
enables diagnosis of where models are clearly lacking predictive ability and evaluation of the
relative benefit when more or alternative components are added to the models (Luo et al., 2015).
The present study is designed to examine the effects of C-N coupling under different schemes
of model representation on ecosystem C storage in the Terrestrial Ecosystem (TECO) model
with traceability analysis framework. Three schemes of model representation were conducted
mainly based on TECO-CN 2.0 (SM1), CLM 4.5 (SM2), and O-CN (SM3, Table 1). The three
C-N schemes differ in degrees of flexibility of C:N ratio in vegetation and soils, plant N uptake
strategies, pathways of N import, and the representations of the competition between plants and
microbes for soil available N. Based on the forcing data of ambient $CO_2$ concentration, N
deposition and meteorological data (i.e., air temperature, soil temperature, relative humidity,
vapour pressure deficit, precipitation, wind speed, photosynthetically active radiation) obtained
from Duke Forest during the period of 1996-2007, we conduct three alternative C-N coupling





schemes (i.e., SM1, SM2 and SM3) as well as C-only in TECO model framework to compare
their effects on the ecosystem C storage capacity using traceability analysis framework. The N-
processes sensitivity analysis was carried out to evaluate the variability in estimated ecosystem C
storage caused by the process-related parameters at steady state.

**2. Materials and methods**
**2.1 Data sources**
The forcing data used in this study were taken from the AmeriFlux site at Duke Forest, located in
the Blackwood Division, North Carolina, USA (35.97$^o$ N, 79.08$^o$ W). The flux tower lies on a
15-year-old loblolly pine (*Pinus taeda L.*) plantation. The meteorological forcing data were
downloaded from the AmeriFlux database at http://ameriflux.lbl.gov, including ambient $CO_2$
concentration ([$CO_2$]), air temperature at the top canopy (*Ta*), soil temperature (*Ts*),
photosynthetically active radiation (*PAR*), relative humidity (*RH*), vapor pressure deficit (*VPD*),
precipitation, wind speed [*Ws*], and N deposition. All forcing data sets are available from 1996 to
2007. Standing biomass and biomass production date at each plot for plant compartments (i.e.,
foliage, fine root and woody biomass, including branches and coarse roots) were taken from
McCarthy et al. (2010). The C and N concentration data for each plant compartment based on
Finzi et al. (2007) were used to estimate C and N stocks and fluxes. Plant N demand and uptake
were calculated from these data following Finzi et al. (2007). The C and N concentrations of
litter and SOM were obtained from Lichter et al. (2008).

**2.2 Model description and C-N schemes**
**2.2.1 TECO-CN 2.0**
The terrestrial ecosystem C-N coupling model (TECO-CN, version 2.0) used in the present study
is a variant of the TECO-Carbon-only version (TECO-C) by incorporating additional key N
processes (Figure 1). TECO-C model is a process-based ecosystem model designed to examine
critical processes regulating interactive responses of plants and ecosystems to climate change. It
has four major components: canopy photosynthesis module, plant growth module, soil water
dynamic module, and soil C dynamic module. The canopy photosynthesis and soil water
dynamic modules run at hourly time step while the plant growth and soil C dynamic modules run



at the daily time step. The detailed description of the TECO-C model can be found in Weng and
Luo (2008).
The N cycle added to the TECO model for this study is simplified following the structure of
Luo & Reynolds (1999), Gerber et al. (2010), and Wang et al. (2010). It has a similar structure to
the TECO-C model (Figure 1). There are nine organic N pools and one inorganic soil N pool,
including plant, litter and soil N pools. The plant N pools include leaves, wood, roots, and
mineral N in plant tissues. The litter and soil N pools include metabolic and structural litter N,
fast, slow, and passive soil organic N (SON), and soil mineral N pools. The total plant N demand
on each time step is calculated following the NPP allocation to new tissue growth based on their
C:N ratios. To meet the demand, the plant N supply is calculated from three parts, including the
retranslocated N from senescing tissues, plant uptake from soil mineral N pool, and external N
sources from atmospheric deposition and biological N fixation. The N absorbed by roots enters
into the mineral N pool in plant tissues, and then is allocated to the remaining plant pools with
plant growth. The N in leaves and fine roots is reabsorbed before senescence. Plant litters will
enter metabolic or structural pools depending on their C:N ratios.
Allocation of assimilated C among the leaves, stems and roots depends on their growth rates,
and varies with phenology (Luo et al., 1995; Denison and Loomis, 1989; Shevliakova et al., 2009;
Weng and Luo, 2008):
$$b_l = \frac{1}{1+c_1+c_2} \qquad (1)$$

$$b_s = \frac{c_2}{1+c_1+c_2} \qquad (2)$$

$$b_r = \frac{c_1}{1+c_1+c_2} \qquad (3)$$

where $b_l$, $b_s$ and $b_r$ are the partitioning coefficient of newly assimilated C to leaves, stems and
roots, respectively. Parameters $c_1$ and $c_2$ are calculated as:
$$c_1 = \frac{bm_l}{bm_r} * \frac{CN_l^i}{CN_l^0} \qquad (4)$$

$$c_2 = 0.5 * 250e^3 * SLA * 0.00021 * h^2 \qquad (5)$$

where $bm_l$ and $bm_r$ are the leaf and root biomass; $CN_l^i$ and $CN_l^0$ represent the C:N ratio of the
leaf pool at 0 and current time step, respectively; $SLA$ is specific leaf area; $h$ is plant height,
which is calculated as:
$$h = h_{max}(1 - \exp(-h_1 * bmP)) \qquad (6)$$





where $h_{max}$ is the maximum canopy height; $h_1$ is an empirical parameter and $bmP$ is plant
biomass.

### 2.2.2 C-N coupling schemes

We conducted four experiments including three simulations with their representations of C-N
coupling schemes (SM1, SM2 and SM3) and an additional C-only simulation in TECO model
framework. The three C-N interaction simulations include one original scheme in TECO-CN2.0
model and the other two schemes representing CLM4.5-BGC and O-CN. The three C-N
coupling schemes differ in the representation of down-regulation of photosynthesis, the degree of
flexibility of C:N ratio in vegetation and soils, plant N uptake strategies, pathways of N import to
the plant reserves, and the competition between plants, and microbes for soil mineral N (Table1,
Figure 2).

### SM1 (TECO-CN2.0)

The N down-regulation of photosynthesis in SM1 is determined by the comparison between
plant N demand and actual supply of N:

$$f_{dreg} = \min(\frac{N_{sup}}{N_{demand}}, 1) \quad (7)$$

where $N_{demand}$ is plant N demand, and $N_{sup}$ is actual supply of N obtained from re-translocated
N, plant N uptake, and biological N fixation.
The re-translocated N is calculated as:

$$N_{retrans} = \sum_{i=leaf,wood,root} r_i \times outC_i/CN_i \quad (8)$$

where $r_i$ is the N resorption coefficient and $outC_i$ is the value of C leaving plant pool $i$ in each
time step.
The plant N uptake from soil mineral N pool is a function of root biomass density ($Root_{total}$, g
C m$^{-2}$) and N demand of plants, following McMurtrie $et$ $al.$ (2012)

$$N_{uptake} = min(\max(0, N_{demand} - N_{retrans}), f_{U,max} \times SN_{mine} \times \frac{Root_{total}}{Root_{total}+Root_0}) \quad (9)$$

where $N_{demand}$ is the N demand of plants; $SN_{mine}$ is the soil mineral N (gN m$^{-2}$); $f_{U,max}$ is the
maximum rate of N absorption per step when $Root_{total}$ approaches infinity; and $Root_0$ is a
constant of root biomass (g C m$^{-2}$) at which the N-uptake rate is half of the parameter $f_{U,max}$.
The biological N fixation is calculated as:





$$N_{BNF} = \min(\max(0, N_{demand} - N_{retrans} - N_{uptake}), n_{fix} \times f_{nsc} \times NSC) \quad (10)$$
where $n_{fix} = 0.0167$ is the maximum N fixation ratio and $f_{nsc}$ is the nutrient concentration
limiting factor. $f_{nsc}$ is calculated as
$$f_{nsc} = \begin{cases} 0, & NSC < NSC_{min} \\ \frac{NSC - NSC_{min}}{NSC_{max} - NSC_{min}}, & NSC_{min} < NSC < NSC_{max} \\ 1, & NSC > NSC_{max} \end{cases} \quad (11)$$
where $NSC_{min}$ and $NSC_{max}$ are the minimal and maximal sizes of nonstructural C pool,
respectively.
Two pathways of N loss are modeled. One is gaseous loss and another is leaching. They both
are proportional to the availability of soil mineral N ( $SN_{min}$). The equations are:
$$N_{gas\_loss} = f_{ngas} \times e^{\frac{T_{soil} - 25}{10}} \times SN_{min} \quad (12)$$
$$N_{leach} = f_{nleach} \times \frac{V_{runoff}}{h_{depth}} \times SN_{min} \quad (13)$$
where $f_{ngas} = 0.001$ and $f_{nleach} = 0.5$, $T_{soil}$ is the soil temperature, $V_{runoff}$ is the value of
runoff, and $h_{depth}$ is the soil depth.

**SM2 (CLM4.5bgc)**
The N down-regulation of photosynthesis in SM2 is calculated as:
$$f_{dreg} = \frac{CF_{allo} - CF_{avail\_alloc}}{CF_{GPP_{pot}}} \quad (14)$$
where $CF_{allo}$ is the total flux of allocated C, which is determined by available mineral N.
$CF_{avail\_alloc}$ is the potential C flux from photosynthesis, which can be allocated to new growth.
$CF_{GPP_{pot}}$ is the potential gross primary productivity (GPP) when there no N limitation.
The re-translocated N is calculated as:
$$N_{retrans} = \min(N_{demand} \times \frac{N_{retrans_{ann}}}{N_{demand_{ann}}}, N_{retrans\_avail}) \quad (15)$$
where $N_{retrans_{ann}}$ is the previous year's annual sum of re-translocated N obtained from
senescing tissues, $N_{demand_{ann}}$ is the previous year's annual sum of plant N demand.
$N_{retrans\_avail}$ is the available re-translocated N in senescing tissues, which is calculated by the
proportional of senescing tissues.
The plant N uptake is described as:





$$N_{uptake} = (N_{demand} - N_{retrans}) \times f_{plant\_demand} \qquad (16)$$

where $f_{plant\_demand}$ is the fraction (from 0 to 1) of the plant N demand, which can be met given
the current soil mineral N supply and competition with heterotrophs. $f_{plant\_demand}$ is set equal to
the fraction of potential immobilization demand ($f_{immob\_demand}$) that is calculated as:
$$f_{plant\_demand} = f_{immob\_demand} = \frac{SN_{min}}{N_{plant\_demand} + N_{immob\_demand}} \qquad (17)$$

where $N_{immob\_demand}$ is the total potential N immobilization demand (i.e., total potential
microbial N demand).
The biological N fixation is calculated as:
$$N_{BNF} = \left. 1.8(1 - e^{-0.03 \times NPP_{py}}) \middle/ (86400 \times 365) \right. \qquad (19)$$

where $NPP_{py}$ is the previous year NPP.

**SM3 (O-CN)**
The N downregulation of photosynthesis in SM3 is calculated as:
$$f_{dreg} = a + b \times N_{leaf/LAI} \qquad (20)$$

where *a* and *b* are empirical constants, and $N_{leaf/LAI}$ is foliage N per unit leaf area.
The re-translocated N is calculated as:
$$N_{retrans} = \sum_{i=leaf,root} \tau_i \times f_{trans,i} \qquad (21)$$

where $\tau$ is the foliage or roots shed each step. $f_{trans,leaf} = 0.5$ and $f_{trans,root} = 0.2$ are the
fractions of N re-translocated when the tissue dying off.
The plant N uptake is calculated as:
$$N_{uptake} = v_{max} \times SN_{min} \times (k_{Nmin} + \frac{1}{N_{min} \times K_{Nmin}}) \times f(T_{soil}) \times f(NC_{plant}) \times C_{root} \qquad (22)$$

where $v_{max}$ is maximum N uptake capacity per unit fine root mass, $k_{Nmin}$ is the rate of N uptake
not associated with Michaelis-Menten Kinetics, $K_{Nmin}$ is the half saturation concentration of fine
root N uptake. $f(T_{soil})$ is calculated as:
$$f(T_{soil}) = \exp\left(308.56 * \left(\frac{1}{56.02} - \frac{1}{T_{soil} + 46.02}\right)\right) \qquad (23)$$

where $T_{soil}$ is soil temperature.





$C_{root}$ is fine root mass. $f(NC_{plant})$ is the dependency of N uptake on plant N status, and is
calculated as:
$$f(NC_{plant}) = \max(\frac{NC_{plant} - nc_{leaf,max}}{nc_{leaf,min} - nc_{leaf,max}}, 0) \qquad (24)$$
where $nc_{leaf,min}$ and $nc_{leaf,max}$ are the minimum and maximum foliage N concentrations,
respectively. $NC_{plant}$ is taken as the mean N concentration of foliage, fine roots, and labile N
pools, representing the active and easily translocatable portion of plant N:
$$NC_{plant} = \frac{N_{leaf} + N_{root} + N_{labile}}{C_{leaf} + C_{root} + C_{labile}} \qquad (25)$$
The biological N fixation is calculated as:
$$N_{BNF} = 0.1 \times \max(0.0234 \times 30 \times AET + 0.172, 0) \qquad (26)$$
where $AET$ is the mean annual evapotranspiration.

**2.3 Traceability analysis framework**
The traceability analysis framework was used to evaluate the variation of the modeled ecosystem
C storage capacity under different C-N schemes (Figure S1). According to the traceability
analysis framework (Xia et al., 2013), the modeled C storage capacity can be traced to (i) a
product of NPP and ecosystem residence time ($\tau_E$). The latter $\tau_E$ can be further traced to (ii)
baseline C residence time ($\tau_E'$), which is usually preset in a model according to vegetation
characteristics and soil types, (iii) N scalar ($\xi_N$), (iv) environmental scalars ($\xi$) including
temperature ($\xi_T$) and water ($\xi_W$) scalars, and (v) the external climate forcing. The framework for
decomposing modeled C storage capacity into a few traceable components is built upon a pool-
and flux- structure, which is adopted in all of the terrestrial C models. The structure can well be
represented by a matrix equation (Luo et al., 2003; Luo and Weng, 2011):
$$\frac{dX(t)}{dt} = BU(t) - A\xi CX(t) \qquad (27)$$
where $X(t) = (X_1(t), X_2(t), \dots, X_8(t))^T$ is an $8 \times 1$ vector describing eight C pool sizes in leaf,
root, wood, metabolic litter, structural litter, fast, slow, and passive soil organic C, respectively,
in the TECO model (Weng and Luo, 2008). $B = (b_1, b_2, b_3, 0, \dots, 0)^T$ represents the partitioning
coefficients of the photosynthetically fixed C into different plant pools. $U(t)$ is the input of fixed
C via plant photosynthesis. $A$ is an $8 \times 8$ matrix representing the C transfer between pools. $\xi$ is an
$8 \times 8$ diagonal matrix of control of plant N status and environmental scalars on C decay rate at





each time step. $C$ is an $8 \times 8$ diagonal matrix representing the C exit rates from a pool at each
time step.

The C storage capacity equals the sum of C in all pools at steady state ($X_{ss}$), which can be

obtained by making Eqn.(27) equal zero as described in Xia et al. (2013):
$$X_{ss} = (A\xi C)^{-1} B U_{ss} \qquad (28)$$
The vector $U_{ss}$ is the ecosystem C influx at steady state. The partitioning ($B$ vector), transfer
coefficients ($A$ matrix) and exit rates ($C$ matrix) in Eqn.(27) together determine the baseline C
residence time ($\tau'_E$):
$$\tau'_E = (AC)^{-1} B \qquad (29)$$
The baseline C residence time ($\tau'_E$) in Eqn.(29), N scalars ($\xi_N$) and environmental scalars ($\xi_E$)
values together determine the C residence time ($\tau_E$):
$$\tau_E = \xi^{-1} \tau'_E = (\xi_N \times \xi_E)^{-1} \tau'_E \qquad (30)$$
Thus, the C storage capacity is jointly determined by the ecosystem residence time ($\tau_E$) and
steady state C influx ($U_{ss}$):
$$X_{ss} = \tau_E U_{ss} \qquad (31)$$
The environmental scalar is further separated into the temperature ($\xi_T$) and water ($\xi_W$) scalar
components, which can be represented as:
$$\xi_E = \xi_T \times \xi_W \qquad (32)$$
The N scalar is given by vector $\xi_N = (\xi_{N1}(t), \; \xi_{N2}(t), \ldots, \xi_{N8}(t))^T$ . The component $\xi_{Ni}(t)$
quantifies the changes of N content at each time step compared with initial condition in the plant
pool $i$. It is calculated as:
$$\xi_{Ni} = \exp(-\frac{CN_i^0 - CN_i^n}{CN_i^0}) \qquad (33)$$
where $CN_i^0$ and $CN_i^n$ are the C:N ratio of the pool $i$ at 0 and $n$ time step, respectively.

**2.4 Model simulations and sensitivity analysis**
To obtain the modeled C storage capacity, we spun up the TECO model with the C-only and
three C-N coupling schemes to steady state using the semi-analytical solution method developed
by Xia et al. (2012). Once the simulations are spun up to steady state, C and N fluxes and state
variables as well as the matrix elements $A$, $C$, $B$, and $\xi$ in Eqn.(28) from all time steps in the last
recycle of the climate forcing were saved for traceability analysis.





The sensitivities of both NPP and MRT to each main N process in three schemes were
calculated as:
$$S_i^{NPP}(P) = \frac{NPP_i^+(P) - NPP_i^-(P)}{NPP_i^0} \qquad (34)$$
$$S_i^{MRT}(P) = \frac{MRT_i^+(P) - MRT_i^-(P)}{MRT_i^0} \qquad (35)$$
where $S_i^{NPP}(P)$ and $S_i^{CRT}(P)$ ($i = 1, 2, 3$) represent the sensitivities of the NPP and MRT to the
N-process $P$ in the scheme $i$, respectively. $NPP_i^+(P)$ and $NPP_i^-(P)$ are the annual mean values
of NPP that were simulated in scheme $i$ based on the value of the N-process $P$ (ie., DRP, PS,
PUN, PMC, BNF, RtrN and SS) increasing 50% and decreasing 50%, respectively. $MRT_i^+(P)$
and $MRT_i^-(P)$ are the annual mean values of MRTs that were simulated in the same way as NPP
and calculated using Eqn.(29) and Eqn.(30). $NPP_i^0$ and $MRT_i^0$ are the annual mean values of
NPP and MRT at the steady state in the scheme $i$.

**3. Results**
**3.1 Simulations of C and N dynamics at steady state**
At the steady state, the dynamics of N fluxes and soil mineral N showed different patterns among
three C-N schemes in the TECO model (Fig 3). The simulated soil N mineralization and plant N
uptake fluxes in SM2 displayed the largest daily variations (0.0015 and 0.00086 g N m$^{-2}$d$^{-1}$,
respectively) and annual mean values (1.26 and 0.23 g N m$^{-2}$yr$^{-1}$, respectively) among three C-N
schemes. For the N leaching flux, SM1 showed the largest daily variation (0.04 g N m$^{-2}$d$^{-1}$) and
annual mean value (0.36 g N m$^{-2}$yr$^{-1}$). However, the biological N fixation (BNF) flux in SM1
showed the largest daily variation (0.028 g N m$^{-2}$d$^{-1}$) but the smallest annual value (0.04 g N m$^{-2}$yr$^{-1}$)
$^{2}$yr$^{-1}$) among three C-N schemes. For the N immobilization fluxes, SM3 simulated the largest
daily variation (0.0013 g N m$^{-2}$d$^{-1}$) and SM1 showed the largest annual mean value (1.15 g N m$^{-2}$yr$^{-1}$)
$^{2}$yr$^{-1}$). The dynamics of soil mineral N in SM2 and SM3 displayed the similar patterns on the
daily and annual dynamics.
Compared with the TECO-C model, the three C-N coupling schemes introduced significant
signs of N limitation on forest growth at the steady state but with varying strength (Fig 4).
Specifically, the three N schemes resulted in significant reductions in GPP (10%, 10% and 12%
for SM1, SM2 and SM3, respectively) compared to the C-only TECO model. Similar response
patterns were also found on NPP, ecosystem respiration, and heterotrophic respiration. Among

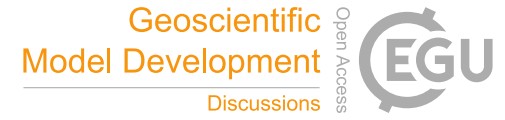

the three schemes, SM3 had the strongest effect (45%, 12% and 45% reduction for NPP,
ecosystem respiration, and heterotrophic respiration, respectively), SM2 had the weakest effect
(15%, 8% and 13%, respectively) and the effect in SM1 was moderate (29%, 10% and 29%,
respectively). However, both the SM1 and SM2 schemes increased the autotrophic respiration
(R-auto) by 12% and 27%, respectively, and SM2 scheme increased the NEE by 32%. Due to the
NSC pool of TECO model, NEE were positive in all the experiments at the steady state (Weng
and Luo, 2008). The NPP and plant N uptake (PNU) joint determine the N use efficiency (NUE).
The divergent effects of three C-N schemes on NPP and PNU lead to different NUE (Fig. 5).
SM1 had the highest NUE (159.1 g C g$^{-1}$ N), mainly resulting from its lowest PNU. In contrast,
SM3 had the lowest NUE (67.3 g C g$^{-1}$ N) as a result of its smallest NPP.

**3.2 Simulation of C storage capacity**

The ecosystem C storage capacity differed greatly among the three C-N coupling schemes as
well as with the C-only version of TECO model (Fig. 6). The C-only version had the largest C
storage capacity (19.5 Kg C m$^{-2}$) among the four simulations, resulting from its highest NPP
(879.9 g C m$^{-2}$ yr$^{-1}$). The C storage capacity in SM1 (15.1 Kg C m$^{-2}$) was close to that in SM2
(13.7 Kg C m$^{-2}$). The SM3 had the lowest C storage capacity (8.9 Kg C m$^{-2}$) among the four
simulations as a result of its smallest NPP (483.9 g C m$^{-2}$ yr$^{-1}$) and relative short MRT (18.6
years). By comparison with the C-only version, the three C-N schemes all induced different
reductions on NPP (-29%, -15% and -45% for SM1, SM2, SM3, respectively) and further
reduced their ecosystem C storage capacity. For the MRT, the three C-N schemes exhibited
contrasting effects between SM1 (+9%) and another two schemes (i.e., -16.9% in SM2 and -16.7%
in SM3) compared with the C-only TECO model.

**3.3. Ecosystem C residence time**

Ecosystem C residence time ($\tau_E$) is collectively determined by baseline residence time, N scalar
and environmental scalar as shown in Eqn (30). Specifically, differences in $\tau_E$ among three C-N
coupling schemes and C-only TECO model are determined by baseline residence time and the
effects of N scalar on eight plant C pools (Fig. 7). For example, SM1 had the longest $\tau_E$ because
the N scalar had very strong control on passive SOM. The baseline residence time were further
determined by the C allocation pattern (Fig. 8). Overall, compared with C-only version, the

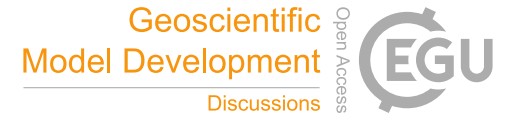



additional N processes enhanced the partitioning coefficient of NPP to roots ( 33%, 82% and
53%, respectively for SM1, SM2 and SM3), while it decreased the partitioning coefficient to
wood ( -25%, -45% and -34%, respectively). Furthermore, the decreased partitioning coefficient
to wood (b2) regulated the variations of the baseline residence time of wood, structural litter,
slow and passive SOM. However, the increased partitioning coefficient to roots (b3) determined
the variations of the baseline residence time of root and metabolic litter.

**3.4. Sensitivity of N processes to NPP and MRT**
For either NPP or MRT, the N processes had different sensitivities among the three C-N schemes
of TECO model (Fig. 9). For NPP, plant C:N ratio had the highest sensitivities in both SM1
(0.32) and SM2 (0.53). However, plant N uptake in SM3 had the highest sensitivity (0.87) for
NPP. For MRT, competition between plants and microbes, down-regulation photosynthesis and
plant C:N had the highest sensitivities in SM1 (0.27), SM2(0.19) and SM3 (0.56), respectively.

**4. Discussions**

**4.1 Underlying N processes and plant production**
Gross or net primary production (i.e., NPP or GPP) is regulated by the amount of N available for
growth through the N demand, which is set by the relative proportion of biomass growth in the
different plant components and their C:N stoichiometry (Zaehle et al., 2014; Thomas et al., 2015).
The limitation of equilibrium N on plant production reflects the effects from multiple processes
of the C-N interaction, mainly including down-regulation of photosynthetic capacity (DRP) by N
availability, the ecosystem's balance of N inputs and losses (net ecosystem N exchange, NNE),
plant N uptake (PNU), soil N mineralization (SNM), and the C:N stoichiometry of vegetation
and soils. However, due to a lack of consensus on the nature of the mechanisms, the
representation of these processes varies greatly among models (Zaehle et al., 2014).

There are two common alternative assumptions of the DRP that have been implemented in

models: (1) the change in photosynthetic capacity is directly associated with the magnitude of
plant available N (e.g., SM2), and (2) N limitation is associated with foliage N, which feeds back
to limit photosynthetic capacity (e.g., SM1 and SM3). Our results showed that both assumptions
had significant limitations with similar effects on GPP (Fig.3). The probable reason is that the





TECO model calculates photosynthesis by light availability vs. carboxylation rate based on the
Farquhar model (Farquhar et al., 1980). The effects of N stress under TECO framework, either
associated with plant available N or associated with foliage N concentration, are estimated
according to limiting factors of photosynthetic biochemistry (the maximum rate of carboxylation,
$V_{cmax}$, and the maximum rate of electron transport at saturating irradiance, $V_{jmax}$).
At or near the steady state, NNE is driven by the processes of N input via deposition and
fixation and N loss via leaching and volatilization stoichiometry (Zaehle et al., 2014; Thomas et
al., 2015). Previous studies have stated that analyzing the steady-state condition is useful to
understand N effects because the balance between external N sources and N losses determine
whether an ecosystem is N limited (Rastetter et al., 1997; Menge et al., 2009; Thomas et al.,
2015). In this study, divergent NPP responses among three schemes might partly result from
their different representations of BNF. For example, SM2 and SM3 simulated BNF explicitly,
which used modified empirical relationships to calculate BNF based on evapotranspiration (ET)
and NPP, respectively. These phenomenological relationships generally captured
biogeographical observations of higher rates of BNF in humid environments with high solar
radiation (Wieder et al., 2015). However, the highest response of NPP in only ET-driven BNF
(i.e., SM3) may illustrate that not only energetic but also C costs of 'fixing' atmospheric di-N
($N_2$) into a biologically usable form ($NH_3$) broadly affect NPP (Gutschick 1981, Rastetter et al.,
2001). This was because SM3 considered C investments in BNF while SM2 did not. On the other
hand, SM1 applied a different strategy, which represents BNF as a complement to the plant N
uptake in terms of C investment, leading to the highest plant NUE but the lowest response of
BNF to NPP. Another driving factor of NNE is the N loss, which depends on the rate of leaching
and volatilization. Using the same formulation as proportional to the size of soil mineral N pool
among three schemes, the divergent annual mean magnitude of N leaching was more correlated
to soil mineral N.
The processes of PNU and net N mineralization determine how N moves through the plant-
soil system, thereby triggering N limitation on plant growth. However, to our knowledge,
exploring those processes exactly in models is limited by inadequate representation of above-
and below-ground interactions that control the patterns of N allocation and whole-plant
stoichiometry (Zaehle et al., 2014; Thomas et al., 2015). Plant tissue, litter, and SOM are the
primary sinks of N in terrestrial ecosystems, while N in these forms is not directly available for



PNU, leading to an increase in N demand due to plant growth. On the other hand, these N must
turn over to become available for plant growth. Therefore, the time for N to stay in these
unavailable pools controls the transactional delay between the incorporation of N into plant
unavailable pool and becomes available for plant uptake. In this way, the residence time of N in
SOM appears to be an important factor for governing plant growth (see next section). In the
present study, SM1 had the highest NUE from the combined effects of PNU based on C
investment strategy (as described above) and flexible tissue C:N ratio. N stress increased tissue
C:N ratio, leading to a high microbial N immobilization and then a lower net N mineralization,
which allowed plant cell construction with a lower N requirement. The inclusion of flexible C:N
stoichiometry (i.e., PS&SS) appeared to be an important feature allowing models to capture the
ecosystem response to climate variability through adjusting the C:N ratio of nonphotosynthetic
tissues or the whole-plant allocation among tissues with different C:N ratios (Zaehle & Friend,
2010). However, it is unclear whether those regulatory mechanisms exist in reality. Further
modelling approaches need more reliable framework to predict stoichiometric flexibility.

**4.2 Ecosystem N status and C residence time**
Ecosystem N status in models, including plant-available and unavailable N forms, is set by N
inputs from N fixation and N deposition, N losses from leaching and denitrification, and N gain
from the turnover of litter and SOM through tissue senescence and decomposition. As noted
above, external N cycle (i.e., N inputs and N losses) couples the N processes within the plant-
litter-SOM system, being mainly associated with the limitation of plant production. The effects
of ecosystem N status on C mean residence time (MRT), however, has been much less studied
than N limitation on productivity of plants and soil organisms, largely because these effects
involve various impacts on C transfer among pools and release from each pool via
decomposition and respiration (Thompson & Randerson, 1999; Xia et al., 2013). Therefore, the
different impacts of ecosystem N status induce oscillating N limitation on MRT due to the
inherently different assumptions of C-N interactions among three C-N coupling schemes.
At the steady state, the different effects of N status on changes in modelled MRT can be
attributed to: the different rate of soil N mineralization dependent on the total amount of N in
SOM and its turnover time, immobilization based on the competition strategy between plants and
microbes and their stoichiometry, and different deployment of reabsorbed N . The traceability





framework in this study can trace those different effects into three components (i.e., climate
forcing, N scalar and baseline MRT) based on three alternative C-N coupling schemes under the
TECO model framework.  Since the forcing data are identical, we assumed the same effects for
this component in all four experiments, which is thus not discussed further in this section.
In our study, the N scalar was based on the dynamics of C:N ratios (Eqn. 33). Therefore, N
scalar had no effect on MRT in SM2, resulting from the assumption of fixed C:N ratio in all C
pools (Fig. 6c). In both SM1 and SM3, however, the N scalar had large effects on the SOM pool,
which is probably related to different mechanisms. Specifically, N scalar in the SM1 had the
contrasting effects on MRT of fast and passive SOM pools (i.e., negative vs. positive,
respectively), which may largely be attributed to the plant and microbe competition strategy
combining with a much larger passive SOM pool in TECO-CN2.0 model (Du et al., 2017; Zhu et
al., 2017). Under N stress, the competition between plants and microbes is expected to be
intensified, resulting in increasing C:N ratio of nonphotosynthetic tissues (e.g., wood and root)
and the total C:N ratio. This effectively prevents N limitation of cell construction and
corresponds to an increase in whole-plant NUE (Thomas et al., 2015). In this case, higher C:N
ratio in those tissues lowers structural litter quality, leading to soil microbes to immobilize more
N to maintain their stoichiometric balance (Hu et al., 2001; Manzoni et al., 2010). However, in
the SM3, increased respiration acted as a mechanism to remove the excess accumulated C, which
is a stoichiometry-based implementation to prevent the accumulation of labile C under N stress
(Zaehle & Friend, 2010; Thomas et al., 2015). This mechanism promotes absorption and
respiration of the faster turnover pools (fast and slow SOM pools), leading to decrease in MRT
in these two pools.
In the traceability framework, the baseline MRT is determined by the potential decomposition
rates of C pools (C matrix), coefficients of C partitioning of NPP (B vector), and transfer
coefficients between C pools (*A* matrix, Eqn. [29]. Xia et al., 2013). The matrices *A* and *C* are
preset in the TECO model according to vegetation characteristics and soil textures (Weng and
Luo., 2008). Therefore, the notable spread in baseline MRT across the C-N schemes was induced
by the *B* vector, which was modified by different N-limitation assumptions. Conceptually, in
order to meet the N demand, plants adjust NPP allocation to N absorption tissues (e.g., roots). In
this study, three schemes all had similar trends of adjusting allocation C from wood to roots (Fig.
7), but with different mechanisms. For both SM1 and SM3, increased root C allocation was



mainly driven by N uptake capacity, which is associated with plant competitiveness in SM1 and
the respiration of excess labile C in SM3, respectively. However, for SM2, increasing root C
allocation may occur in spin-up stage from plant adjustment to whole-plant allocation among
tissues to fit fixed C:N ratio.

**5. Conclusions**
The C-N coupling has been represented in ecosystem and land surface models with different
schemes, generating great uncertainties in model predictions. The most striking difference
among terrestrial C-N coupling models occurs with the degree of flexibility of C:N ratio in
vegetation and soils, plant N uptake strategies, pathways of N import, and the representations of
the competition between plants and microbes for soil mineral N. In this study, we evaluated
alternative representations of C-N interactions and their impacts on C cycle using the TECO
model framework. Our traceability analysis showed that different representations of C-N
coupling processes lead to divergent effects on both plant production and C residence time, and
thus the ecosystem C storage capacity. The plant production are mainly affected by the different
assumptions on net ecosystem N exchange, plant N uptake, net N mineralization, and the C:N
ratio of vegetation and soil. In comparison, the alternative representations of the plant and
microbe competition strategy, combining with the flexible C:N ratio in vegetation and soils, led
to a notable spread effects on C residence time. Identifying the representations of main C-N
processes under different schemes can help us to improve the N-limitation assumptions
employed in terrestrial ecosystem models and forecasting future C sink dynamic in response to
climate change.

*Code availability*. The code for TECO-CN2.0 and the three C-N coupling schemes is available at
https://github.com/zgdu/TECO-CN-2.0-new.
*Data availability*. The data for this paper are available upon request to the corresponding author.
*Competing interests*. The authors declare that they have no conflict of interest.



**Acknowledgements**
This work was financially supported by the National Key R&D Program of China
(2017YFA06046), the National Natural Science Foundation of China (31770559, 31722009),
National 1000 Young Talents Program of China, and the Fundamental Research Funds for
Central Universities. Zhenggang Du also thanks the China Scholarship Council (201606140130)
for scholarship support.




**Figure legends**

**Figure 1.** Schematic diagram of the terrestrial ecosystem carbon (C) and nitrogen (N) coupling model (TECO-CN2.0). (A) Canopy module, (B) Plant growth module, (C) Soil water dynamics module, (D) Soil carbon-nitrogen coupling module. Rectangles represent the carbon and nitrogen pools. $R_a$, autotrophic respiration. $R_h$, heterotrophic respiration. Retr., re-translocation. NSC, nonstructural carbohydrate. MNP, mineral N in plant tissues. SOM, soil organic matter.

**Figure 2.** Schematic diagram illustrating the major carbon (C) and nitrogen (N) flows and stores in a terrestrial ecosystem. Light-blue arrows indicate C-cycle processes and red arrows show N-cycle processes.[1,2,3] alternative assumptions of N processes represent in scheme 1, 2 and 3, respectively. Met./Str. Litter, metabolic and/or structural litters; SOM, soil organic matter.

**Figure 3.** Simulated nitrogen fluxes and soil mineral nitrogen from three carbon-nitrogen coupling schemes (SM1, SM2 and SM3) in TECO-CN model for 1996 to 2007 at Duke Forest.

**Figure 4.** Simulated annual (a-f) and mean (g-l) carbon fluxes from carbon-only version and carbon-nitrogen coupled with three schemes (SM1, SM2 and SM3) of TECO model for 1996 to 2007 at Duke Forest. GPP, gross primary productivity; NPP, net primary productivity; NEE, net ecosystem exchange of $CO_2$; R-eco, ecosystem respiration; R-heter, heterotrophic respiration; R-auto, autotrophic respiration.

**Figure 5.** The nitrogen use efficiency (NUE) in three C-N schemes of TECO model (SM1, SM2 and SM3).

**Figure 6.** Simulation of annual ecosystem carbon storage capacity for 1996 to 2006 at Duke Forest by carbon in flux (NPP, x axis) and ecosystem residence time ($\tau_E$, y axis) in TECO model framework with three carbon-nitrogen coupling schemes (SM1, SM2 and SM3) and in TECO C-only model (C). Inset (a), ecosystem carbon residence time ($\tau_E$) in SM1, SM2, SM3 and C-only model; inset (b), mean ecosystem carbon storage simulated among SM1, SM2, SM3 and C-only model; inset (c), relative change of NPP and ecosystem residence time simulated among three schemes compared with in C-only model.

**Figure 7.** Determination of carbon-pool residence times based on traceability framework in TECO C-N model with three C-N coupling schemes (SM1, SM2 and SM3) and TECO C-only model (C). Panel (a), baseline residence time; panel (b), mean residence time, and panel (c), nitrogen scalar.

**Figure 8.** Coefficients of partitioning of NPP to nonstructural C (NSC), root, woody and leaf in C-only model (C) and C-N coupling model with three schemes (SM1, SM2 and SM3).

**Figure 9.** The sensitivity of nitrogen processes to NPP (panel a) and ecosystem residence time ($\tau_E$, panel b) among three carbon-nitrogen coupling schemes (SM1, SM2 and SM3). DRP, down-regulation photosynthesis; PS, plant tissue C:N ratio; PNU, plant N uptake; PMC: plant and microbe competition; BNF, biological N fixation; RtrN, re-tranlocation N; SS, soil pool C:N ratio.



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



**Table1.** Summary of the nitrogen-carbon coupling schemes used and the representation of key
processes in the carbon-nitrogen cycle.

|  | **SM1 (TECO-CN2.0)**[a] | **SM2 (CLM4.5)**[b,c] | **SM3 (O-CN)**[d,e] |
|---|---|---|---|
| **Photosynthesis down regulation by N availability (DRP)** | Based on the comparison between plant N demand and actual supply | Based on the available soil mineral N relative to the N demanded to allocate photosynthate to tissue | Based on foliage N concentration, which varies with N deficiency |
| **Plant tissue stoichiometry (PS)** | Flexible plant C:N ratio | Fixed plant C:N ratio | Flexible plant C:N ratio |
| **Plant N uptake (PNU)** | Based on fine root biomass, soil mineral N and N demand of plant.  Plants itself choose the strategy between uptake from soil mineral N and fix $N_2$ by comparing C investment | Based on N required to allocate NPP to tissue.  Plants uptake N for free | Combining active and passive uptake of mineral N based on fine root C, soil mineral N, plant transpiration flux, increases with increased plant N demand |
| **N competition between plants and microbes (PMC)** | Microbes have first access to soil mineral N | Based on demand by both microbial immobilization and plant N uptake | Microbes have first access to soil mineral N, the competitive strength of plants increases under nutrient stress |
| **Biological N fixation (BNF)** | Based on the nitrogen demand of plants and maximum N fixing ratio considering nutrient concentration | $f(NPP)$ | $f(ET)$ |
| **Deployment of re-translocated N (RtrN)** | Fixed fraction of litter | Based on available N in the tissue and the previous year's annual sum of plant N demand | Fixed fraction of dying leaf and root tissue |
| **Soil organic matter stoichiometry (SS)** | Flexible soil C:N ratio | Fixed soil C:N ratio | Flexible soil C:N ratio |
| **N leaching** | Function of soil mineral N pool and runoff | Function of soil mineral N pool and runoff | Function of soil mineral N and runoff |

[a]See this study; [b]Thornton et al. (2007), [c]Thornton et al. (2009); [d] Zaehle &Friend (2010),
[e]Zaehle et al. (2011).
C, carbon; N, nitrogen; NPP, net primary productivity; ET, evapotranspiration.



**Figure 1. TECO-CN 2.0**

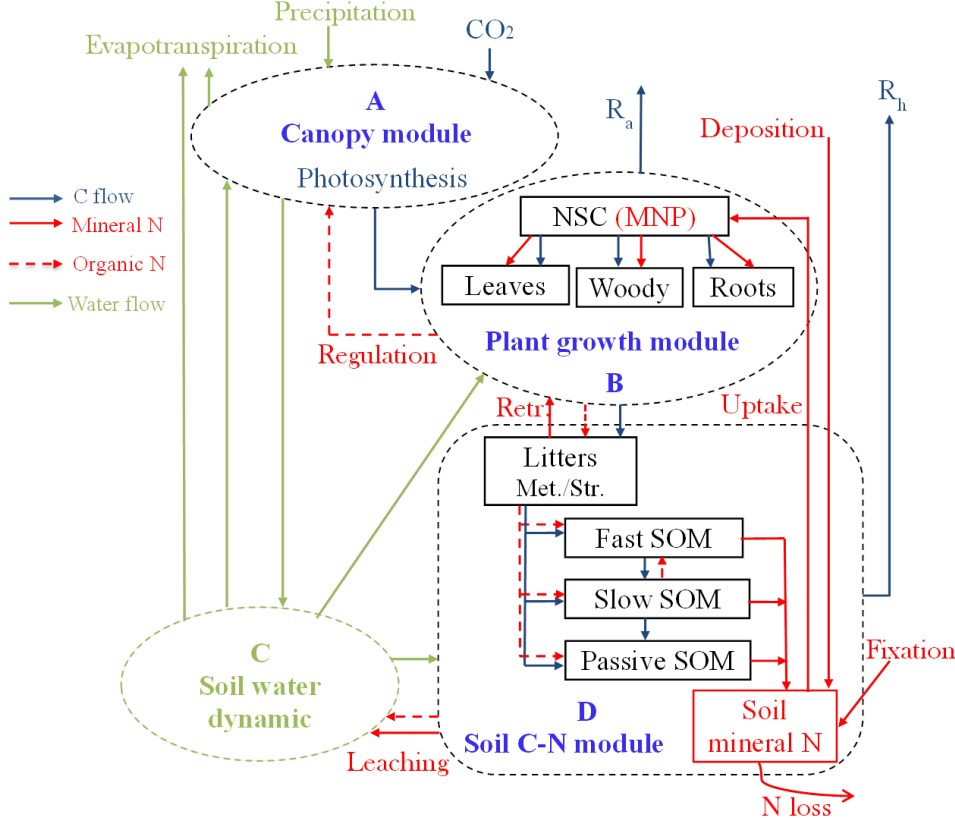


**Figure 1.** Schematic diagram of the terrestrial ecosystem carbon (C) and nitrogen (N) coupling
model (TECO-CN2.0). (A) Canopy module, (B) Plant growth module, (C) Soil water dynamics
module, (D) Soil carbon-nitrogen coupling module. Rectangles represent the carbon and nitrogen
pools. $R_a$, autotrophic respiration. $R_h$, heterotrophic respiration. Retr., re-translocation. NSC,
nonstructural carbohydrate. MNP, mineral N in plant tissues. SOM, soil organic matter.



**Figure 2**

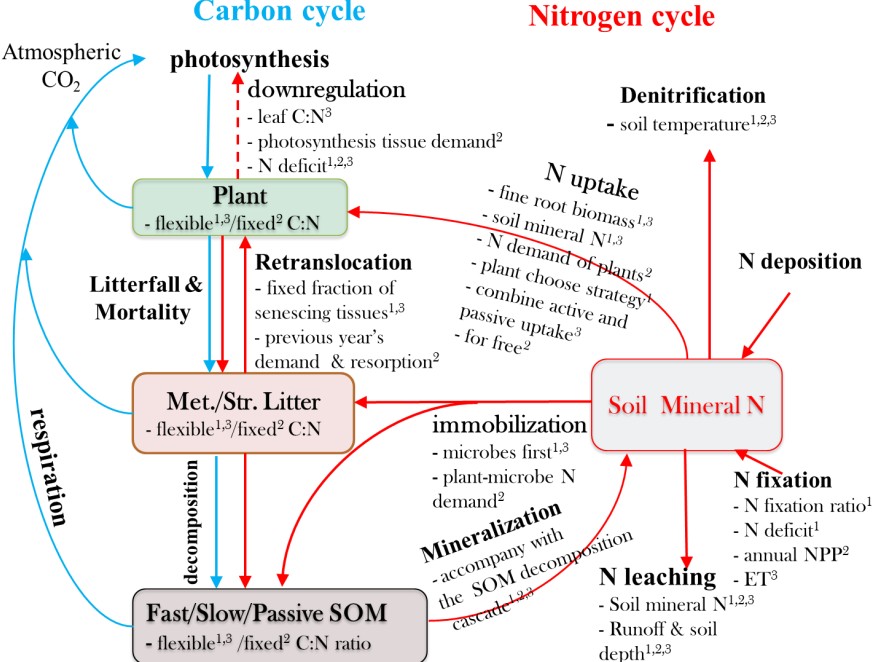



**Figure 2.** Schematic diagram illustrating the major carbon (C) and nitrogen (N) flows and stores
in a terrestrial ecosystem. Light-blue arrows indicate C-cycle processes and red arrows show N-
cycle processes.[1,2,3] alternative assumptions of N processes represent in scheme 1, 2 and 3,
respectively. Met./Str. Litter, metabolic and/or structural litters; SOM, soil organic matter.



**Figure 3**

**Figure 3.** Simulated nitrogen fluxes and soil mineral nitrogen from three carbon-nitrogen
coupling schemes (SM1, SM2 and SM3) in TECO-CN model for 1996 to 2007 at Duke Forest.



**Figure 4**

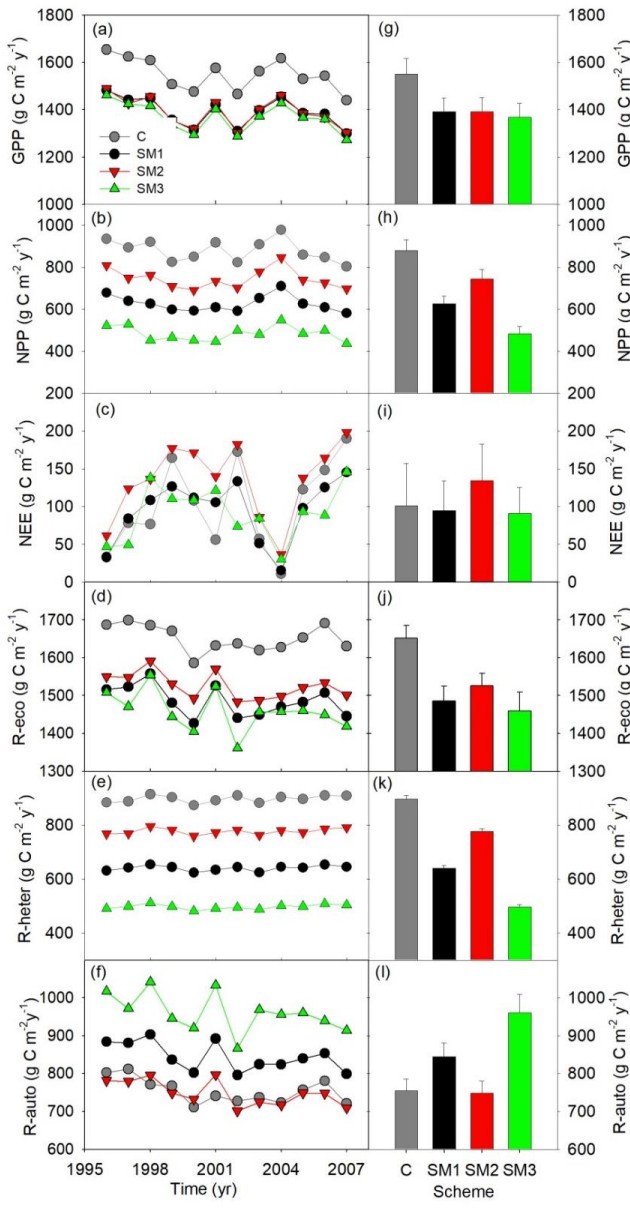


**Figure 4.** Simulated annual (a-f) and mean (g-l) carbon fluxes from carbon-only version and
carbon-nitrogen coupled with three schemes (SM1, SM2 and SM3) of TECO model for 1996 to
2007 at Duke Forest. GPP, gross primary productivity; NPP, net primary productivity; NEE, net
ecosystem exchange of $CO_2$; R-eco, ecosystem respiration; R-heter, heterotrophic respiration; R-
auto, autotrophic respiration.





**Figure 5**

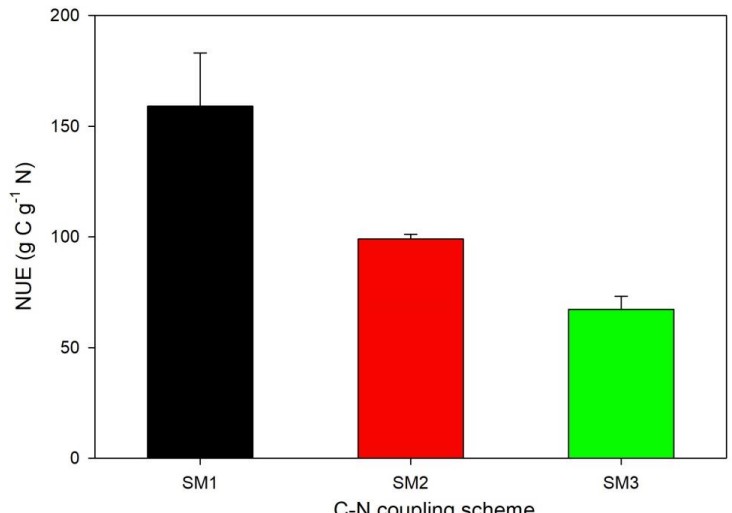


**Figure 5.** The nitrogen use efficiency (NUE) in three C-N schemes of TECO model (SM1, SM2
and SM3).



**Figure 6**

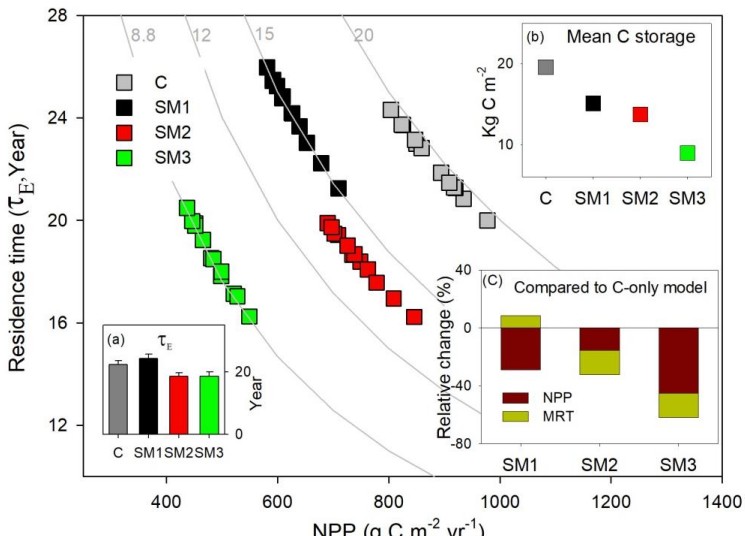


**Figure 6.** Simulation of annual ecosystem carbon storage capacity for 1996 to 2006 at Duke
Forest by carbon in flux (NPP, x axis) and ecosystem residence time ($\tau_E$, y axis) in TECO model
framework with three carbon-nitrogen coupling schemes (SM1, SM2 and SM3) and in TECO C-
only model (C). The hyperbolic curves represent constant values (shown across the curves) of
ecosystem carbon storage capacity. Inset (a), ecosystem carbon residence time ($\tau_E$) in SM1, SM2,
SM3 and C-only model; inset (b), mean ecosystem carbon storage simulated among SM1, SM2,
SM3 and C-only model; inset (c), relative change of NPP and ecosystem residence time
simulated among three schemes compared with in C-only model.





**Figure 7**

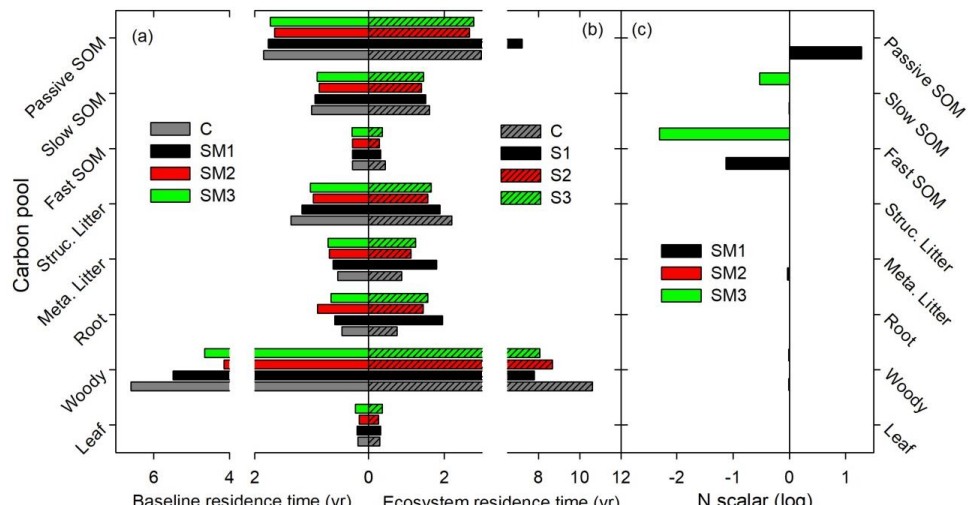


**Figure 7.** Determination of carbon-pool residence times based on traceability framework in
TECO C-N model with three C-N coupling schemes (SM1, SM2 and SM3) and TECO C-only
model (C). Panel (a), baseline residence time; panel (b), mean residence time, and panel (c),
nitrogen scalar.





**Figure 8**

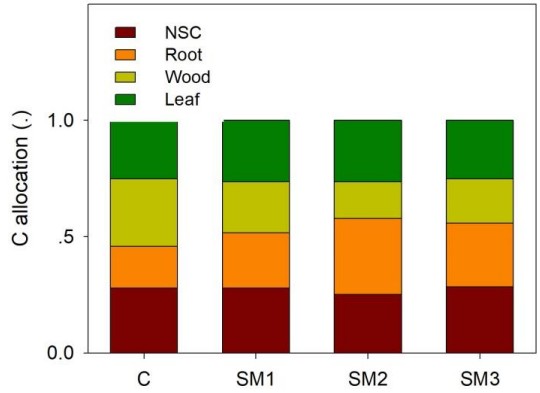


**Figure 8.** Coefficients of partitioning of NPP to nonstructural C (NSC), root, woody and leaf in
C-only model (C) and C-N coupling model with three schemes (SM1, SM2 and SM3).





**Figure 9**

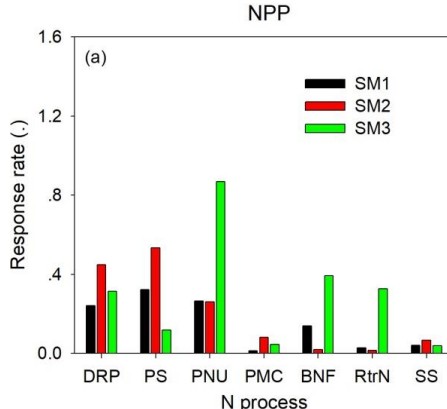
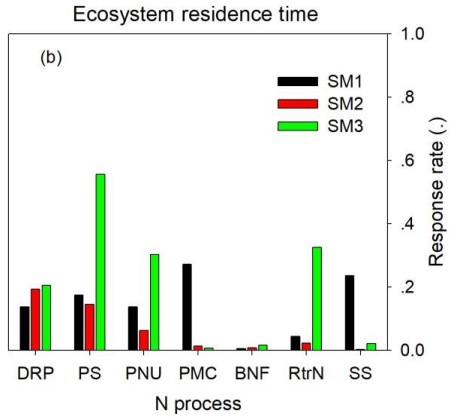


**Figure 9.** The sensitivity of nitrogen processes to NPP (panel a) and ecosystem residence time
($\tau_E$, panel b) among three carbon-nitrogen coupling schemes (SM1, SM2 and SM3). DRP, down-
regulation photosynthesis; PS, plant tissue C:N ratio; PNU, plant N uptake; PMC: plant and
microbe competition; BNF, biological N fixation; RtrN, re-tranlocation N; SS, soil pool C:N
ratio.