# Peer review of "Carbon-nitrogen coupling under three schemes of model representation: Traceability analysis"

_Geoscientific Model Development, 2018_

## Referee Comment (RC1) · W. Wieder (Referee) · 16 Apr 2018

General comments

Du and co-authors present a very interesting study using a matrix approach to compare the implementation of three distinct representations of C-N biogeochemistry in the TECO land model. The mathematical approach seems very powerful and the results are compelling.

I'd encourage the authors to unpack their results more to make findings more accessible to readers not familiar with any of the N schemes presented here. Refocusing

the text around big differences in assumptions being made between each modeling approach and how that translates to the different C stocks and fluxes would be very helpful.

The discussion only sparing refers to the display items presented in the results, making me wonder if the ideas being discussed are just the authors' opinions or if they can clearly be demonstrated by results presented here. On revision, please reference display items to support claims being made in the discussion.

Finally, there are enough grammatical errors to be distracting in the text. Some of these a highlighted in technical corrections, below, but revisions to the manuscript should be made for language fluency.

Specific comments

Line 60: For a paper that's more generally about the implementation and assumptions of C-N coupling in land models it strikes me as odd to lead off the introduction with an immediate nod to nitrogen fixation. Fixation is important, but leading off with a brief discussion sets up unrealistic expectations for the reader for what's ultimately being discussed in the paper.

Line 84: References are needed to support these claims, as it seems to conflate C cycle uncertainty (e.g. Arora et al. 2013) with C-N representation in models, which is not accurate

Line 86: Similarly, references are needed as the 'contradictory results' from implementation of C-N models have not been clearly established in the literature

Line 97: I may be forgetting something, but don't recall the Xia et al (2013) paper accomplishing all that it's being credited for here. Maybe other references are needed where the authors demonstrate how the matrix approach has been used for 'benchmark analyses, model intercomparisons, and data model fusion, and improved model predictive power'? Otherwise revise this sentence to avoid implying a single paper did

all this work.

Figs 1 & 2. How is mineral N retranslocated from the litter pool? After a leaf has fallen do plants still have access to this N? Doesn't retranslocation occur before senescence?

Fig 2. I really appreciate the effort to clearly spell out different assumptions between different C-N coupling schemes and map onto the structure of TECO's C and N pools. I fear this figure is too jumbled with small, tilted text to be useful, and would encourage authors to spend some time cleaning up this display item so it's more clear & useful.

From the description in the methods, it seems like the entire coupling of C-N biogeochemistry occurs through the different implementation of the N scalar from each scheme (Eq. 30). Is this true? If so, documenting how the aspects summarized in Table 1 are actually being implemented seems important (either in the main text, SI, or an appendix). If this is where the magic happens it should be clearly spelled out using language from the N related (red) text in Fig 1.

In previous work this author group has demonstrated that the matrix approach gives identical results to the conventional system of differential equations. Can a similar plot be made with a CN version of TECO? That is, can lumping a coupled C:N model into a "N scalar" (eq. 33) account for everything that's going on in the model? I'm assuming it can, but this is never clearly demonstrated in the results.

Besides difference in NUE (Fig. 5) I'm struck by the differences in carbon use efficiency (CUE, the ratio of NPP:GPP) among N models that's attributable to large difference in autotrophic respiration among models. Is this worth displaying or discussing further?

Why did SM1 increase the mean residence time of C relative to the control model (Figs 6 inset & 7). I'm assuming it's because of N 'limitation' of passive C turnover? Does this seem realistic? It must be caused by relatively quick turnover of this pool and an low C:N ratio of SOM in SM1, or low respiration coefficient in fluxes between slow and passive pools that are driving a high immobilization flux in SM1 (Fig. 3)?

Alternatively, does the stoichiometry of litter quality drive these results? More details on these mechanisms seem worth discussing?

Figures 7 and 9 seem like really interesting, powerful strengths of the tractability analysis presented here. In my estimation there's not nearly enough text in the results or discussion to walk readers through what's being shown here. Unpacking the information communicated in these figures would help readers access what's being shown and how the tractability analysis helps us understand differences among model formulations. (Note, some of this could even fall into the introduction and methods by foreshadowing key differences among model formulations that are important to the results presented here from the start).

Line 508: If this is the most striking difference, is there a take home figure that clearly communicated this message? As presented, I'm not sure this conclusion is well supported by the results or discussion.

Technical corrections

Line 37: For clarity, replace 'them' with 'the three C-N coupling schemes'

Line 43: Consider replacing 'divergent' with 'differences in'?

Line 58 & 64: Avoid starting a sentence with an abbreviation, that is write out 'Nitrogen'.

Line 59: 'Requires' should be plural

Line 66, I'd add Hungate et al. (2003) to this list of references

Line 71: It seems odd to talk about progressive N limitation as occurring with "growth enhancement when N mineralization increases". Is Dr. Luo comfortable with this definition?

Line 72: Awkward. Please revise for fluency & clarity

Line 80: These are from Cleveland et al (1999), not my work, and their implementation

in models is summarized nicely by Meyerholt et al. (2016).

Line 129: Should this be 'data', not 'date'? Also from what plots, the paragraph starts off discussing the AmeriFlux tower, but are the biomass data from the control FACE plots?

Line 138, 180: I'm a little confused. Is this the first publication of TECO-CN2.0, if so they should be referenced? If not, are there other versions of TECO-CN and how does the implementation of C-N biogeochemistry differ in the present model?

Table 1: References to Thorton et al are actually for CLM4cn (not CLM4.5bgc, as implied in the table). The implementation of C-N biogeochemistry is similar in each model, but the structure and stoichiometry of SOM pools are different in each? Please clarify in the text and references which version of the model is used for SM2.

Fig. 1. It seems odd to have N fixation going directly to soil mineral N pools. I realize that CLM (and likely other models) do this, but the simplification should at least be noted in the text?

Fig. 1 Why doesn't the soil C-N module need to take up mineral N? This seems to contradict Fig. 2, and could be corrected with two-sided arrows?

Throughout section 2.2.2 should units for fluxes be communicated?

Eq. 19. This would give a fixation flux in gN/m2/s, but TECO doesn't work on that time step?

Line 321. What are all these abbreviations? Regardless, there's too many here to be coherent, and I'd encourage these to be written out fully throughout the text

Line 349. These differences are relative to the C only control? If so restating this here may help clarify?

Line 351 this sentence is awkward and needs to be revised?

Line 396: this list of abbreviations is neither intuitive, commonly used, or helpful. I find the later use of the abbreviations confusing and recommend just writing out the processes being discussed in full.

Line 420: doesn't SM2 use NPP to calculate BNF rates?

Lines 445-450: Where are these results shown in the work presented here?

Line 463: where are these oscillations shown in the work presented?

Line 473: This line really makes me wonder if the approach outlined here is 'right'? Regardless, it makes me think that differences among models are 100% attributable to differences in stoichiometric assumptions among models. If so, should a list of pools and their C:N ratio SM1, 2, and 3 be communicated?

Line 483: Ah, so win SM1, is there a progressive decline in litter quality that ends driving high soil N demand as the decomposition cascade tries to meet stoichiometric demand, whereas SM3 allow this extra C to be blown off through heterotrophic respiration? Alternatively, is it higher autotrophic respiration in SM3 (through increased fine root C allocation) that allows the extra C to be blown off (line 501) Sorry, I'm not familiar enough with all of these approaches to understand what each model is doing.

Line 488 what's being absorbed?

Line 490: I'm still confused about what's causing differences between SM1 and SM3. For readers less familiar with these schemes can the difference between the approached be unpacked a bit more, as this seems like a powerful strength of the traceability analysis?

References: Arora et al. 2103. Journal of Climate doi:10.1175/JCLI-D-12-00494.1 Cleveland et al. 1999. Global Biogeochemical Cycles doi: 10.1029/1999GB900014 Hungate et al. 2003. Science DOI: 10.1126/science.1091390 Meyerholt et al. 2016. Biogeosciences doi: 10.5194/bg-13-1491-2016

---

## Referee Comment (RC2) · Anonymous Referee #2 · 26 Apr 2018

[General comments] In this paper, the authors evaluate three different schemes of Carbon-Nitrogen coupling in a terrestrial model, which can largely change both C and N dynamics reproduced by models. For this, they used an existent framework for analyzing the difference between the models. This paper is clearly written, and the results are informative for readers. I recognize the importance of this study because CN coupling is one of the emergent processes to be evaluated / constrained in such land ecosystem modeling. However, I think there are places to be improved: the figures are informative, but the explanation is not enough for readers. My comments will not require a lot of effort to improve.

[Figure]

[Detailed comments] - P6, L129: "biomass production date" should be "biomass production rate" ?

- P6, L129: What purpose the data "standing biomass and biomass production date" used for your study? Do you mean the datasets are used to determine the parameters associated with the processes? In addition, CN concentration for plant and soil (Finzi et al., and Lichter et al.) are also used for your analysis (I suppose the SM2 simulation need such data because of the fixed CN ratio, but it is not clear in the text). Please clarify them.

- P6, L138: It might be better to clearly mention first that the model is newly developed and used in this study for the first time.

- P7, eq(1)-(6): The detail description of C allocation scheme of TECO-CN v2 is shown here, but it seems the equations are not referred in other places. In my simple thinking, the detail descriptions with the equations are not necessary for your analysis, and it looks no problem if your put them into supplement. If you want to keep the eqs in the main body, it should be qualitative explanations how the C allocation scheme act on CN dynamics in simulations.

- P8, L177- : Although it is shown in the Table 1, it will be helpful for readers to mention here again the fact that CN ratio in SM2 scheme is fixed, while other two are flexible.

- P12, L309: Which level of $CO_2$ concentration do you give to the model in the spin-up? Are the $CO_2$ concentration and climate forcing in simulations given as a cyclic manner? Please clarify them.

- P13, L319: "SˆCRT" should be "SˆMRT" ?

- P15, L390: It looks less references to your figures and tables in the discussion section: It was a bit difficult for me to figure out which claims in the discussion section are supported by your own results.

- P15, L405: You mention here that SM1 has a feedback from leaf N concentration

to photosynthetic capacity, but eq.(7) seems not. I have overlooked something, but if the SM1 actually has leaf-N concentration feedback, you should touch it in the section 2.1.1.

- P16, L426 "C cost of fixing": Is the effect of C cost actually considered in your simulation of SM3? Which equation in the section 2.2.2 represents the effect? In addition, if you consider the C cost in the SM3 simulation, does the lowest NPP of SM3 attribute to the increase of autotrophic respiration in SM3? It would be nice if you can discuss on this.

- P16, L427: I will appreciate if you can add more explanation why BNF of SM1 lead to the highest NUE. In my understanding, if BNF in SM1 works as the complement to nitrogen uptake, the process works to increase the uptake, and then the NUE(=NPP/PNU) should be decreased. I wonder the SM1 has a mechanism to have BNF that satisfy a minimum N requirement by plants, but it was not clear.

- P16, L428: Although same N loss process are shared between the schemes, I suppose the original models (TECO-CN/CLM/OCN) actually differ in that point. Readers can get benefit if you can discuss it briefly.

- P17, L443: You discuss here how CN ratio in SM1 scheme affects the N regulation on plant production processes. As you discuss in the section 4.2, SM3 also has the mechanism of flexible CN ratio. How did the flexibility of SM3 act on plant production processes?

- P17, L445 "leading to a high microbial N immobilization": I cannot understand why high CN ratio in plant tissues bring models to have a high microbial N immobilization. Need further detail.

- P18, L475 "Fig. 6c" is likely to be "Fig. 7c" ?

- P18, L484: What does "structural litter quality" mean?

- P18, L498: Maybe "Fig. 7" is likely to be "Fig. 8".

- In the analysis, plant production and C/N status are evaluated in steady state. Although I recognize the usefulness of the analysis using steady states, I believe many readers get interested how your conclusions can be extended to non-steady state simulations, because N limitation on C cycle can be intensified in the condition where $CO_2$ concentration increasing. I will be happy if I can see the discussion on this. In addition, displaying N status in the three simulations will be helpful for readers to get the whole picture of the CN dynamics: mineral N is displayed (in Fig.3), but others (plant, litter, and SOM) are not. Since your analysis is based on steady-state, such information can be a support to understand the relationship between N-fluxes and N-pools. My suggestion is to include it in supplement.

---

## Author Response (AR1)

Tomomichi Kato

University of Tsukuba, Tsukuba, Ibaraki, Japan

Handling Topical Editor, Geoscientific Model Development (GMD)

Re: GMD-2018-41

Dear Dr. Kato,

Thanks so much for sending us two referees' comments and suggestions on our manuscript
"Carbon-nitrogen coupling under three schemes of model representation: Traceability
analysis" (GMD-2018-41). We greatly appreciate the two reviewers for their valuable
comments and suggested amendments. Their inputs have helped improve the paper
tremendously. We have carefully studied the comments from the reviews and made revisions
based on them in this version of manuscript.

In the revised manuscript, we have addressed all the comments from the two referees.
Specifically, we added the Figure 5 for annual averaged size and C:N ratio of each C pool
and the Figure 6b for the CUE in the C-only version and the three C-N schemes of TECO
model. We also have added a new figure (Figure S2) in the supplemental information to show
the annual averaged N content for each C pool among the three C-N schemes. In the
Materials and Methods and Results sections, we also have shown the different effects of C-N
coupling hypotheses among three simulations as suggested by both referees. We clarified our
discussions based on more referenced results as suggested by the referee #1 and discussed the
differences with the original models (CLM4.5bgc and O-CN) as suggested by referee #2. We
greatly appreciate the suggestions from the reviewers, as addressing them has strengthened
the manuscript.

We confirm that all authors have met the authorship criteria.

We also declare that the submitted work is our own and that copyright has not been breached
in seeking its publication.

Here are our detailed responses to the reviews. Please note that the comments from the
referees are in *italics* followed by our responses in **regular** text.

We hope you will find our revision satisfactory for publication in *Geoscientific Model*
*Development*.

Yours Sincerely,

Xuhui & Jianyang

Xuhui Zhou, Jianyang Xia

School of Ecological and Environmental Sciences, East China Normal University

500 Dongchuan Road, Shanghai 200062, China

Email: xhzhou@des.ecnu.edu.cn, jyxia@des.ecnu.edu.cn

**Response letter to comments (gmd-2018-41)**

**Will Wieder's comment (Referee #1)**

*General comments*

*Du and co-authors present a very interesting study using a matrix approach to compare the*
*implementation of three distinct representations of C-N biogeochemistry in the TECO land*
*model. The mathematical approach seems very powerful and the results are compelling.*

[**Response**] Thanks so much for your positive comment.

*I'd encourage the authors to unpack their results more to make findings more accessible to*
*readers not familiar with any of the N schemes presented here. Refocusing the text around*
*big differences in assumptions being made between each modeling approach and how that*
*translates to the different C stocks and fluxes would be very helpful.*

[**Response**] Thanks very much for your suggestions. In this revised version, we added more
results (e.g., C pool sizes and C:N ratio in Figure 5, CUE in Figure 6b, the sensitivity of N
processes to ecosystem C storage capacity in Figure 10c, and N pool sizes in Figure S2) to
support our findings. Based on our results, we displayed the different N and C fluxes under
different C-N schemes (Figures 3 and 4) and the different C and N status among plant tissues,
litter and soil pools (Figures 5 and 6) as well as the ecosystem C storage capacity (Figure 7).
To evaluate the alternative representations of C-N processes dominating the ecosystem C
storage capacity, we applied the traceability analysis framework to trace the key factors in
different schemes. We found that different process assumptions caused divergent C residence
time and plant production among different C-N schemes in this study (Figures 8-10). We
added the detailed information and discussion in both Result and Discussion sections in Lines
372-381, 386-389, 425-428 and 498-501.

*The discussion only sparing refers to the display items presented in the results, making me*
*wonder if the ideas being discussed are just the authors' opinions or if they can clearly be*
*demonstrated by results presented here. On revision, please reference display items to*
*support claims being made in the discussion.*

[**Response**] Sorry for the confusion. We carefully revised the whole manuscript and also
referenced more necessary results in the Discussion section accordingly. As a consequence,
our manuscript has been considerably improved. We hope you will find our revision
satisfactory.

*Finally, there are enough grammatical errors to be distracting in the text. Some of these a*
*highlighted in technical corrections, below, but revisions to the manuscript should be made*
*for language fluency.*

[**Response**] We carefully revised the manuscript according to the comments, paid attention to the grammar, and made necessary changes. We also asked a native English speaker (Mrs. Megan C. Foster) to revise the whole manuscript. Please see below for the detailed responses point by point. As a consequence, our manuscript has been considerably improved. We hope you will find our revision satisfactory.

*Specific comments*

*Line 60: For a paper that's more generally about the implementation and assumptions of C-N coupling in land models it strikes me as odd to lead off the introduction with an immediate nod to nitrogen fixation. Fixation is important, but leading off with a brief discussion sets up unrealistic expectations for the reader for what's ultimately being discussed in the paper.*

[**Response**] Thanks for your comments and suggestions. We deleted the description of nitrogen fixation and have rephrased this paragraph carefully, especially emphasizing the processes of carbon-nitrogen coupling in affecting the terrestrial ecosystem C storage.

*Line 84: References are needed to support these claims, as it seems to conflate C cycle uncertainty (e.g. Arora et al. 2013) with C-N representation in models, which is not accurate.*

*Line 86: Similarly, references are needed as the 'contradictory results' from implementation of C-N models have not been clearly established in the literature.*

[**Response**] We revised the descriptions of the related references (Arora et al., 2013; Zaehle et al., 2015; Sokolov et al., 2008; Wania et al., 2012; Walker et al., 2015) and added them in our revised manuscript in Lines 85 and 89.

*Line 97: I may be forgetting something, but don't recall the Xia et al (2013) paper accomplishing all that it's being credited for here. Maybe other references are needed where the authors demonstrate how the matrix approach has been used for 'benchmark analyses, model intercomparisons, and data model fusion, and improved model predictive power'? Otherwise revise this sentence to avoid implying a single paper did all this work.*

[**Response**] Thanks for your comments and suggestions. We added some references and modified the sentence as "The traceability analysis has been developed to diagnose the simulation results within (Xia et al. 2013; Ahlström et al., 2015) and among (Rafique et al., 2016; Zhou et al., 20) models.".

*Figs 1 & 2. How is mineral N retranslocated from the litter pool? After a leaf has fallen do plants still have access to this N? Doesn't retranslocation occur before senescence?*

[**Response**] Sorry for the mistake. The mineral N was retranslocated to other tissues before
the live tissues (i.e., leaves, fine roots and live stems) senescence in TECO model. We simply
added an arrow to plant growth module to represent the retranslocation of the mineral N to
other tissues in the Figs. 1 and 2. We described it in Lines 158.

*Fig 2. I really appreciate the effort to clearly spell out different assumptions between*
*different C-N coupling schemes and map onto the structure of TECO's C and N pools. I fear*
*this figure is too jumbled with small, tilted text to be useful, and would encourage authors to*
*spend some time cleaning up this display item so it's more clear & useful.*

[**Response**] Thanks so much for your suggestions. We deleted all the numbers and rearranged
the text in the figure to clarify the display.

*From the description in the methods, it seems like the entire coupling of C-N biogeochemistry*
*occurs through the different implementation of the N scalar from each scheme (Eq. 30). Is*
*this true? If so, documenting how the aspects summarized in Table 1 are actually being*
*implemented seems important (either in the main text, SI, or an appendix). If this is where the*
*magic happens it should be clearly spelled out using language from the N related (red) text in*
*Fig 1.*

[**Response**] Sorry for the confusion. The N scalar is set as the respiration and decomposition
rate modifier, which considers the changes of N content to compare with the initial condition
(Eq. 33). Depending on both the N supply and loss for each C pool, the N scalar mainly
affects the C residence time directly (Fig 7). The different aspects among three C-N coupling
schemes introduce different effects on N supply and losses directly and/or indirectly, and thus
affect the C residence time via N scalar. Beside the N scalar, the different representations
summarized in Table 1 also introduce other aspects to affect the C storage. For example, the
different implementations of the N down regulation have differently constrained power on
GPP (although those powers were not significant in this study); the different assumptions on
tissue C:N ratio led to different C allocation ratio (eq.1-6) and further affect the baseline
residence time (eq.30); the different representations of plant N uptake and biological N
fixation result in different C investment, and thus the different autotrophic respiration. In this
revised version, we added those description and discussion in both Method and Discussion
sections in Lines 319, 406-409, and 529-536.

*In previous work this author group has demonstrated that the matrix approach gives*
*identical results to the conventional system of differential equations. Can a similar plot be*
*made with a CN version of TECO? That is, can lumping a coupled C:N model into a "N*
*scalar" (eq. 33) account for everything that's going on in the model? I'm assuming it can,*
*but this is never clearly demonstrated in the results.*

[**Response**] That version of TECO-CN had incorporated the "N scalar" into the respiration
and decomposition rate modifier (Du et al. 2017), which had been used in the previous work
(e.g., Zaehle et al., 2014). In this study, N scalar is a key factor, and we separated it from the
environmental scalar ($\xi_E$) and baseline carbon residence time ($\tau_E^{'}$) in the traceability analysis framework to trace the different effects that were introduced by the three C-N schemes. We
also compared our TECO-CN version with the version used in Zaehle et al., 2014. We found
that the results matched well (See Figure R1 below).

[Figure]

Figure R1. Comparisons of GPP, NPP, ecosystem C storage and ecosystem N storage at the
steady state from this study vs. the TECO-CN version used previous work.

*Besides difference in NUE (Fig. 5) I'm struck by the differences in carbon use efficiency*
*(CUE, the ratio of NPP:GPP) among N models that's attributable to large difference in*
*autotrophic respiration among models. Is this worth displaying or discussing further?*

[**Response**] Thanks for your suggestions. Yes, we found that carbon use efficiency (CUE)
varied among three N schemes. The SM2 has the highest CUE while SM3 has the lowest
CUE among three C-N schemes. We added this result in Fig 5b. The direct factors of those
differences mainly attribute to difference in autotrophic respiration and N limitation on
production (i.e., down-regulation effect). For the SM2, plant uptake N does not need to cost C,
which lead to the highest CUE. In the SM3, however, the lowest CUE is due to both the C
cost of plant actively uptake N and the assumption that increases respiration to remove the
excess C. In this revised version, we added those Results and Discussion sections in Lines
386-389 and 498-501.

*Why did SM1 increase the mean residence time of C relative to the control model (Figs 6*
*inset & 7). I'm assuming it's because of N 'limitation' of passive C turnover? Does this seem*
*realistic? It must be caused by relatively quick turnover of this pool and an low C:N ratio of*
*SOM in SM1, or low respiration coefficient in fluxes between slow and passive pools that are*
*driving a high immobilization flux in SM1 (Fig. 3)? Alternatively, does the stoichiometry of*
*litter quality drive these results? More details on these mechanisms seem worth discussing?*

[**Response**] Thanks so much for your comments and suggestions. Yes, the slower turnover
rate of passive SOM pool dominated a longer mean ecosystem residence time in SM1
compared with those in C-only version. Our results showed that lower heterotrophic
respiration rate (Figure 4) and C:N ratio of passive SOM (Figure 5b) as well as higher
immobilization flux (Figure 3) jointly 'limited' the turnover rate of passive SOM pool. For
the SM1, the microbe immobilization dominates a low C:N ratio and then affects the
decomposition cascade for passive SOM (Fig 8). The reason is that the representation of N
immobilization in TECO-CN has the potential to accumulate N:

$$Imm_N = \begin{cases} \sum_{i=4}^{8} min\left(\left(\frac{C_i}{CN0_i} - \frac{C_i}{CN_i}\right), 0.1 * SN_{min}\right) \ for \ CN_i \geq CN0_i \\ \sum_{i=4}^{8} min\left(\left(\frac{C_i}{CN_i} - \frac{C_i}{CN0_i}\right), 0.1 * SN_{min}\right) \ for \ CN_i < CN0_i \end{cases}$$

We added this equation and more information in Method and Discussion sections.

*Figures 7 and 9 seem like really interesting, powerful strengths of the tractability analysis*
*presented here. In my estimation there's not nearly enough text in the results or discussion to*
*walk readers through what's being shown here. Unpacking the information communicated in*
*these figures would help readers access what's being shown and how the tractability analysis*
*helps us understand differences among model formulations. (Note, some of this could even*
*fall into the introduction and methods by foreshadowing key differences among model*
*formulations that are important to the results presented here from the start).*

[**Response**] Thanks for your comments and suggestions. In the revised version, we
reorganized the information communicated in these figures carefully, mainly tracing how the
different hypotheses among C-N coupling schemes modulate the ecosystem C storage based
on traceability analysis. We hope that you satisfy our revision.

*Line 508: If this is the most striking difference, is there a take home figure that clearly*
*communicated this message? As presented, I'm not sure this conclusion is well supported by*
*the results or discussion.*

[**Response**] Sorry for the confusion. Originally, we used the sensitivity of N processes to
NPP and ecosystem residence time ($\tau_E$) among three C-N coupling schemes to display this difference, which was shown in the previous Fig. 9. In the revised version, we extended this
sensitivity to ecosystem C storage (NPP $\times$ $\tau_E$) in Fig. 10 based on the different
representations among three C-N schemes. We emphasized the difference and added more
discussion in Lines 425-428 and 501-505.

***Technical corrections***

*Line 37: For clarity, replace 'them' with 'the three C-N coupling schemes'*

*Line 43: Consider replacing 'divergent' with 'differences in'?*

*Line 58 & 64: Avoid starting a sentence with an abbreviation, that is write out 'Nitrogen'.*

*Line 59: 'Requires' should be plural*

*Line 66, I'd add Hungate et al. (2003) to this list of references*

[**Response**] Done as suggested.

*Line 71: It seems odd to talk about progressive N limitation as occurring with "growth*
*enhancement when N mineralization increases". Is Dr. Luo comfortable with this definition?*

[**Response**] Sorry for the confusion. We revised the sentence as "Early C-N coupled models
demonstrated that the N availability limits ecosystem C storage capacity with associated
effects on plant photosynthesis and growth in many terrestrial ecosystems…"

*Line 72: Awkward. Please revise for fluency & clarity.*

[**Response**] Sorry for the confusion. We revised this sentence as "Recent studies have largely
confirmed these results by improving C-N coupling models with multiple hypotheses."

*Line 80: These are from Cleveland et al (1999), not my work, and their implementation in*
*models is summarized nicely by Meyerholt et al. (2016).*

[**Response**] Thanks for pointing out our mistake. We added these two references and replaced
the "Wieder et al., 2015" to "Wieder et al., 2015a".

*Line 129: Should this be 'data', not 'date'?*

[**Response**] Sorry for the mistake. We replaced "date" by "data".

*Also from what plots, the* meteorological *paragraph starts off discussing the AmeriFlux tower,*
*but are the biomass data from the control FACE plots?*

[**Response**] Sorry for the confusion. The forcing data used in this study were taken from the
AmeriFlux database, while the biomass data were taken from the reference study. To clarify
this point, we revised the first sentence of this paragraph as "The forcing data used in this
study were taken from the Duke free-air $CO_2$ enrichment (FACE) experiment….".

*Line 138, 180: I'm a little confused. Is this the first publication of TECO-CN2.0, if so they*
*should be referenced? If not, are there other versions of TECO-CN and how does the*
*implementation of C-N biogeochemistry differ in the present model?*

[**Response**] Sorry for the confusion. There are two versions of TECO-CN model. The first
version was used in Zaehel et al., 2014 and this study, and the second version is a simplified
version used for data assimilation (e.g., models in Shi et al., 2015 and Du et al., 2017). Both
versions are the variant of the TECO-C version published in Weng and Luo, 2008. To make it
clear in this study, we replaced "TECO-CN" with "TECO-CN2.0" accordingly.

*Table 1: References to Thorton et al are actually for CLM4cn (not CLM4.5bgc, as implied in*
*the table). The implementation of C-N biogeochemistry is similar in each model, but the*
*structure and stoichiometry of SOM pools are different in each? Please clarify in the text and*
*references which version of the model is used for SM2.*

[**Response**] The version of CLM4.5bgc is used for SM2 in this study.  We changed the
references as "Koven et al., 2013" and "Oleson et al., 2013".

*Fig. 1. It seems odd to have N fixation going directly to soil mineral N pools. I realize that*
*CLM (and likely other models) do this, but the simplification should at least be noted in the*
*text?*

[**Response**] Thanks for your comments. We added a new dotted arrows from N fixation to
plant part in Figure1 and the description "*set N fixation as an option when the plant N
uptake is enough for growth in terms of C investment" in the legend of Figure1.

*Fig.1 Why doesn't the soil C-N module need to take up mineral N? This seems to contradict*
*Fig. 2, and could be corrected with two-sided arrows?*

[**Response**] Thanks for pointing out our mistake. As suggested, we replaced those one-sided
arrows with two-sided arrows in Figure 1.

*Throughout section 2.2.2 should units for fluxes be communicated?*

[**Response**] Thanks for pointing out what we have neglected. The units were added in the
revised version.

*Eq. 19. This would give a fixation flux in gN/m2/s, but TECO doesn't work on that time step?*

[**Response**] Yes, the unit of biological N fixation flux is g N $m^{-2}$ $s^{-1}$. We added it in the
revised version.

*Line 321. What are all these abbreviations? Regardless, there's too many here to be coherent,*
*and I'd encourage these to be written out fully throughout the text.*

[**Response**] Thanks for your comments and suggestions. We deleted "i.e., DRP, PS, PUN,
PMC, BNF, RtrN and SS" in this section.

*Line 349. These differences are relative to the C only control? If so restating this here may*
*help clarify?*

[**Response**] Yes, these differences are relative to the results of TECO-C. In the revised
version, we added "by comparison with the TECO-C version" in this sentence.

*Line 351 this sentence is awkward and needs to be revised?*

[**Response**] Sorry for the confusion. In the revised version, we deleted this sentence "The
NPP and plant N uptake (PNU) jointly determine the N use efficiency (NUE)."

*Line 396: this list of abbreviations is neither intuitive, commonly used, nor helpful. I find the*
*later use of the abbreviations confusing and recommend just writing out the processes being*
*discussed in full.*

[**Response**] Sorry for the confusion. As suggested, we wrote out these processes in full and
deleted these abbreviations in this section.

*Line 420: doesn't SM2 use NPP to calculate BNF rates?*

[**Response**] Thanks for pointing out our mistake. Yes, SM2 used NPP not ET to calculate
BNF rate in this study. We revised the sentence as "… SM2 and SM3 simulated BNF
explicitly, which used the modified empirical relationships of BNF with NPP and
evapotranspiration (ET), respectively.".

*Lines 445-450: Where are these results shown in the work presented here?*

[**Response**] Thanks for pointing out what we have neglected. After we added a new figure
(Figure 5) about C pools and their C:N ration for different treatments, these results are mainly
shown in Figure 3 and Figure 5. We revised those sentences as "N stress increased tissue C:N
ratio (Figure 5b), leading to a high microbial N immobilization (Figure 3) and then a lower
net N mineralization (Fig 3a, g and m), which allowed plant cell construction with a lower N
requirement. The inclusion of flexible C:N stoichiometry appeared to be an important feature
allowing models to capture the ecosystem response to climate variability through adjusting the C:N ratio of nonphotosynthetic tissues or the whole-plant allocation among tissues
(Figure 9) with different C:N ratios…".

*Line 463: where are these oscillations shown in the work presented?*

[**Response**] We added the related results in this sentence as "Therefore, the different impacts
of ecosystem N status induce oscillating N limitation on MRT (Figure 8) due to the
inherently different assumptions of C-N interactions among three C-N coupling schemes".

*Line 473: This line really makes me wonder if the approach outlined here is 'right'?*
*Regardless, it makes me think that differences among models are 100% attributable to*
*differences in stoichiometric assumptions among models. If so, should a list of pools and their*
*C:N ratio SM1, 2, and 3 be communicated?*

[**Response**] Thanks for your comments. We added a new figure in the revised version (Figure
5). Please see below for details.

[Figure]

**Figure 5.** The annual average sizes of carbon pools (panel a) at the steady-state among 1996-
2007 for C-only version and the three C-N schemes (SM1, SM2 and SM3) and the C:N ratio
(panel b) of each carbon pools for the three C-N schemes (SM1, SM2 and SM3) in TECO-
CN model.

*Line 483: Ah, so win SM1, is there a progressive decline in litter quality that ends driving*
*high soil N demand as the decomposition cascade tries to meet stoichiometric demand,*
*whereas SM3 allow this extra C to be blown off through heterotrophic respiration?*
*Alternatively, is it higher autotrophic respiration in SM3 (through increased fine root C*
*allocation) that allows the extra C to be blown off (line 501) Sorry, I'm not familiar enough*
*with all of these approaches to understand what each model is doing.*

[**Response**] Sorry for the confusion. Yes. For the SM1, our results showed that plant
nonphotosynthetic tissues (mainly wood) and litter quality impact the C:N ratio (Figure 5)
and further affect their decomposition cascade for fast and slow SOM pools (Figure 6 and

Figure 8). However, this was not the case for the passive SOM pool, where microbe immobilization dominates a low C:N ratio and then affects the decomposition cascade (please see response above).

For the SM3, both the hypothesis of increasing respiration to remove the excess C accumulated under N stress and the higher C investment for the BNF led to decrease in C input and then limit the microbe immobilization for the passive SOM pool.

*Line 488 what's being absorbed?*

[**Response**] Sorry for the confusion. We removed the "absorption" and revised the sentence as "This mechanism promotes the respiration of the faster turnover pools (fast and slow SOM pools), leading to decrease in MRT in these two pools (Figure 8)"

*Line 490: I'm still confused about what's causing differences between SM1 and SM3. For readers less familiar with these schemes can the difference between the approaches be unpacked a bit more, as this seems like a powerful strength of the traceability analysis?*

[**Response**] Sorry for the confusion. Based on the different hypotheses (list in Table 1) between SM1 and SM3, we found that SM1 mainly adjusted plant tissue and soil C:N ratio to reach equilibrium under N stress, while SM3 mainly cost the excess C via increasing respiration to get equilibrium under N stress. The two different strategies lead to different C allocation (Figure 9) and stoichiometric status (Figure 5), and then affect plant production (Figures 4 and 5), baseline residence time and ecosystem residence time (Figure 8) as well as ecosystem C storage (Figure 7). We added these results in the revised manuscript according to your suggestions.

**Anonymous Referee #2**

*[General comments] In this paper, the authors evaluate three different schemes of Carbon-Nitrogen coupling in a terrestrial model, which can largely change both C and N dynamics reproduced by models. For this, they used an existent framework for analyzing the difference between the models. This paper is clearly written, and the results are informative for readers. I recognize the importance of this study because CN coupling is one of the emergent processes to be evaluated / constrained in such land ecosystem modeling.*

[**Response**] Thank so much for your positive comment. No responses needed.

*However, I think there are places to be improved: the figures are informative, but the explanation is not enough for readers. My comments will not require a lot of effort to improve.*

[**Response**] Thanks so much for your comments and suggestions. We carefully revised the whole manuscript according to your comments and suggestions. We went through the text several times and made necessary changes. Please see below for the detailed responses.

*[Detailed comments] P6, L129: "biomass production date" should be "biomass production rate"?*

[**Response**] Thanks so much for pointing out our mistake. Here it is not "rate", either. It should be "data". We replaced "date" by "data" as suggested by referee #1.

*P6, L129: What purpose the data "standing biomass and biomass production date" used for your study? Do you mean the datasets are used to determine the parameters associated with the processes? In addition, CN concentration for plant and soil (Finzi et al., and Lichter et al.) are also used for your analysis (I suppose the SM2 simulation need such data because of the fixed CN ratio, but it is not clear in the text). Please clarify them.*

[**Response**] Sorry for the confusion. In this study, the data of both biomass and CN concentration are used to set initial values of C, N pool sizes and CN ratio for TECO-C and TECO-CN model. To make it clear, we added "To set the initial condition for the models, we collected the related datasets from previous studies." in the Lines 130-131.

*P6, L138: It might be better to clearly mention first that the model is newly developed and used in this study for the first time.*

[**Response**] Sorry for the confusion. There are two versions of TECO-CN model. The first version is used in Zaehel et al., 2014 and this study, and the second version is a simplified version used for data assimilation (e.g., Shi et al., 2015 and Du et al., 2017). Both versions are the variant of the TECO-C version published in Weng and Luo (2008). To make it clear in this study, we replaced "TECO-CN" with "TECO-CN 2.0" accordingly.

*P7, eq(1)-(6): The detail description of C allocation scheme of TECO-CN v2 is shown here, but it seems the equations are not referred in other places. In my simple thinking, the detail descriptions with the equations are not necessary for your analysis, and it looks no problem if your put them into supplement. If you want to keep the eqs in the main body, it should be qualitative explanations how the C allocation scheme act on CN dynamics in simulations.*

[**Response**] Thanks for your suggestions. Under the traceability analysis framework, the C allocation coefficients are used to calculate the baseline C residence time (Eq. 29). In this study, since both the matrix *A* and *C* are the same among different treatments (i.e., C-only, SM1, SM2 and SM3), the allocation coefficients (vector *B*) act as the key factor to determine the baseline C residence time. To clarify it, we added "The allocation coefficients act as the key factor to determine the baseline C residence time in this study" in the Lines 164-165.

*P8, L177- : Although it is shown in the Table 1, it will be helpful for readers to mention here again the fact that CN ratio in SM2 scheme is fixed, while other two are flexible.*

[**Response**] Thanks for your comments. We added "(i.e., fixed C:N ratio in SM2, flexible
C:N ratio in SM1 and SM3)" in the Lines 188-189.

*P12, L309: Which level of $CO_2$ concentration do you give to the model in the spin-up? Are*
*the $CO_2$ concentration and climate forcing in simulations given as a cyclic manner? Please*
*clarify them*

[**Response**] We used the $CO_2$ concentration of 1996-2007 from 361.3 to 382.0 ppmv. Yes,
we recycled the $CO_2$ concentration and climate forcing in simulations to the steady state
(more than 1000 cycles for each simulation). To clarify it, we added "In this study, the
meteorological forcings of 1996-2007 with the time step of half an hour were used to run the
models to the steady state" in the Lines 328-329.

.

*- P13, L319: "SˆCRT" should be "SˆMRT"?*

[**Response**] Thanks for pointing out our mistake. We corrected to "$S_i^{MRT}$" in this revised
version.

*- P15, L390: It looks less references to your figures and tables in the discussion section: It*
*was a bit difficult for me to figure out which claims in the discussion section are supported by*
*your own results.*

[**Response**] Thanks for pointing out this issue. We added more references in the Discussion
section. In addition, we added more figures (Figs 5b and 6) to show our results to support the
Discussion section. Please also see the responses to the first comment above.

*- P15, L405: You mention here that SM1 has a feedback from leaf N concentration to*
*photosynthetic capacity, but eq.(7) seems not. I have overlooked something, but if the SM1*
*actually has leaf-N concentration feedback, you should touch it in the section 2.1.1.*

[**Response**] Thanks for your comments. The plant N demand in the Eq.7 is calculated as:

$$N_{demand} = \frac{C_{leaf}}{CN_{leaf}} + \frac{C_{wood}}{CN_{wood}} + \frac{C_{root}}{CN_{root}}$$

$C_{leaf}$, $C_{wood}$ and $C_{root}$ are the current time step C pool sizes of plant tissues, $CN_{leaf}$, $CN_{wood}$
and $CN_{root}$ are the last time step C:N ratio of leaf, wood and root, respectively. To make it
clear, we added this equation to Line 200.

*P16, L426 "C cost of fixing": Is the effect of C cost actually considered in your simulation of*
*SM3? Which equation in the section 2.2.2 represents the effect? In addition, if you consider*
*the C cost in the SM3 simulation, does the lowest NPP of SM3 attribute to the increase of*
*autotrophic respiration in SM3? It would be nice if you can discuss on this.*

[**Response**] Thanks for your comments and suggestions. Yes, we used the same C cost
coefficient for N fixation (BNF) in SM1 and SM3. The different values of C investment for N
fixation are due to the different strategies between SM1 and SM3, resulting in the different
autotrophic respiration and NPP (Figure 3). For SM3, the calculation of BNF used the
empirical relationship of BNF with evapotranspiration explicitly, while SM1 represents BNF
as an option combining with the plant N uptake as the N source in terms of C investment
(Table 1). In other word, plant actively selects the N source on the basis of investment. Our
results showed that the strategy in SM1 lead to higher plant NUE than that in SM3 (Figure 5).
We added those information in the Discussion section in Lines 472-474 and 494-496.

*- P16, L427: I will appreciate if you can add more explanation why BNF of SM1 lead to the*
*highest NUE. In my understanding, if BNF in SM1 works as the complement to nitrogen*
*uptake, the process works to increase the uptake, and then the NUE(=NPP/PNU) should be*
*decreased. I wonder the SM1 has a mechanism to have BNF that satisfy a minimum N*
*requirement by plants, but it was not clear.*

[**Response**] Sorry for the confusion. As our response above, SM1 represents BNF as an
option combining with the plant N uptake as the N source in terms of C investment. Our
results showed that this strategy lead to the highest NUE among three C-N schemes. In order
to eliminate confusion, we revised the sentence as "On the other hand, SM1 applied a
different strategy, which set BNF as an option when the plant N uptake is not enough in terms
of C investment, leading to the highest plant NUE but the lowest response of BNF to NPP".

*- P16, L428: Although same N loss process are shared between the schemes, I suppose the*
*original models (TECO-CN/CLM/OCN) actually differ in that point. Readers can get benefit*
*if you can discuss it briefly.*

[**Response**] Thanks for your comments and suggestions. We added "In the original CLM4.5
and O-CN (Oleson et al., 2013; Zaehle et al., 2010), soil mineral N pool is divided into two
pools (ammonium and nitrate). The leaching is only active on the nitrate pool, while the
ammonium pool is assumed to be unaffected by leaching. This hypothesis may reduce the
correlation between leaching and total soil mineral N." in the Lines 478-482.

*- P17, L443: You discuss here how CN ratio in SM1 scheme affects the N regulation on plant*
*production processes. As you discuss in the section 4.2, SM3 also has the mechanism of*
*flexible CN ratio. How did the flexibility of SM3 act on plant production processes?*

[**Response**] Thanks for pointing out what we have neglected. In this revised version, we
added "However, this was not the case for the SM3 since both hypotheses of increasing
respiration to remove the excess C under N stress and the higher C investment for the BNF
lead to the decrease in C input and then limits the microbial immobilization for the passive
SOM pool." in the Lines 498-501.

*- P17, L445 "leading to a high microbial N immobilization": I cannot understand why high*
*CN ratio in plant tissues bring models to have a high microbial N immobilization. Need*
*further detail.*

[**Response**] Most previous studies showed that litter quality (ie., C:N ratio) could affect the
rate of microbial N immobilization (i.e., Zaehle et al., 2014; Thomas et al., 2015). When the
fresh litter inputs soil part with higher C:N ratio than SOM, the microbial demand for mineral
N increases to maintain the stoichiometry balance itself, which enhances the N
immobilization potential. We revised the sentence as "N stress increased litter C:N ratio,
leading to a high microbial N immobilization to keep their stoichiometry balance and then a
lower net N mineralization…."

*-P18, L475 "Fig. 6c" is likely to be "Fig. 7c"?*

*- P18, L498: Maybe "Fig. 7" is likely to be "Fig. 8".*

[**Response**] Thanks for pointing out our mistakes. In this revised version, we added a new
figure (i.e., Figure 5) and changed those figure numbers accordingly.

*- P18, L484: What does "structural litter quality" mean?*

[**Response**] Sorry for the confusion. In the TECO-CN model, based on different
decomposability, the plant litter is divided into two parts: metabolic litter and structural litter.
Based on our results, we deleted the "structural" in this sentence.

*In the analysis, plant production and C/N status are evaluated in steady state. Although I*
*recognize the usefulness of the analysis using steady states, I believe many readers get*
*interested how your conclusions can be extended to non-steady state simulations, because N*
*limitation on C cycle can be intensified in the condition where CO2 concentration increasing.*
*I will be happy if I can see the discussion on this. In addition, displaying N status in the three*
*simulations will be helpful for readers to get the whole picture of the CN dynamics: mineral*
*N is displayed (in Fig.3), but others (plant, litter, and SOM) are not. Since your analysis is*
*based on steady-state, such information can be a support to understand the relationship*
*between N-fluxes and N-pools. My suggestion is to include it in supplement.*

[**Response**] Thanks so much for your comments and suggestions. We agree that analysis of N
limitation on C cycle on the non-steady state is really interesting and critical. However, it is
difficult to simulate ecosystem C processes on the non-steady state. In this study, the
traceability analysis method is only for the steady-state simulations. Our next step is to
develop a transient traceability analysis for the non-steady state. In this revised version, we
added some discussion to show this caveat for the non-steady state in the Lines 452-454 and
469-471.

In addition, we added a new figure (Figure 5, please see above) for the sizes of C pools and
C:N ratios according to your and the fist referee's comments. We also added a single figure
(please see below) for N pools in supplement. We hope you will find our revision satisfactory.

[Figure]

**Carbon-nitrogen coupling under three schemes of model representation: a  traceability analysis**

Zhenggang Du[1,2], Ensheng Weng[2], Jianyang Xia[1,2]*, Lifen Jiang[3], Yiqi Luo[3,4,5], Xuhui Zhou[1,2,65]*

[1]*Center for Global Change and Ecological Forecasting, Tiantong National Field Observation Station for Forest Ecosystem, School of Ecological and Environmental Sciences, East China Normal University, Shanghai 200062, China*

[2]*Tiantong National Field Observation Station for Forest Ecosystem, School of Ecological and Environmental Sciences, East China Normal University, Shanghai 200062, China*

[23]*Department of Ecology & Evolutionary Biology, Princeton University, Princeton, NJ, USA*

[4]* [3]Center for Ecosystem Science and Society, Northern Arizona University, AZ, USA*

[5]* [4]Department for Earth System Science, Tsinghua University, Beijing 100084, China*

[6]* [5]Shanghai Institute of Pollution Control and Ecological Security, 1515 North Zhongshan Rd, Shanghai 200437, China*

**\*For correspondence:**

Xuhui Zhou & Jianyang Xia

*School of Ecological and Environmental Sciences*

*East China Normal University*

*500 Dongchuan Road, Shanghai 200062, China*

**Email**: xhzhou@des.ecnu.edu.cn, jyxia@des.ecnu.edu.cn

**Tel/Fax:** +86 21 54341275

*School of Ecological and Environmental Sciences*

*East China Normal University*

*500 Dongchuan Road, Shanghai 200062, China*

**Email**: jyxia@des.ecnu.edu.cn

**Abstract** The interaction between terrestrial carbon (C) and nitrogen (N) cycles has been incorporated into more and more land surface models. However, the scheme of C-N coupling differs greatly among models, and how these diverse representations of C-N interactions will affect C-cycle modeling remains unclear. In this study, we explored how the simulated ecosystem C storage capacity in the terrestrial ecosystem (TECO) model varied with three different commonly-used schemes of C-N coupling. The three schemes (SM1, SM2, and SM3) have been used in three different coupled C-N models (i.e., TECO-CN, CLM 4.5, and O-CN, respectively). They differ mainly in the stoichiometry of C and N in vegetation and soils, plant N uptake strategies, down-regulation of photosynthesis, and the pathways of N import. We incorporated the three C-N coupling schemes into the C-only version of TECO model, and evaluated their impacts on the C cycle with a traceability framework. Our results showed that all of the three C-N schemes caused significant reductions in steady-state C storage capacity compared with the C-only version with -23%, -30% and -54% for SM1, SM2, SM3, respectively. These reduced C storage capacity was mainly derived from the combined effects of decreases in net primary productivity (NPP, -29%, -15% and -45%) and changes in mean C residence time (MRT, 9%, -17% and -17%) for SM1, SM2, and SM3, respectively. The differences in NPP are mainly attributed to the different assumptions on plant N uptake, plant tissue C:N ratio, down-regulation of photosynthesis, and biological N fixation. In comparison, the alternative representations of the plant vs. microbe competition strategy and the plant N uptake, combining with the flexible C:N ratio in vegetation and soils, led to a notable spread MRT. These results highlight that the diverse assumptions on N processes represented among different C-N coupled models could cause additional uncertainty to land surface models. Understanding their difference can help us improve the capability of models to predict future biogeochemical cycles of terrestrial ecosystems.

**Keywords:** carbon-nitrogen coupling, traceability analysis, carbon storage capacity, nitrogen limitation, carbon residence time
* * *
Commented [ZD1]: Based on the results of Fig 10c

**1. Introduction**

The terrestrial ecosystem carbon (C) storage is jointly determined by ecosystem C input (i.e., net primary productivity, NPP) and mean residence time (MRT), which are   by the terrestrial nitrogen (N) availability  (Vitousek et al., 1991; Hungate et al., 2003;  Luo et al., 2017). Nitrogen is an essential component of enzymes, proteins, and secondary metabolites (van Oijen and Levy, 2004). Plant and microbial production require N to meet their stoichiometric demands, affecting the C balance and nutrient turnover of ecosystems  Since N limitation is widespread for plant growth in terrestrial ecosystems (LeBauer et al., 2008), ~~On one hand, increasing ecosystem C assimilation with atmospheric $CO_2$ increases the C:N ratios both in plant and soil, thus reduces the amount of additional N required (Rastetter et al., 1992). On the other hand, increasing soil C:N ratio leads to decomposing microorganisms costing more nitrogen, further affecting nitrogen mineralization and reducing efficiency of C assimilation (Gill et al., 2002). Although there is abundant N in the atmosphere, it is difficultavailable for biological systems (Houlton et al., 2008). As a consequence, the biological, which strongly affects C storage in ecosystems,metabolicratesphotosynthesisplantandandssC balance and turnover of~~ terrestrial ecosystem C storage (García-Palacios et al., 2013; Shi et al., 2015).

Given the importance of N availability on C sink projections (Hungate et al., 2003; Wang and Houlton 2009, Zaehle et al., 2015, Wieder et al., 2015b), N processes are increasingly incorporated into biogeochemical models. The representation of N cycling and their feedback to C cycling in models reflects what has been established in the ecosystem research community. Early C-N coupled models demonstrated that the N availability  limited C storage capacity with associated  effects on plant photosynthesis and growth  in many terrestrial ecosystems  (Melillo et al., 1993; Luo et al., 2004). Recent studies have largely confirmed these results by improving C-N coupling models with multiple hypotheses (Zhou et al., 2013; Zaehle et al., 2014; Thomas et al., 2015). These hypotheses include the plant down-regulation productivity based on N required for cell construction or N availability for plant absorption (Thornton et al., 2009; Gerber et al., 2010), constant or flexible stoichiometry for allocation and tissue (Wang et al., 2001; Shevliakova et al., 2009; Zaehle et al., 2010), competition between plants and microbes for soil nutrients (Zhu et al., 2017),

Evapotranspiration (ET)- or NPP-driven empirical functions to generate spatial estimates of biological N fixation (BNF) (Cleveland et al., 1999; Wieder et al., 2015a; Meyerholt et al.,

2016), and respiration of excess C to obtain N from environment and/or to prevent the accumulation of C beyond the storage capacity (Zaehle et al., 2010). These knowledge has significantly helped improve our understanding of the terrestrial C-N coupling and is an important basis to develop comprehensive terrestrial process-based models (Thornton et al.,

2007; Thomas et al., 2013). However, simulated results of the terrestrial C cycle illustrated considerable spread among models, and much of uncertainty arose from predictions of N

effects on C dynamic (Arora et al., 2013; Zaehle et al., 2015). The contradictory results were largely from different representations of fundamental N processes (e.g., the degree of flexibility of C:N ratio in vegetation and soils, plant N uptake strategies, pathways of N

import, decomposition, and the representations of the competition between plants and microbes for mineral N) (Sokolov et al., 2008; Wania et al., 2012; Walker et al., 2015).

Furthermore, the methodology used to derive the C-N coupling schemes among models varied largely, which may be invalid for the model intercomparisons to provide insight into the underlying mechanism of N status for terrestrial C cycle projection.

In the past decades, terrestrial models integrated more and more processes to improve model performance (Koven et al., 2013; Todd-Brown et al., 2013; Wieder et al., 2014).

The more processes incorporated, the more difficult it becomes to understand or evaluate model behavior (Luo et al., 2015). The traceability analysis has been developed to diagnose the simulation results within (Xia et al. 2013; Ahlström et al., 2015) and among (Rafique et al., 2016; Zhou et al., 2018) models

, facilitated benchmark analyses (Luo et al.,

2012), model intercomparisons (Zhou et al., 2018), and data-model fusion (Hararuk et al.,

2014),  (Huang et al., 2018). Based on the traceability analysis framework, key traceable elements, including fundamental properties of the terrestrial C cycle and their representations in shared structures among existing models, can be identified and characterized under different sources of variation (e.g., external forcing and uncertainty in processes) compared to the achieved predictive ability. The traceability analysis framework enables diagnosis of where models are clearly lacking predictive ability and evaluation of the relative benefit when more or alternative components are added to the models (Luo et al., 2015).

The presentis study is designed to examine the effects of C-N coupling under different schemes of model representation on ecosystem C storage in the Terrestrial Ecosystem (TECO)

model with the traceability analysis framework. Three schemes of model representation were conducted mainly based on TECO-CN 2.0 (SM1), CLM 4.5 (SM2), and O-CN (SM3, Table

1). The three C-N schemes differ in degrees of flexibility of C:N ratio in vegetation and soils, plant N uptake strategies, pathways of N import, and the representations of the competition between plants and microbes for soil available N. Based on the forcing data of ambient $CO_2$

concentration, N deposition, and meteorological data (i.e., air temperature, soil temperature, relative humidity, vapour pressure deficit, precipitation, wind speed, photosynthetically active radiation) obtained from Duke Forest during the period of 1996-2007, we conduct three alternative C-N coupling schemes (i.e., SM1, SM2 and SM3) as well as C-only in

TECO model framework to compare their effects on the ecosystem C storage capacity using traceability analysis framework. The N-processes sensitivity analysis was carried out to evaluate the variability in estimated ecosystem C storage caused by the process-related parameters at the steady state.

**2. Materials and methods**

**2.1 Data sources**

The forcing datasets used in this study were taken from the AmeriFlux site at Duke free-air

$CO_2$ enrichment (FACE)Forest experiment, located in the Blackwood Division, North

[revised manuscript text omitted]

$$f_{dreg} = \min(\frac{N_{sup}}{N_{demand}}, 1) \tag{7}$$

where  $N_{sup}$ (g N m$^{-2}$ s$^{-1}$) is actual supply of N obtained from re-translocated N, plant N uptake, and biological N fixation. $N_{demand}$ (g N m$^{-2}$ s$^{-1}$) is plant N

demand, which is calculated as:

$$N_{demand} = \sum_{i=leaf,wood,root} \frac{C_i}{CN_i^0} \tag{8}$$

where $C_i$ is the  C pool size of plant tissue at the current time step, and $CN_i^0$

is the  C:N ratio of plant tissue at the last time step.

The re-translocated N is calculated as:

$$N_{retrans} = \sum_{i=leaf,wood,root} r_i \times outC_i / CN_i \tag{89}$$

where $r_i$ is the N resorption coefficient, $CN_i$ is the C:N ratio and $outC_i$ (g C m$^{-2}$ s$^{-1}$) is the value of C leaving plant pool $i$ at each time step.

The plant N uptake (g N m$^{-2}$ s$^{-1}$) from soil mineral N pool is a function of root biomass density (Root$_{total}$, g C m$^{-2}$) and N demand of plants, following McMurtrie *et al.* (2012)

$$N_{uptake} = min(\max(0, N_{demand} - N_{retrans}), f_{U,max} \times SN_{mine} \times \frac{Root_{total}}{Root_{total}+Root_0}) \tag{910}$$

where $N_{demand}$ is the N demand of plants; $SN_{mine}$ (g N m$^{-2}$) is the soil mineral N ;

$f_{U,max}$ is the maximum rate of N absorption per step when $Root_{total}$ approaches infinity; and

$Root_0$ (g C m$^{-2}$) is a constant of root biomass  at which the N-uptake rate is half of the parameter $f_{U,max}$.

The biological N fixation (g N m$^{-2}$ s$^{-1}$) is calculated as:

$$N_{BNF} = \min(\max(0, N_{demand} - N_{retrans} - N_{uptake}), n_{fix} \times f_{nsc} \times NSC) \tag{1011}$$

where $n_{fix} = 0.0167$ is the maximum N fixation ratio and $f_{nsc}$ is the nutrient limiting factor. $f_{nsc}$ is calculated as

$$f_{nsc} = \begin{cases} 0, & NSC < NSC_{min} \\ \frac{NSC-NSC_{min}}{NSC_{max}-NSC_{min}}, & NSC_{min} < NSC < NSC_{max} \\ 1, & NSC > NSC_{max} \end{cases} \tag{1112}$$

where $NSC_{min}$ (g C m$^{-2}$) and $NSC_{max}$ (g C m$^{-2}$) are the minimal and maximal sizes of nonstructural C pool, respectively.

The soil microbial immobilization (g N m$^{-2}$ s$^{-1}$) is calculated as:

$$Imm_N = \begin{cases} \sum_{i=4}^{8} min\left(\left(\frac{C_i}{CN0_i} - \frac{C_i}{CN_i}\right), 0.1 * SN_{min}\right) & for\ CN_i \geq CN0_i \\ \sum_{i=4}^{8} min\left(\left(\frac{C_i}{CN_i} - \frac{C_i}{CN0_i}\right), 0.1 * SN_{min}\right) & for\ CN_i < CN0_i \end{cases}$$ (13)

Two pathways of N loss are modeled. One is gaseous loss ($N_{gas\_loss}$, g N m$^{-2}$ s$^{-1}$) and another is leaching ($N_{leach}$, g N m$^{-2}$ s$^{-1}$). 
[revised manuscript text omitted]
. (2012). In this study, the meteorological forcings of 1996-2007 with the time step of half an hour were used to run the models to the steady state. Once the simulations are spun up to the steady state, C and N fluxes and state variables as well as the matrix elements $A$, $C$, $B$, and $\xi$ in Eqn.(29) from all time steps in the last recycle of the climate forcing were saved for the traceability analysis.

The sensitivities of both NPP and mean C residence time (MRT) as well as ecosystem

C storage capacity to each main N process in three schemes were calculated as:

$$S_i^{NPP}(P) = \frac{NPP_i^+(P) - NPP_i^-(P)}{NPP_i^0} \tag{35}$$

$$S_i^{MRT}(P) = \frac{MRT_i^+(P) - MRT_i^-(P)}{MRT_i^0} \tag{36}$$

$$S_i^{ECSC}(P) = S_i^{NPP}(P) \times S_i^{MRT}(P) \qquad\qquad (37)$$

where $S_i^{NPP}(P)$, and $S_i^{MRT}(P)$, and $S_i^{ECSC}(P)$ ($i = 1, 2, 3$) represent the sensitivities of the NPP, and MRT and ecosystem C storage capacity to the N-process $P$ in the scheme $i$, respectively. $NPP_i^+(P)$ and $NPP_i^-(P)$ are the annual mean values of NPP that were simulated in scheme $i$ based on the value of the N-process $P$ (i.e., list in Table 1) (ie., DRP, PS, PUN, PMC, BNF, RtrN and SS) by increasing 50% and decreasing 50%, respectively. $MRT_i^+(P)$ and $MRT_i^-(P)$ are the annual mean values of MRTs that were simulated at in the same way as NPP and calculated using Eqn.(2930) and Eqn.(3031). $NPP_i^0$ and $MRT_i^0$ are the annual mean values of NPP and MRT at the steady state in the scheme $i$.

**3. Results**

**3.1 Simulations of C and N dynamics at steady state**

At the steady state, the dynamics of N fluxes and soil mineral N showed different patterns among three C-N schemes in the TECO model (FigureFig. 3). The simulated soil N mineralization and plant N uptake fluxes in SM2 displayed the largest daily variations (0.0015 and 0.00086 g N m$^{-2}$d$^{-1}$, respectively) and annual mean values (1.26 and 0.23 g N m$^{-2}$yr$^{-1}$, respectively) among three C-N schemes. For the N leaching flux, SM1 showed the largest daily variation (0.04 g N m$^{-2}$d$^{-1}$) and annual mean value (0.36 g N m$^{-2}$yr$^{-1}$). However, the biological N fixation (BNF) flux in SM1 showed the largest daily variation (0.028 g N m$^{-2}$d$^{-1}$) but but with the smallest annual value (0.04 g N m$^{-2}$yr$^{-1}$) among three C-N schemes. For tThe N immobilization fluxes in ,SM3 simulated displayed the largest daily variation (0.0013 g N m$^{-2}$d$^{-1}$) and SM1 showed the largest annual mean value (1.15 g N m$^{-2}$yr$^{-1}$). The dynamics of soil mineral N in SM2 and SM3 displayed the similar patterns on the daily and annual dynamics.

Compared with the TECO-C model, the three C-N coupling schemes introduced significant signs of N limitation on forest growth at the steady state but with varying strength magnitude (FigureFig. 4). Specifically, the three N schemes resulted incaused significant reductions in GPP (10%, 10% and 12% for SM1, SM2 and SM3, respectively) compared to the C-only TECO model. Similar response patterns were also found on NPP, ecosystem respiration, and heterotrophic respiration. Among the three schemes, SM3 had the strongest effect (45%, 12% and 45% reduction for NPP, ecosystem respiration, and heterotrophic respiration, respectively), while SM2 had the weakest effect (15%, 8% and 13%, respectively)

and  the effect of SM1 was relatively moderate   (29%, 10% and 29%, respectively). However, by comparison with the TECO-C version, both the SM1 and SM2 schemes increased the autotrophic respiration  by 12% and 27%, respectively, and SM2 scheme increased the NEE by 32%. Due to the NSC pool of TECO model, NEE were positive in all the experiments at the steady state (Weng and Luo, 2008).

Three C-N coupling schemes  induced  different effects on C and N stoichiometric status for different pools (Figs. 5 and  S2). All three schemes had significant limitation signs on woody, structural litter, fast and slow SOM pools but with different  magnitudes ( Fig. 5a). SM2 had the highest C sizes for the roots (731.8 g C m$^{-2}$) and metabolic litter (1252.1 g C m$^{-2}$)  while. SM1 had the highest C size for passive SOM pool (4249.5 g C m$^{-2}$).  SM2 had the constant C:N ratios for all the displaying pools ( Fig. 5b).  while the C:N ratios for three displaying pools (leaf, root and structural litter) had no significant change in both SM1 and SM3.  As for both woody and metabolic litter pools, SM1 and SM3 had higher C:N ratios (357.2 and 357.9, respectively) compared with SM2 (354). SM1 had the lowest C:N ratio (4.6) for soil passive SOM pool among the three schemes.

 The divergent effects of three C-N schemes on plant N uptake ( Fig. 3), autotrophic respiration,  and  NPP ( Fig. 4) lead to different N use efficiency (NUE) and carbon use efficiency (CUE) ( Fig. 5̶6). SM1 had the highest NUE (159.1 g C g$^{-1}$ N), mainly resulting from its lowest plant N uptake. In contrast, SM3 had the lowest NUE (67.3 g C g$^{-1}$ N) as a result of its smallest NPP. Because of the hypothesis of N uptake for free, SM2 had the highest CUE (0.54) among three C-N schemes, which was close to that in the C-only version (0.57). However, SM3 had the lowest CUE (0.35) , due to both C cost for plant actively uptake N  and the assumption that increase respiration to remove the excess C.

**3.2 Simulation of C storage capacity**

The ecosystem C storage capacity also differed greatly among the three C-N coupling schemes  and the C-only version of TECO model (Fig. 6̶7). The C-only version had the largest C storage capacity (19.5 Kg C m$^{-2}$) among the four simulations.  due to its highest NPP (879.9 g C m$^{-2}$ yr$^{-1}$). The C storage capacity in SM1 (15.1 Kg C m$^{-2}$) was close to that in SM2 (13.7 Kg C m$^{-2}$). The SM3 had the lowest C storage capacity (8.9 Kg C m$^{-2}$) among the four simulations as a result of its smallest NPP (483.9 g C m$^{-2}$ yr$^{-1}$) and relative short MRT (18.6 years). By comparison with the C-only version, the three C-N schemes all induced different reductions on NPP (-29%, -15% and -45% for SM1, SM2, SM3, respectively) and further reduced their ecosystem C storage capacity. For the MRT, SM1  exhibited positive  effects  (+9%) relative to that in the C-only version, while  another two schemes induced negative ones (i.e., -16.9% in SM2 and -16.7% in SM3) .

**3.3. Ecosystem C residence time**

Ecosystem C residence time ($\tau_E$) is collectively determined by baseline residence time, N scalar, and environmental scalars as shown in Eqn. (31). Specifically, differences in $\tau_E$ among three C-N coupling schemes and C-only TECO model are determined by baseline residence time and the effects of N scalar on eight plant C pools (Fig. 8). For example, SM1 had the longest $\tau_E$ because the N scalar had very strong control on passive SOM. The baseline residence time  was further determined by the C allocation  (Fig. 9). Overall, compared with C-only version, the additional N processes enhanced the partitioning coefficient of NPP to roots (33%, 82% and 53% for SM1, SM2 and SM3, respectively) but decreased the partitioning coefficient to wood (-25%, -45% and -34%, respectively). Furthermore, the decreased partitioning coefficient to wood (b2) regulated the variations of the baseline residence time of wood, structural litter, slow and passive SOM. However, the increased partitioning coefficient to roots (b3) determined the variations of the baseline residence time of roots and metabolic litter.

**3.4. Sensitivity of N processes to NPP and MRT**

For either NPP or MRT, the N processes had different sensitivities among the three C-N schemes of TECO model (Fig. 10). For NPP, plant C:N ratio had the highest sensitivities in both SM1 (0.32) and SM2 (0.53). However, plant N uptake in SM3 had the highest sensitivity (0.87) for NPP. For MRT, competition between plants and microbes, down-regulation of photosynthesis and plant C:N had the highest sensitivities in SM1 (0.27), SM2 (0.19) and SM3 (0.56), respectively. As the NPP and MRT jointly determined the ecosystem C storage capacity, the plant tissue C:N ratio, down-regulation of photosynthesis, and plant N uptake had the highest sensitivities for the ecosystem C storage capacity in SM1 (0.06), SM2 (0.09) and SM3 (0.26), respectively.

**4. Discussions**

**4.1 Underlying N processes and plant production**

Gross or net primary production (i.e., GPP or NPP or GPP) is regulated by the amount of N availabilityie for plant growth through the N demand, which is set by the relative proportion of biomass growth in the different plant components and their C:N stoichiometry (Zaehle et al., 2014; Thomas et al., 2015). The limitation of equilibrium N on plant production reflects the effects from multiple processes of the C-N interaction, mainly including down-regulation of photosynthetic capacity (DRP) by N availability, the ecosystem's balance of N inputs and losses (i.e., net ecosystem N exchange, NNE), plant N uptake (PNU), soil N mineralization (SNM), and the C:N stoichiometry of vegetation and soils. However, due to a lack of consensus on the nature of the mechanisms, the representation of these processes varies greatly among diverse models (Zaehle et al., 2014).

There are two common alternative assumptions of for the down-regulation of photosynthesisDRP that have been implemented in models: (1) the change in photosynthetic capacity is directly associated with the magnitude of plant available N (e.g., SM2), and (2) N limitation is associated with foliage N, which feeds back to limit photosynthetic capacity (e.g., SM1 and SM3). Our results showed that both assumptions had significant limitations with similar effects on GPP (Figures. 3a and 3g). The probable reason is that the TECO model calculates photosynthesis by light availability vs.and carboxylation rate based on the Farquhar model (Farquhar et al., 1980). The effects of N stress under the TECO framework, either associated with plant available N or associated with foliage N concentration, are estimated according to limiting factors of photosynthetic biochemistry (the maximum rate of carboxylation, $V_{cmax}$, and the maximum rate of electron transport at saturating irradiance, $V_{jmax}$). Th Note that the two assumptions of N down-regulation of photosynthesis may have different time-dependent effects on GPP in nonsteady-state systems (Xu et al., 2012; Walker et al., 2017).

At or near the steady state, NNE net ecosystem N exchange is driven by the processes of N input via deposition and fixation and N loss via leaching and volatilization stoichiometry (Zaehle et al., 2014; Thomas et al., 2015). Previous studies have stated that analyzing the steady-state condition is useful to understand N effects because the balance between external N sources and N losses determine whether an ecosystem is N limited (Rastetter et al., 1997; Menge et al., 2009; Thomas et al., 2015). In this study, divergent NPP responses among the three schemes might partly result from their different representations of BNF (Figures. 3 and

10). For exampleSpecifically, SM2 and SM3 simulated BNF explicitly, which used modified empirical relationships ofto calculate BNF based on with NPP evapotranspiration (ET) and evapotranspiration (ET)NPP, respectively. These phenomenological relationships generally captured biogeographical observations of higher rates of BNF in humid environments with high solar radiation (Wieder et al., 2015a). However, the highest response of NPP in only ET-driven BNF (i.e., SM3) may illustrate that not only energetic but also C costs of 'fixing' atmospheric di-N ($N_2$) into a biologically usable form ($NH_3$) broadly affect NPP (Gutschick 1981, Rastetter et al., 2001). This was because SM3 considered C investments in BNF while SM2 did not. By contrast, for the nonsteady state, the NPP-driven BNF creates a positive feedback between BNF and NPP, possibly causing large impact on C dynamic and terrestrial C storage (Wieder et al., 2015a). On the other hand, SM1 applied a different strategy, which represents set BNF as an complement option when to thecombining with the plant N uptake is enough for growth as the sources in terms of C investment, leading to the highest plant NUE (FigureFig. 6a) but the a lowest lower response of BNF to NPP (FigureFig. 10a). Another driving factor of NNE the net ecosystem N exchange is the N loss, which depends on the rate of leaching and volatilization. In this study, Uusing the same formulation as proportional to the size of soil mineral N pool among the three schemes, the divergent different annual mean magnitude of N leaching was more correlated to soil mineral N. In the original CLM4.5 and O-CN (Oleson et al., 2013; Zaehle et al., 2010), the soil mineral N pool is divided into two pools (ammonium and nitrate). T the N leaching is only validacts only on the nitrate pool, while the ammonium pool is assumed to be unaffected by leaching. This hypothesis may reduce the correlation between leaching and total soil mineral N.

The processes of PNU plant N uptake and net N mineralization determine how N moves through the plant-soil system, thereby triggering N limitation on plant growth and C storage capacity (FigureFig. 10). However, to our knowledge, exploring those processes exactly in models is limited by inadequate representation of above- and below-ground interactions that control the patterns of N allocation and whole-plant stoichiometry (Zaehle et al., 2014; Thomas et al., 2015). Plant tissue, litter, and SOM are the primary sinks of N in terrestrial ecosystems, while N in these forms is not directly available for PNUplant uptake, leading to an increase in N demand due tofor plant growth. On the other hand, t These N must turn over to become available for plant uptakegrowth. Therefore, the time for N to stay in these unavailable pools controls the transactional delay between the incorporation of N into plant unavailable pool and becomes available for plant uptake. In this way, the residence time of N in SOM appears to be an important factor for governing plant growth (see next section). In the presentis study, SM1 had the highest NUE from due to the combined effects of PNU plant N uptake based on C investment strategy (as described above) and flexible tissue C:N ratio. Nitrogen stress increased tissue C:N ratio (FigureFig. 5b), leading to a high microbial N immobilization and then a lower net N mineralization (FigureFig. 3), which allowed plant cell construction with a lower N requirement. However, this was not the case for the SM3 since both the hypotheseis of increasing respiration to remove the excess C accumulated under N stress and the higher C investment for the BNF lead to the decrease in C- input and then limits the microbial immobilization for the passive SOM pool. .The inclusion of flexible C:N stoichiometry (i.e., PS&SS) appeared to be an important feature allowing models to capture responses of the ecosystem C storage capacity response to climate variability through adjusting the C:N ratio of nonphotosynthetic tissues or the whole-plant allocation among tissues (Figures. 9 and 1Figure 10) with different C:N ratios (Zaehle & Friend, 2010). However, it is unclear whether those regulatory mechanisms exist in reality. Further modelling approaches need more reliable framework to predict stoichiometric flexibility.

**4.2 Ecosystem N status and C residence time**

[revised manuscript text omitted]

**5. Conclusions**

The C-N coupling has been represented in ecosystem and land surface models with different schemes, generating great uncertainties in model predictions. The most  difference among terrestrial C-N coupling models occurs with the degree of flexibility of C:N ratio in vegetation and soils, plant N uptake strategies, down-regulation of photosynthesis,- and the representations of the pathways of N import. In this study, we evaluated alternative representations of C-N interactions and their impacts on C cycle using the TECO model framework. Our traceability analysis showed that different representations of C-N coupling processes lead to divergent effects on both plant production and C residence time, and thus the ecosystem C storage capacity. The plant production are mainly affected by the different assumptions on net ecosystem N exchange, plant N uptake, net N mineralization, and the C:N ratio of vegetation and soil. In comparison, the alternative representations of the plant and microbe competition strategy, combining with the flexible C:N ratio in vegetation and soils, led to a notable spread effects on C residence time. Identifying the representations of main C-N processes under different schemes can help us  improve the N-limitation assumptions employed in terrestrial ecosystem models and forecast future C sink  in response to climate change.

*Code availability*. The code for TECO-CN and the three C-N coupling schemes is available at https://github.com/zgdu/TECO-CN-2.0-new.

*Data availability*. The data for this paper are available upon request to the corresponding authors.

*Competing interests*. The authors declare that they have no conflict of interest.

**Acknowledgements**

This work was financially supported by the National Key R&D Program of China
(2017YFA06046), the National Natural Science Foundation of China (31770559, 31722009),
National 1000 Young Talents Program of China, and the Fundamental Research Funds for
Central Universities. Zhenggang Du also thanks the China Scholarship Council
(201606140130) for scholarship support.

**Figure legends**

**Figure 1.** Schematic diagram of the terrestrial ecosystem carbon (C) and nitrogen (N) coupling model (TECO-CN2.0). (A) Canopy module, (B) Plant growth module, (C) Soil water dynamics module, (D) Soil carbon-nitrogen coupling module. Rectangles represent the carbon and nitrogen pools. $R_a$, autotrophic respiration. $R_h$, heterotrophic respiration. Retr., re-translocation. NSC, nonstructural carbohydrate. MNP, mineral N in plant tissues. SOM, soil organic matter. * set N fixation as an option when the plant N uptake is enough for growth in terms of C investment.

**Figure 2.** Schematic diagram illustrating the major carbon (C) and nitrogen (N) flows and stores in a terrestrial ecosystem, enclosing with alternative assumptions of N processes represent in SM1, SM2 and SM3, respectively. Light-blue arrows indicate C-cycle processes and red arrows show N-cycle processes.[1,2,3] alternative assumptions of N processes represent in scheme 1, 2 and 3, respectively. Met./Str. Litter, metabolic and/or structural litters; SOM, soil organic matter. * set N fixation as an option when the plant N uptake is enough for growth in terms of C investment in SM1, but go directly to soil mineral N pool in SM2 and SM3.

**Figure 3.** Simulated nitrogen fluxes and soil mineral nitrogen from three carbon-nitrogen coupling schemes (SM1, SM2 and SM3) in TECO-CN model for 1996 to 2007 at Duke Forest.

**Figure 4.** Simulated annual (a-f) and mean (g-l) carbon fluxes from carbon-only version and carbon-nitrogen coupled with three schemes (SM1, SM2 and SM3) of TECO model for 1996 to 2007 at Duke Forest. GPP, gross primary productivity; NPP, net primary productivity; NEE, net ecosystem exchange of $CO_2$; R-eco, ecosystem respiration; R-heter, heterotrophic respiration; R-auto, autotrophic respiration.

**Figure 5.** The annual average sizes of carbon pools (panel a) at the steady-state among 1996-2007 for C-only version and the three C-N schemes (SM1, SM2 and SM3) and the C:N ratio (panel b) of each carbon pools for the three C-N schemes (SM1, SM2 and SM3) in TECO-CN model.

**Figure 56.** The nitrogen use efficiency (NUE, panel a) in three C-N schemes of TECO model (SM1, SM2 and SM3) and the carbon use efficiency (CUE, panel b) at the steady-state among C-only version and the three C-N schemes of TECO model (SM1, SM2 and SM3).

The nitrogen use efficiency (NUE) in three C-N schemes of TECO model (SM1, SM2 and SM3).

**Figure 67.** Simulation of annual ecosystem carbon storage capacity for 1996 to 2006 at Duke Forest by carbon in flux (NPP, x axis) and ecosystem residence time ($\tau_E$, y axis) in TECO model framework with three carbon-nitrogen coupling schemes (SM1, SM2 and SM3) and in TECO C-only model (C). Inset (a), ecosystem carbon residence time ($\tau_E$) in SM1, SM2, SM3 and C-only model; inset (b), mean ecosystem carbon storage simulated among SM1, SM2, SM3 and C-only model; inset (c), relative change of NPP and ecosystem residence time simulated among three schemes compared with in C-only model.

**Figure 7.** Determination of carbon-pool residence times based on traceability framework in TECO C-N model with three C-N coupling schemes (SM1, SM2 and SM3) and TECO C-only model (C). Panel (a), baseline residence time; panel (b), mean residence time, and panel (c), nitrogen scalar.

**Figure 8.** Coefficients of partitioning of NPP to nonstructural C (NSC), root, woody and leaf in C-only model (C) and C-N coupling model with three schemes (SM1, SM2 and SM3).

**Figure 9.** The sensitivity of nitrogen processes to NPP (panel a),  ecosystem residence time ($\tau_E$, panel b), and ecosystem C storage capacity (panel c) among three carbon-nitrogen coupling schemes (SM1, SM2 and SM3). DRP, down-regulation of photosynthesis; PS, plant tissue C:N ratio; PNU, plant N uptake; PMC: plant and microbe competition; BNF, biological N fixation; RtrN, re-tranlocation N; SS, soil pool C:N ratio.

**Table1.** Summary of the nitrogen-carbon coupling schemes used and the representation of key processes in the carbon-nitrogen cycle.

| | **SM1 (TECO-CN2.0)[a]** | **SM2 (CLM4.5)[b,c]** | **SM3 (O-CN)[d,e]** |
|---|---|---|---|
| **Photosynthesis dDown- regulation of photosynthesis by N availability (DRP)** | Based on the comparison between plant N demand and actual supply | Based on the available soil mineral N relative to the N demanded to allocate photosynthate to tissue | Based on foliage N concentration, which varies with N deficiency |
| **Plant tissue stoichiometry (PS)** | Flexible plant C:N ratio | Fixed plant C:N ratio | Flexible plant C:N ratio |
| **Plant N uptake (PNU)** | Based on fine root biomass, soil mineral N and N demand of plant.

Plants itself choose the strategy between uptake from soil mineral N and fix $N_2$ by comparing C investment | Based on N required to allocate NPP to tissue.

Plants uptake N for free | Combining active and passive uptake of mineral N based on fine root C, soil mineral N, plant transpiration flux, increases with increased plant N demand |
| **N competition between plants and microbes (PMC)** | Microbes have first access to soil mineral N | Based on demand by both microbial immobilization and plant N uptake | Microbes have first access to soil mineral N, the competitive strength of plants increases under nutrient stress |
| **Biological N fixation (BNF)** | Based on the nitrogen demand of plants and maximum N fixing ratio considering nutrient concentration | $f(NPP)$ | $f(ET)$ |
| **Deployment of re-translocated N (RtrN)** | Fixed fraction of litter | Based on available N in the tissue and the previous year's annual sum of plant N demand | Fixed fraction of dying leaf and root tissue |
| **Soil organic matter stoichiometry (SS)** | Flexible soil C:N ratio | Fixed soil C:N ratio | Flexible soil C:N ratio |
| **N leaching** | Function of soil mineral N pool and runoff | Function of soil mineral N pool and runoff | Function of soil mineral N and runoff |

[a]See this study; [b]ThorntonKoven et al. (20132007), [c]ThorntonOleson et al. (20132009); [d] Zaehle &Friend (2010), [e]Zaehle et al. (2011).

C, carbon; N, nitrogen; NPP, net primary productivity; ET, evapotranspiration.

**Figure 1. TECO-CN**

[Figure]

**Figure 1.** Schematic diagram of the terrestrial ecosystem carbon (C) and nitrogen (N)
coupling model (TECO-CN). (A) Canopy module, (B) Plant growth module, (C) Soil
water dynamics module, (D) Soil carbon-nitrogen coupling module. Rectangles represent the
carbon and nitrogen pools. $R_a$, autotrophic respiration. $R_h$, heterotrophic respiration. Retr., re-
translocation. NSC, nonstructural carbohydrate. MNP, mineral N in plant tissues. SOM, soil
organic matter. * set N fixation as an option when the plant N uptake is enough for growth in
terms of C investment.

**Figure 2**

[Figure]

**Figure 2.** Schematic diagram illustrating the major carbon (C) and nitrogen (N) flows and
stores in a terrestrial ecosystem, enclosing with alternative assumptions of N processes
represent in SM1, SM2 and SM3, respectively. Light-blue arrows indicate C-cycle processes
and red arrows show N-cycle processes.[1,2,3] alternative assumptions of N processes represent
in scheme 1, 2 and 3, respectively. Met./Str. Litter, metabolic and/or structural litters; SOM,
soil organic matter. * set N fixation as an option when the plant N uptake is enough for
growth in terms of C investment in SM1, but go directly to soil mineral N pool in SM2 and
SM3.

**Figure 3**

[Figure]

**Figure 3.** Simulated nitrogen fluxes and soil mineral nitrogen from three carbon-nitrogen
coupling schemes (SM1, SM2 and SM3) in TECO-CN model for 1996 to 2007 at Duke
Forest.

**Figure 4**

[Figure]

**Figure 4.** Simulated annual (a-f) and mean (g-l) carbon fluxes from carbon-only version and
carbon-nitrogen coupled with three schemes (SM1, SM2 and SM3) of TECO model for 1996
to 2007 at Duke Forest. GPP, gross primary productivity; NPP, net primary productivity;
NEE, net ecosystem exchange of $CO_2$; R-eco, ecosystem respiration; R-heter, heterotrophic
respiration; R-auto, autotrophic respiration.

**Figure 5**

[Figure]

**Figure 5.** The annual average sizes of carbon pools (panel a) at the steady state among 1996-
2007 for C-only version and the three C-N schemes (SM1, SM2 and SM3) and the C:N ratio
(panel b) of each carbon pools for the three C-N schemes (SM1, SM2 and SM3) in TECO-
CN model.

**Figure 6**

[Figure]

**Figure 6.** The nitrogen use efficiency (NUE, the ratio of NPP:PNU, panel a) in three C-N
schemes of TECO model (SM1, SM2 and SM3) and the carbon use efficiency (CUE, the
ratio of NPP:GPP, panel b) at the steady-state among C-only version and the three C-N
schemes of TECO model (SM1, SM2 and SM3).

**Figure 7**

[Figure]

**Figure 7.** Simulation of annual ecosystem carbon storage capacity for 1996 to 2006 at Duke Forest by carbon in flux (NPP, x axis) and ecosystem residence time ($\tau_E$, y axis) in TECO model framework with three carbon-nitrogen coupling schemes (SM1, SM2 and SM3) and in TECO C-only model (C). The hyperbolic curves represent constant values (shown across the curves) of ecosystem carbon storage capacity. Inset (a), ecosystem carbon residence time ($\tau_E$) in SM1, SM2, SM3 and C-only model; inset (b), mean ecosystem carbon storage simulated among SM1, SM2, SM3 and C-only model; inset (c), relative change of NPP and ecosystem residence time simulated among three schemes compared with in C-only model.

**Figure 78**

[Figure]

**Figure 78.** Determination of carbon-pool residence times based on traceability analysis framework in TECO C-N model with three C-N coupling schemes (SM1, SM2 and SM3) and TECO C-only model (C). Panel (a), baseline residence time; panel (b), mean residence time, and panel (c), nitrogen scalar.

**Figure 8 9**

[Figure]

**Figure 89.** Coefficients of partitioning of NPP to nonstructural C (NSC), root, woody and
leaf in C-only model (C) and C-N coupling model with three schemes (SM1, SM2 and SM3).

**Figure 9̶10**

[Figure]

**Figure 9̶10.** The sensitivity of nitrogen processes to NPP (panel a) a̶n̶d̶ ,̲ ecosystem residence
time (τ_E, panel b)̲,̲ ̲a̲n̲d̲ ̲e̲c̲o̲s̲y̲s̲t̲e̲m̲ ̲C̲ ̲s̲t̲o̲r̲a̲g̲e̲ ̲c̲a̲p̲a̲c̲i̲t̲y̲ ̲(̲p̲a̲n̲e̲l̲ ̲c̲)̲ a̶mong three carbon-nitrogen
coupling schemes (SM1, SM2 and SM3). DRP, down-regulation o̲f̲ photosynthesis; PS, plant
tissue C:N ratio; PNU, plant N uptake; PMC: plant and microbe competition; BNF, biological
N fixation; RtrN, re-tranlocation N; SS, soil pool C:N ratio.

---

## Referee Report (RR1)

[General comments]

I appreciate the authors to respond to my comments to the previous manuscript. The revised manuscript has been much improved, but I think there still remains several points that should be clearer, including technical corrections.

- Eq (13): The numbers for pools (i=4~8) are defined in L296 and thus not yet defined here.

- Eq (23): how did you get the parameter value for vmax? Nuptake in SM3 looks strongly dependent on the choice of this parameter, but not specified in the text.
  In addition, in the third factor of "1/(Nmin x KNmin)", should "Nmin" be replaced by "SNmin"?

- Eq (31): I apologize if I misunderstand, but it seems "Tau_E = Xi$^{-1}$ Tau'_E" should be "Tau_E = Xi Tau'_E". Please check again.

- Eq (34): $CN_i^0$ is defined here as the ratio at t=0, but defined in Eq (8) as that of last time step.

- L341: In the sensitivity test, you increased/decreased 50% of each N-process. How did you make such changes in each process? For example, it is easy to change BNF by 50%, but I cannot imagine how you made the changes in the processes of PMC, PS, SS, etc. Readers will need brief explanations on this issue.

- Fig. 3o and 3q: In my understanding, since your analysis is based on steady-state simulations, the N budgets should be closed: BNF+Ndeposition should be comparable with the magnitude of Nleaching + Ngassing. However, in SM3, BNF looks much larger than Nleaching (and looks much smaller in SM1). 1000 years spin-up was not enough for the simulations? or other reasons? Do I miss something?

- L368-369, "SM1 and SM2 schemes increased " .

- L386, "Because of the hypothesis of Nuptake for free, SM2 had the highest CUE among three C-N schemes": This is slightly ambiguous for me. Does "Nuptake for free" mean "no C-cost on N uptake"?

- L446: Maybe "Our results showed … (Figs. 3a and 3g)" is "Our results showed … (Figs. 4a and 4g)"

- L473, "plant N uptake is enough for growth": maybe you forget "not".

- L483-L505: I'm still suffering from understanding the logical linking between the first half of this part (general understandings(?), L483-494) and the latter (claims obtained from your analysis(?), L494-505). For example, in the former part, you

mention "the residence time of N in SOM appears to be an important factor", but such discussion on residence time does not appear in the latter part...
Or you may intend to discuss first the residence time effect on plant production and then the effect of stoichiometry. If so, you should discuss the effect of residence time by referring more to your own results in the first part. The first part sounds like general understanding / background.

-   L544- "This mechanism promotes respiration of the faster turnover pools": This sentence is not obvious for me. Why does the excess-C removal process in SM3 promote the respiration of the faster turnover pools?

-   L557- "which is associated with plant competitiveness in SM1 and the respiration of excess labile C in SM3": how did you get to this conclusion? Readers will need more explanation on this.

-   Just a suggestion: As you noted in L513, "The effects of ecosystem N status on C mean residence time, however, has been much less studied than N limitation on ~", the N impact on MRT has been unclear when understanding model's behavior. I suppose you can address more in your conclusion section that your analysis framework can quantify the degree of N regulation on C storage capacity, with breaking down it into BOTH primary production and MRT (as a steady state). I think this will give more significance to your work and could be a strong message for readers.

-

---

## Author Response (AR2)

Tomomichi Kato

University of Tsukuba, Tsukuba, Ibaraki, Japan

Handling Topical Editor, Geoscientific Model Development (GMD)

Re: GMD-2018-41

Dear Dr. Kato,

Thanks so much for sending us two referees' assessment again on our resubmitted manuscript "Carbon-nitrogen coupling under three schemes of model representation: a traceability analysis" (No. GMD-2018-41). We appreciate the positive comments and further suggested amendments from the referees, which are very helpful to improve the paper. We have carefully studied the reviews, and revised our manuscript accordingly. As a consequence, our manuscript has been further improved.

We confirm that all authors have met the authorship criteria.

We also declare that the submitted work is our own and that copyright has not been breached in seeking its publication.

Here are our detailed responses to the reviews. Please note that the comments from the reviewers are in *italics* followed by our responses in **regular** text.

We hope you will find our revision satisfactory for publication in *Geoscientific Model Development*.

Yours Sincerely,

Xuhui & Jianyang

Xuhui Zhou, Jianyang Xia

School of Ecological and Environmental Sciences, East China Normal University

Dongchuan Road, Shanghai 200062, China

Email: xhzhou@des.ecnu.edu.cn, jyxia@des.ecnu.edu.cn

**Response letter to comments (gmd-2018-41)**

**Topical Editor Decision: Publish subject to minor revisions (review by editor)**

Dr. Tomomichi Kato

*I like to inform you that your paper is again subject to minor revision.*
*If you resubmit your article, please note that you carefully respond to all the comments one*
*by one.*

[**Response**] Thanks so much for your and the referees' assessments. We carefully revised the manuscript according to the referees' comments and suggestions and made necessary changes. Please see below for the detailed responses point by point. We hope you will find our revision satisfactory.

**Will Wieder's comment (Referee #1)**

*I appreciate the revisions made to this manuscript. Two additional questions and several technical corrections arose after reading the text. I trust these can be addressed without too much trouble.*

[**Response**] Thank so much for your positive comment. We carefully revised the whole manuscript according to your comments and suggestions. Please see responses below.

*Why aren't denitrification, or gaseous losses shown in Fig 3 or discussed with leaching losses (e.g. line 475)? Thomas et al. 2013 found big difference in the denitrification rates simulated by CLM4 and OC-N, so I'd assume at steady state the models make very different projections?*

[**Response**] Thanks for pointing out what we have neglected. Yes, the nitrogen gaseous losses showed large variation among three carbon-nitrogen schemes in our study. SM1, SM3 and SM2 had the smallest, biggest and moderate annual mean value (0.46, 0.77 and 1.39 g N $m^{-2}$ $yr^{-1}$), respectively. The differences of nitrogen gaseous losses among three C-N coupling schemes mainly due to both the nitrogen balance requirement and dynamics of soil mineral nitrogen (Eq. 15).

We have added three new panels (Figs. 3f, 3m and 3t) in the Figure 3, and revised the method, result and discussion sections in the revised version accordingly.

*I'm not sure I completely understand the explanation ~ lines 530 and 558. It's true, that CLM4.5 has fixed tissue stoichiometry, but it uses a dynamic C allocation scheme that should modify the C allocation to wood, stoichiometry, and MRT (at least in transient simulations)? This would also affect the allocation to wood vs. fine roots during spinup, with proportionally less wood C allocation associated with lower NPP? Are these nuances of C allocation from*

*SM2 & 3 brought into the TECO simulations presented? If not, maybe this is more of a*
*nuance of how the sensitivities were calculated, but perhaps worth clarifying?*

[**Response**] Sorry for the confusion. Yes, the nitrogen effects modify the C allocation (vector
*B*) and thus the baseline C residence time (baseline MRT, Eq. 30, Fig 8a) for the SM2
(CLM4.5bgc), which further affect the mean ecosystem residence time (MRT). Based on our
traceability analysis framework, we divided the nitrogen effects on MRT into two parts, one
is from C allocation (vector *B* in Eq. 29 and Eq. 30) and the other is the N scalar on the *C*
matrix (i.e., $\xi_N$ in Eq. 31, we named it as in this study). We have discussed the N scalar (i.e.,
$\xi_N$) on Lines 551-569 in this study.

*Technical corrections*
*L 85: should be 'dynamics'*

[**Response**] Done as suggested.
*L 103: I'm not really sure what "achieved predictive ability" means? Maybe delete this*
*entire clause after the end of the parenthesis in line 102*
[**Response**] Sorry for the confusion. As suggested, we deleted the "compared to the achieved
predictive ability" in revised version.

*L 109: are references needed for these models, as well as a description of their abbreviated*
*names?*

[**Response**] We have added the references and description of abbreviated names for each
mode. The sentence was revised as "Three schemes of model representation were conducted
mainly based on carbon-nitrogen coupling version of TECO (TECO-CN, Weng and Luo,
2008, [SM1]), Community Land Model Version 4.5 (CLM 4.5, Koven et al., 2013; Oleson et
al., 2013, [SM2]) and carbon-nitrogen coupling version of the Organizing Carbon and
Hydrology in Dynamic Ecosystems model (O-CN, Zaehle and Friend, 2010; Zaehle et al.,
2011, [SM3]) (Table 1)."
*L 153: it seems to introduce talk about "plant, litter and soil N pools" immediately following*
*mention of the inorganic N pool. Consider revising "There are nine organic N pools -*
*including plant, litter and soil N pools - and one inorganic soil N pool".*

[**Response**] We revised the sentence as "There are nine organic N pools, including plant,
litter and soil N pools, and one inorganic soil N pool"

*Eq 20 & Line 463, Maybe worth citing Cleveland et al 1999, which is the source of this*
*CLM4.5 approach for BNF (and I'm assuming for O-CN).*

[**Response**] We added the reference in Line 481.
*L 350, why not convert to mg N m^-2 d^-1 so the values are intelligible? Also, is the daily*
*variation important here, or the total flux calculated by each model? My guess is both, but*
*the latter is never really described (although it's displayed in the right column of Fig 3)?*

[**Response**] Thanks for your comments and suggestions. The variation of N processes is
mainly driven by the environment forcing, external N supply and ecosystem N demand. In
this study, we can demonstrate the different effects on daily variations from the three C-N
coupling representations.

We converted the units to mg N $m^{-2}$ $d^{-1}$ and added those description in this revised version.
*L 353 & 354, no daily variation for leaching or $N_{fix}$ is reported here?*

[**Response**] Thanks for pointing out what we have neglected. The daily variations for
leaching and biological nitrogen fixation were added in the revised version.
*L 370 & Fig 4, how does one generate a positive NEE for all years in all model*
*configurations if steady state conditions were achieved before starting the transient*
*simulation? By definition, it seems NEE should be zero over the years correspond the*
*equilibrium conditions? I don't understand how / or why a NSC pool would affect the*
*calculation of NEE, is the model just storing up NSC?*

[**Response**] Sorry for the confusion. The NSC pool is used to meet excess demand for
maintenance respiration during periods with low photosynthesis (e.g. at night, during winter
for perennial vegetation) in the TECO model. The initial value of NSC pool is set to
eliminate running a deficit of NPP (negative state), while this effect does not include in
calculating NEE in original version of TECO model. In this case, the total respiration is
greater than the GPP for each year, thus generate a positive NEE. To eliminate confusion, we
have recalculated the NEE, and revised the methods, results and discussion sections
accordingly.
*Fig 5, would it make more sense to plot both panels with a log y axis to show variation in*
*bools and stoichiometry?*

[**Response**] We have replotted Fig 5 with a log y axis as suggested.
*Line 414 and 416, what to b2 and b3 refer to? Is this eq 2 & 3, if so maybe refer to these*
*equations instead?*

[**Response**] Sorry for the confusion. The b2 and b3 represent the coefficients of partitioning
of NPP to wood and root. In order to eliminate confusion, we delete "(b2)" and "(b3)" in the
Line 432 and 434.
*L 452, isn't thus just Jmax in the Farquhar model (not Vjmax)?*

[**Response**] Thanks for your correction. We replaced "Vjmax" with "Jmax" in the revised
version.
*L 538, is this more specifically related to vegetation C:N ratio (not total)*

[**Response**] Done as suggested. We replaced the "total C:N ratio" to "vegetation C:N ratio"
in the revised version.

**(Anonymous Referee #2)**

*[General comments]*

*I appreciate the authors to respond to my comments to the previous manuscript. The revised*
*manuscript has been much improved, but I think there still remains several points that should*
*be clearer, including technical corrections.*

[**Response**] Thank so much for your positive comment. We carefully revised the whole
manuscript according to your comments and suggestions. Please see responses below.

*- Eq (13): The numbers for pools (i=4~8) are defined in L296 and thus not yet defined here.*

[**Response**] We added "where $CN0_i$ and $CN_i$ ($i = 4, 5, 6, 7, 8$) are the C:N ratios of metabolic
litter, structural litter, fast, slow and passive soil organic C pools at first- and current-time
step, respectively." in the revised version.

*- Eq (23): how did you get the parameter value for vmax? Nuptake in SM3 looks strongly*
*dependent on the choice of this parameter, but not specified in the text.*

[**Response**] For the O-CN model, the value of *vmax* is an empirical constant (Zaehle and
Friend, 2010; Kronzucker et al., 1995, 1996), which set as 0.514. We added those information
in this revised version.

*In addition, in the third factor of "1/(Nmin x KNmin)", should "Nmin" be replaced by*
*"SNmin"?*

[**Response**] Yes, Thanks for your correction. We replaced the "*Nmin*" with "*SNmin*" in
revised version.

*- Eq (31): I apologize if I misunderstand, but it seems "Tau_E = Xi-1 Tau'_E" should be*
*"Tau_E = Xi Tau'_E". Please check again.*

[**Response**] Based on the Eq.28: $\frac{dX(t)}{dt} = BU(t) - A\xi CX(t)$, which makes left part equal to
zero, the steady-state values of all carbon pools ($X_{ss}$) can be rearranged as:

$X_{ss} = (A\xi C)^{-1}BU_{ss} = \xi C^{-1}A^{-1}BU_{ss} = \xi^{-1}C^{-1}A^{-1}BU_{ss} = \xi^{-1}\tau'_E U_{ss} = \tau_E U_{ss}$

So, $\tau_E = \xi^{-1}\tau'_E$.

*- Eq (34): $CN_i^0$ is defined here as the ratio at t=0, but defined in Eq (8) as that of last time*
*step.*

[**Response**] Thanks for pointing our mistake. We corrected the "last time step" to "first time
step" in the revised version.

*- L341: In the sensitivity test, you increased/decreased 50% of each N-process. How did you*
*make such changes in each process? For example, it is easy to change BNF by 50%, but I*
*cannot imagine how you made the changes in the processes of PMC, PS, SS, etc. Readers will*
*need brief explanations on this issue.*

[**Response**] Thanks for your comments and suggestions. To do sensitivity test, we 1) run the
model to the steady state, 2) set the steady state as the initial state (C and N pool sizes) and
make change (increased/decreased 50%) in one nitrogen process to run the model to steady
state again, 3) calculate the relative changes of NPP, MRT and ecosystem C storage capacity
between the two steady states using Eq 35, 36 and 37. We added those description in Lines
350-353.

*- Fig. 3o and 3q: In my understanding, since your analysis is based on steady-state*
*simulations, the N budgets should be closed: BNF+Ndeposition should be comparable with*
*the magnitude of Nleaching + Ngassing. However, in SM3, BNF looks much larger than*
*Nleaching (and looks much smaller in SM1). 1000 years spin-up was not enough for the*
*simulations? or other reasons? Do I miss something?*

[**Response**] Thanks for pointing out what we have neglected. Yes, for the TECO model in
our study, $BNF+N_{deposition} \approx N_{leaching} +N_{gas\ losing}$. We added the results for nitrogen gaseous
losses in the revised version, including three new panels (Figs. 3f, 3m and 3t) in the Figure 3,
and revised the method, result and discussion sections accordingly.

*- L368-369, "SM1 and SM2 schemes increased ~~ 12% and 27%": maybe "SM1 and SM2"*
*is "SM1 and SM3 ~~".*

[**Response**] Thanks. We have corrected the "SM1 and SM2" to "SM1 and SM3" in the
revised version.

*- L386, "Because of the hypothesis of Nuptake for free, SM2 had the highest CUE among*
*three C-N schemes": This is slightly ambiguous for me. Does "Nuptake for free" mean "no*
*C-cost on N uptake"?*

[**Response**] Yes, in the CLM4.5, plant uptake nitrogen from soil do not require the
expenditure of energy in the form of carbon.

*- L446: Maybe "Our results showed ... (Figs. 3a and 3g)" is "Our results showed ...(Figs.*
*4a and 4g)"*

[**Response**] Corrected.

*- L473, "plant N uptake is enough for growth": maybe you forget "not".*

[**Response**] We added "not" in Lines 491

*- L483-L505: I'm still suffering from understanding the logical linking between the first half*
*of this part (general understandings(?), L483-494) and the latter (claims obtained from your*
*analysis(?), L494-505). For example, in the former part, you mention "the residence time of*

*N in SOM appears to be an important factor", but such discussion on residence time does not*
*appear in the latter part...*

*Or you may intend to discuss first the residence time effect on plant production and then the*
*effect of stoichiometry. If so, you should discuss the effect of residence time by referring more*
*to your own results in the first part. The first part sounds like general understanding /*
*background.*

[**Response**] Thanks for your comments and suggestions. Yes, we stated general
understandings on the nitrogen limitation from plant N uptake and net N mineralization based
on both steady- and nonsteady- state in the first half of this paragraph. For our analysis based
on the steady state, we mainly discussed the three C-N schemes referring our results at the
steady state in the latter of this paragraph. To make it clearer, we have revised this paragraph,
and added "This N limitation mainly occurs in nonsteady state, because accumulation of N in
the slow SOM pools reduces N available for plant uptake (Thomas et al., 2015). At or near
the steady state, however, the sequestration of N in SOM mainly affects the C residence time
(Fig. 8 and 10b). In this study, the different NUE among the three C-N schemes are induced
by different mechanisms." In Lines 512-516.

*- L544- "This mechanism promotes respiration of the faster turnover pools": This sentence is*
*not obvious for me. Why does the excess-C removal process in SM3 promote the respiration*
*of the faster turnover pools?*

[**Response**] Sorry for the confusion. For the SM3 (O-CN model), the excess C is respired to
prevent the accumulation of C beyond the storage capacity. At steady state, those SOM pools
with faster turnover rates have smaller storage capacity than those with slower turnover rates.
As a result, the excess C promotes the respiration of the faster turnover pools primarily. To
make it clear in this study, we added "to prevent the accumulation of C beyond the storage
capacity" in Line 567.

*- L557- "which is associated with plant competitiveness in SM1 and the respiration of excess*
*labile C in SM3": how did you get to this conclusion? Readers will need more explanation on*
*this.*

[**Response**] Thanks for your comments. Based on the traceability analysis (Eq.30 and Fig. 8),
the baseline C residence time (and thus the C residence time Eq. 31) is calculated by
allocation coefficients (*B* vector). Based on sensitivity analysis (Fig. 10b), the plant and
microbe competition (PMC) and the plant tissue C:N (PS) have the highest sensitivities to C
residence time for SM1 and SM3, respectively. For the SM3, the representation of respiration
of excess labile C mainly drives the change of plant C:N ratio (Fig. 4f and 4l). In the revised
version, we linked those results in Lines 580-581.

*- Just a suggestion: As you noted in L513, "The effects of ecosystem N status on C mean*
*residence time, however, has been much less studied than N limitation on ~", the N impact on*
*MRT has been unclear when understanding model's behavior. I suppose you can address*
*more in your conclusion section that your analysis framework can quantify the degree of N*
*regulation on C storage capacity, with breaking down it into BOTH primary production and*

*MRT (as a steady state). I think this will give more significance to your work and could be a*
*strong message for readers.*

[**Response**] Thanks so much for your suggestions. In this revised version, we added more
conclusion on the N regulation on ecosystem C storage capacity based on our results. We
hope you will find our revision satisfactory.

**Carbon-nitrogen coupling under three schemes of model representation: a traceability analysis**

Zhenggang Du[1], Ensheng Weng[2],  Lifen Jiang[3], Yiqi Luo[3,4], Jianyang Xia[1,5*], Xuhui Zhou[1,6*]

[1]*Center for Global Change and Ecological Forecasting, Tiantong National Field Observation Station for Forest Ecosystem, School of Ecological and Environmental Sciences, East China Normal University, Shanghai 200062, China*

[2]*Department of Ecology & Evolutionary Biology, Princeton University, Princeton, NJ, USA*

[3]*Center for Ecosystem Science and Society, Northern Arizona University, AZ, USA*

[4]*Department for Earth System Science, Tsinghua University, Beijing 100084, China*

[5]*Forest Ecosystem Research and Observation Station in Putuo Island, School of Ecological and Environmental Sciences, East China Normal University, Shanghai 200062, China*

[5] [6]*Shanghai Institute of Pollution Control and Ecological Security, 1515 North Zhongshan Rd, Shanghai 200437, China*

**\*For correspondence:**

Xuhui Zhou & Jianyang Xia

*School of Ecological and Environmental Sciences*

*East China Normal University*

*500 Dongchuan Road, Shanghai 200062, China*

**Email**: xhzhou@des.ecnu.edu.cn, jyxia@des.ecnu.edu.cn

**Tel/Fax:** +86 21 54341275

**Abstract** The interaction between terrestrial carbon (C) and nitrogen (N) cycles has been incorporated into more and more land surface models. However, the scheme of C-N coupling differs greatly among models, and how these diverse representations of C-N interactions will affect C-cycle modeling remains unclear. In this study, we explored how the simulated ecosystem C storage capacity in the terrestrial ecosystem (TECO) model varied with three different commonly-used schemes of C-N coupling. The three schemes (SM1, SM2, and SM3) have been used in three different coupled C-N models (i.e., TECO-CN, CLM 4.5, and O-CN, respectively). They differ mainly in the stoichiometry of C and N in vegetation and soils, plant N uptake strategies, down-regulation of photosynthesis, and the pathways of N import. We incorporated the three C-N coupling schemes into the C-only version of TECO model, and evaluated their impacts on the C cycle with a traceability framework. Our results showed that all of the three C-N schemes caused significant reductions in steady-state C storage capacity compared with the C-only version with the magnitudes of -23%, -30% and -54% for SM1, SM2, SM3, respectively. These reduced C storage capacity was mainly derived from the combined effects of decreases in net primary productivity (NPP, -29%, -15% and -45%) and changes in mean C residence time (MRT, 9%, -17% and -17%) for SM1, SM2, and SM3, respectively. The differences in NPP are mainly attributed to the different assumptions on plant N uptake, plant tissue C:N ratio, down-regulation of photosynthesis, and biological N fixation. In comparison, the alternative representations of the plant vs. microbe competition strategy and the plant N uptake, combining with the flexible C:N ratio in vegetation and soils, led to a notable spread MRT. These results highlight that the diverse assumptions on N processes represented  by different C-N coupled models could cause additional uncertainty to land surface models. Understanding their difference can help us improve the capability of models to predict future biogeochemical cycles of terrestrial ecosystems.

**Keywords:** carbon-nitrogen coupling, traceability analysis, carbon storage capacity, nitrogen limitation, carbon residence time

**1. Introduction**

Terrestrial ecosystem carbon (C) storage is jointly determined by ecosystem C input (i.e., net primary productivity, NPP) and mean residence time (MRT), both of which are strongly affected by the terrestrial nitrogen (N) availability (Vitousek et al., 1991; Hungate et al., 2003; Luo et al., 2017). Nitrogen is an essential component of enzymes, proteins, and secondary metabolites (van Oijen and Levy, 2004). Plant and microbial production require N to meet their stoichiometric demands, thus affecting the C balance and nutrient turnover of ecosystems (Cleveland et al., 2013; Wieder et al., 2015b). Since N limitation is widespread for plant growth in terrestrial ecosystems (LeBauer et al., 2008; Xia and Wan, 2008), N availability is often highly correlated with key ecological processes, such as C assimilation (Field and Mooney, 1986; Du et al., 2017), allocation (Kuzyakov et al., 2013), plant respiration (Sprugel et al., 1996), and litter and soil organic matter (SOM) decomposition (Terrer et al., 2016). Nitrogen dynamics thus plays an important role in governing the terrestrial ecosystem C storage (García-Palacios et al., 2013; Shi et al., 2015).

Given the importance of N availability on C sink projections (Hungate et al., 2003; Wang and Houlton 2009, Zaehle et al., 2015, Wieder et al., 2015b), N processes are increasingly incorporated into biogeochemical models. The representation of N cycling and their feedback to C cycling in models reflects what has been established in the ecosystem research community. Early C-N coupled models demonstrated that the N availability limited C storage capacity with associated effects on plant photosynthesis and growth in many terrestrial ecosystems (Melillo et al., 1993; Luo et al., 2004). Recent studies have largely confirmed these results by improving C-N coupling models with multiple hypotheses (Zhou et al., 2013; Zaehle et al., 2014; Thomas et al., 2015). These hypotheses include the plant down-regulation productivity based on N required for cell construction or N availability for plant absorption (Thornton et al., 2009; Gerber et al., 2010), constant or flexible stoichiometry for allocation and tissue (Wang et al., 2001; Shevliakova et al., 2009; Zaehle et al., 2010), competition between plants and microbes for soil nutrients (Zhu et al., 2017), Evapotranspiration (ET)- or NPP-driven empirical functions to generate spatial estimates of biological N fixation (BNF) (Cleveland et al., 1999; Wieder et al., 2015a; Meyerholt et al., 2016), and respiration of excess C to obtain N from environment and/or to prevent the accumulation of C beyond the storage capacity (Zaehle et al., 2010). The knowledge has significantly helped improve our understanding of the terrestrial C-N coupling and is an important basis to develop comprehensive terrestrial process-based models (Thornton et al., 2007; Thomas et al., 2013).

However, simulated results of the terrestrial C cycle illustrated considerable spread among models, and much of uncertainty arose from predictions of N effects on C dynamics (Arora et al., 2013; Zaehle et al., 2015). The contradictory results were largely from different representations of fundamental N processes (e.g., the degree of flexibility of C:N ratio in vegetation and soils, plant N uptake strategies, pathways of N import, decomposition, and the representations of the competition between plants and microbes for mineral N) (Sokolov et al., 2008; Wania et al., 2012; Walker et al., 2015). Furthermore, the methodology used to derive the C-N coupling schemes among models varied largely, which might be invalid for the model intercomparisons to provide insight into the underlying mechanism of N status for terrestrial C cycle projection.

In the past decades, terrestrial models integrated more and more processes to improve model performance (Koven et al., 2013; Todd-Brown et al., 2013; Wieder et al., 2014). The more processes incorporated, the more difficult it becomes to understand or evaluate model behavior (Luo et al., 2015). The traceability analysis has been developed to diagnose the simulation results within (Xia et al. 2013; Ahlström et al., 2015) and among (Rafique et al.,

2016; Zhou et al., 2018) models. Based on the traceability analysis framework, key traceable elements, including fundamental properties of the terrestrial C cycle and their representations in shared structures among existing models, can be identified and characterized under different sources of variation (e.g., external forcing and uncertainty in processes) compared to the achieved predictive ability. The traceability analysis enables diagnosis of where models are clearly lacking predictive ability and evaluation of the relative benefit when more or alternative components are added to the models (Luo et al., 2015).

This study is designed to examine the effects of C-N coupling under different schemes of model representation on ecosystem C storage in the Terrestrial Ecosystem (TECO) model with the traceability analysis framework. Three schemes of model representation were conducted mainly based on carbon-nitrogen coupling version of TECO (TECO-CN (SM1),

Weng and Luo, 2008, [SM1]), Community Land Model Version 4.5 (CLM 4.5 (SM2), Koven et al., 2013; Oleson et al., 2013, [SM2]) and carbon-nitrogen coupling version of the

Organizing Carbon and Hydrology in Dynamic Ecosystems model (O-CN (SM3, Zaehle and

Friend, 2010; Zaehle et al., 2011, [SM3]) (Table 1). The three C-N schemes differ in degrees of flexibility of C:N ratio in vegetation and soils, plant N uptake strategies, pathways of N

import, and the representations of the competition between plants and microbes for soil available N. Based on the forcing data of ambient $CO_2$ concentration, N deposition, and meteorological data (i.e., air temperature, soil temperature, relative humidity, vapour pressure deficit, precipitation, wind speed, photosynthetically active radiation) obtained from Duke

Forest during the period of 1996-2007, we conduct three alternative C-N coupling schemes (i.e., SM1, SM2 and SM3) as well as C-only in TECO model framework to compare their effects on the ecosystem C storage capacity. The N-processes sensitivity analysis was carried out to evaluate the variability in estimated ecosystem C storage caused by the process-related parameters at the steady state.

**2. Materials and methods**

**2.1 Data sources**

The datasets used in this study were taken from the Duke free-air $CO_2$ enrichment (FACE)

experiment, located in the Blackwood Division, North Carolina, USA (35.97º N, 79.08º W).

The flux tower lies on a 15-year-old loblolly pine (*Pinus taeda L.*) plantation. The meteorological forcing data were downloaded from the AmeriFlux database at http://ameriflux.lbl.gov, including ambient $CO_2$ concentration ([$CO_2$]), air temperature at the top canopy (*Ta*), soil temperature (*Ts*), photosynthetically active radiation (*PAR*), relative humidity (*RH*), vapor pressure deficit (*VPD*), precipitation, wind speed [*Ws*], and N

deposition. All forcing data sets are available from 1996 to 2007. To set the initial condition for the models, we collected the related datasets from the previous studies. Standing biomass and biomass production data at each plot for plant compartments (i.e., foliage, fine root and woody biomass, including branches and coarse roots) were taken from McCarthy et al.

(2010). The C and N concentration data for each plant compartment based on Finzi et al.

(2007) were used to estimate C and N stocks and fluxes. Plant N demand and uptake were calculated from these data measured by Finzi et al. (2007). The C and N concentrations of litter and SOM were obtained from Lichter et al. (2008).

**2.2 Model description and C-N schemes**

**2.2.1 TECO-CN**

The terrestrial ecosystem C-N coupling model (TECO-CN) used in the present study is a variant of the TECO-Carbon-only version (TECO-C) by incorporating additional key N

processes (Fig. 1). TECO-C model is a process-based ecosystem model designed to examine critical processes regulating interactive responses of plants and ecosystems to climate change.

It has four major components: canopy photosynthesis module, plant growth module, soil water dynamic module, and soil C dynamic module. The canopy photosynthesis and soil water dynamic modules run at hourly time step while the plant growth and soil C dynamic modules run at the daily time step. The detailed description of the TECO-C model can be found in Weng and Luo (2008).

The N cycle added to the TECO model for this study is simplified following the structure of Luo  and Reynolds (1999), Gerber et al. (2010), and Wang et al. (2010). It has a similar structure to the TECO-C model (Fig. 1). There are nine organic N pools

, including plant, litter and soil N pools, and one inorganic soil N pool. The plant

N pools include leaves, wood, roots, and mineral N in plant tissues. The litter and soil N

pools include metabolic and structural litter N, fast, slow, and passive soil organic N (SON), and soil mineral N pools. The total plant N demand on each time step is calculated following the NPP allocation to new tissue growth based on their C:N ratios. To meet the demand, the plant N supply is calculated from three parts, including the retranslocated N from senescing tissues, plant uptake from soil mineral N pool, and external N sources from atmospheric deposition and biological N fixation. The N absorbed by roots enters into the mineral N pool in plant tissues, and then is allocated to the remaining plant pools with plant growth. The N in leaves and fine roots is reabsorbed before senescence. Plant litters will enter metabolic or structural pools depending on their C:N ratios.

The allocation coefficients act as the key factor to determine the baseline C residence time in this study. Plant assimilated C allocating to the leaves, stems and roots depends on their growth rates, which vary with phenology (Luo et al., 1995; Denison and Loomis, 1989;

Shevliakova et al., 2009; Weng and Luo, 2008):

$$b_l = \frac{1}{1+c_1+c_2} \tag{1}$$

$$b_s = \frac{c_2}{1+c_1+c_2} \tag{2}$$

$$b_r = \frac{c_1}{1+c_1+c_2} \tag{3}$$

where $b_l$, $b_s$ and $b_r$ are the partitioning coefficient of newly assimilated C to leaves, stems and roots, respectively. Parameters $c_1$ and $c_2$ are calculated as:

$$c_1 = \frac{bm_l}{bm_r} * \frac{CN_l^i}{CN_l^0} \tag{4}$$

$$c_2 = 0.5 * 250e^3 * SLA * 0.00021 * h^2 \tag{5}$$

where $bm_l$ and $bm_r$ are the leaf and root biomass; $CN_l^i$ and $CN_l^0$ represent the C:N ratios of the leaf pool at 0 and current time step, respectively; $SLA$ is specific leaf area; $h$ is plant height, which is calculated as:

$$h = h_{max}(1 - \exp(-h_1 * bmP)) \qquad (6)$$

where $h_{max}$ is the maximum canopy height; $h_1$ is an empirical parameter and $bmP$ is plant biomass.

**2.2.2 C-N coupling schemes**

We conducted four experiments including three simulations with their representations of C-N

coupling schemes (SM1, SM2 and SM3) and an additional C-only simulation in TECO model framework. The three C-N interaction simulations include one original scheme in TECO-CN

model and the other two schemes representing CLM4.5-BGC and O-CN. The three C-N

coupling schemes differ in the representation of down-regulation of photosynthesis, the degree of flexibility of C:N ratio in vegetation and soils (i.e., fixed C:N ratio in SM2, flexible

C:N ratio in SM1 and SM3), plant N uptake strategies, pathways of N import to the plant reserves, and the competition between plants, and microbes for soil mineral N (Table1, Fig.

2).

**SM1 (TECO-CN)**

The N down-regulation of photosynthesis in SM1 is determined by the comparison between plant N demand and actual supply of N:

$$f_{dreg} = \min(\frac{N_{sup}}{N_{demand}}, 1) \qquad (7)$$

where $N_{sup}$ (g N m$^{-2}$ s$^{-1}$) is actual supply of N obtained from re-translocated N, plant N

uptake, and biological N fixation. $N_{demand}$ (g N m$^{-2}$ s$^{-1}$) is plant N demand, which is calculated as:

$$N_{demand} = \sum_{i=leaf,wood,root} \frac{C_i}{CN_i^0} \qquad (8)$$

where $C_i$ is the C pool size of plant tissue at the current time step, and $CN_i^0$ is the C:N ratio of plant tissue at the  first time step.

The re-translocated N is calculated as:

$$N_{retrans} = \sum_{i=leaf,wood,root} r_i \times outC_i/CN_i \qquad (9)$$

where $r_i$ is the N resorption coefficient, $CN_i$ is the C:N ratio and $outC_i$ (g C m$^{-2}$ s$^{-1}$) is the value of C leaving plant pool $i$ at each time step.

The plant N uptake (g N m$^{-2}$ s$^{-1}$) from soil mineral N pool is a function of root biomass density (Root$_{total}$, g C m$^{-2}$) and N demand of plants, following McMurtrie *et al*. (2012)

$$N_{uptake} = min(max(0, N_{demand} - N_{retrans}), f_{U,max} \times SN_{mine} \times \frac{Root_{total}}{Root_{total}+Root_0}) \quad (10)$$

where $N_{demand}$ is the N demand of plants; $SN_{mine}$ (g N m$^{-2}$) is the soil mineral N; $f_{U,max}$ is the maximum rate of N absorption per step when $Root_{total}$ approaches infinity; and $Root_0$ (g C m$^{-}$

$^{2}$) is a constant of root biomass at which the N-uptake rate is half of the parameter $f_{U,max}$.

 The biological N fixation (g N m$^{-2}$ s$^{-1}$) is calculated as:

$$N_{BNF} = min(max(0, N_{demand} - N_{retrans} - N_{uptake}), n_{fix} \times f_{nsc} \times NSC) \quad (11)$$

where $n_{fix} = 0.0167$ is the maximum N fixation ratio and $f_{nsc}$ is the nutrient limiting factor.

$f_{nsc}$ is calculated as

$$f_{nsc} = \begin{cases} 0, & NSC < NSC_{min} \\ \frac{NSC-NSC_{min}}{NSC_{max}-NSC_{min}}, & NSC_{min} < NSC < NSC_{max} \\ 1, & NSC > NSC_{max} \end{cases} \quad (12)$$

where $NSC_{min}$ (g C m$^{-2}$) and $NSC_{max}$ (g C m$^{-2}$) are the minimal and maximal sizes of nonstructural C pool, respectively.

 The soil microbial immobilization (g N m$^{-2}$ s$^{-1}$) is calculated as:

$$Imm_N = \begin{cases} \sum_{i=4}^{8} min\left(\left(\frac{C_i}{CN0_i} - \frac{C_i}{CN_i}\right), 0.1 * SN_{min}\right) & for\ CN_i \geq CN0_i \\ \sum_{i=4}^{8} min\left(\left(\frac{C_i}{CN_i} - \frac{C_i}{CN0_i}\right), 0.1 * SN_{min}\right) & for\ CN_i < CN0_i \end{cases} \quad (13)$$

where $CN0_i$ and $CN_i$ ($i = 4, 5, 6, 7, 8$) are the C:N ratios of metabolic litter, structural litter, fast, slow and passive soil organic C pools at first- and current-time step, respectively.

 Two pathways of N loss are modeled. One is gaseous loss ($N_{gas\_loss}$, g N m$^{-2}$ s$^{-1}$) and another is leaching ($N_{leach}$, g N m$^{-2}$ s$^{-1}$). Both are proportional to the availability of soil mineral N ( $SN_{min}$, g N m$^{-2}$). The equations are:

 (14)

$$N_{leach} = f_{nleach} \times \frac{V_{runoff}}{h_{depth}} \times SN_{min} \quad (\text{15}14)$$

$$N_{gas\_loss} = max( f_{ngas} \times e^{\frac{T_{soil}-25}{10}} \times SN_{min}, N_{BNF}+N_{depos}-N_{leaching})$$

where $f_{ngas} = 0.001$ and $f_{nleach} = 0.5$, $T_{soil}$(°C) is the soil temperature, $V_{runoff}$ (mm s$^{-1}$) is the value of runoff, and $h_{depth}$ ($mm$) is the soil depth, Ndepos = 0.78 gN m$^{-2}$ yr$^{-1}$, is N

deposition used in this study.

**SM2 (CLM4.5bgc)**

The N down-regulation of photosynthesis in SM2 is calculated as:

$$f_{dreg} = \frac{CF_{allo} - CF_{avail\_alloc}}{CF_{GPP_{pot}}} \tag{16}$$

where $CF_{allo}$ (g C m$^{-2}$ s$^{-1}$) is the total flux of allocated C, which is determined by available mineral N. $CF_{avail\_alloc}$ (g C m$^{-2}$ s$^{-1}$) is the potential C flux from photosynthesis, which can be allocated to new growth. $CF_{GPP_{pot}}$ (g C m$^{-2}$ s$^{-1}$) is the potential gross primary productivity (GPP) when there is no N limitation.

The re-translocated N (g N m$^{-2}$ s$^{-1}$) is calculated as:

[revised manuscript text omitted]
. (2012). In this study, the meteorological forcings of 1996-2007 with the time step of half an hour were used to run the models to the steady state. Once the simulations are spun up to the steady state, C and N fluxes and state variables as well as the matrix elements $A$, $C$, $B$, and $\xi$ in Eqn.(29) from all time steps in the last recycle of the climate forcing were saved for the traceability analysis.

The sensitivities of both NPP and mean C residence time (MRT) as well as ecosystem C

storage capacity to each main N process in three schemes were calculated as:

$$S_i^{NPP}(P) = \frac{NPP_i^+(P) - NPP_i^-(P)}{NPP_i^0} \tag{35}$$

$$S_i^{MRT}(P) = \frac{MRT_i^+(P) - MRT_i^-(P)}{MRT_i^0} \tag{36}$$

$$S_i^{ECSC}(P) = S_i^{NPP}(P) \times S_i^{MRT}(P) \tag{37}$$

where $S_i^{NPP}(P)$, $S_i^{MRT}(P)$, and $S_i^{ECSC}(P)$ $(i = 1, 2, 3)$ represent the sensitivities of NPP, MRT

and ecosystem C storage capacity to the N-process $P$ in the scheme $i$, respectively. $NPP_i^0$ and

$MRT_i^0$ are the annual mean values of NPP and MRT at the steady state in the scheme $i$.

$NPP_i^+(P)$ and $NPP_i^-(P)$ are the annual mean values of NPP that were simulated to steady state again in scheme $i$ based on the value of the N-process $P$ (i.e., list in Table 1) by increasing 50% and decreasing 50%, respectively. $MRT_i^+(P)$ and $MRT_i^-(P)$ are the annual mean values of MRTs that were simulated at the same way as NPP and calculated using

Eqn.(30) and Eqn.(31).

**3. Results**

**3.1 Simulations of C and N dynamics at steady state**

At the steady state, the dynamics of N fluxes and soil mineral N showed different patterns among three C-N schemes in the TECO model (Fig. 3). The simulated soil N mineralization and plant N uptake fluxes in SM2 displayed the largest daily variation (1.5 and 0.86 mg N m$^{-2}$d$^{-1}$, respectively) and annual mean values (1.26 and 0.23 g N m$^{-2}$yr$^{-1}$, respectively) among three C-N schemes. This variation mainly resulted from both the plant N demand and the available N in soil (Fig. 3g).  The dynamic of soil mineral N also drove the variation of the N leaching flux, which the SM1 showed the largest daily variation (0 mg N m$^{-2}$d$^{-1}$) and annual mean value (0.36 g N m$^{-2}$yr$^{-1}$). However, the representation of biological N fixation (BNF) as an option when the plant uptake is not enough for growth led to  the largest daily variation (28 mg N m$^{-2}$d$^{-1}$) but with the smallest annual value (0.04 g N m$^{-2}$yr$^{-1}$) in SM1 in comparison with other two C-N schemes. Both the nitrogen balance requirement and the dynamic of soil mineral N resulted to the largest daily variation (1.97 mg N m$^{-2}$d$^{-1}$) and annual value of gaseous N loss (1.39 g N m$^{-2}$yr$^{-1}$) in SM3. The combined effect of flexible C:N ratio and soil mineral N drove the largest daily variation of N immobilization fluxes (1.3 mg N m$^{-2}$d$^{-1}$) in SM3  and  the largest annual mean value (1.15 g N m$^{-2}$yr$^{-1}$) in SM1. The dynamics of soil mineral N in SM2 and SM3 displayed the similar patterns on the daily and annual dynamics.

Compared with the TECO-C model, the three C-N coupling schemes introduced significant signs of N limitation on forest growth at the steady state but with varying magnitude (Fig. 4). Specifically, the three N schemes caused significant reductions in GPP (10%, 10% and 12% for SM1, SM2 and SM3, respectively) compared to the C-only TECO model. Similar response patterns were also found on NPP, ecosystem respiration, and heterotrophic respiration. Among the three schemes, SM3 had the strongest effect (45%, 12% and 45% reduction for NPP, ecosystem respiration, and heterotrophic respiration, respectively), while SM2 had the weakest effect (15%, 8% and 13%, respectively) and the effect of SM1 was relatively moderate (29%, 10% and 29%, respectively). However, by comparison with the TECO-C version, both the SM1 and  SM3 schemes increased the autotrophic respiration by 12% and 27%, respectively. At or near the steady state, NEE in both TECO-C and three C-N coupling schemes had similarly mean values (1.37, -0.13, 0.66 and 0.84 g C m$^{-2}$ yr$^{-1}$) which were equal to zero approximately but with large variations (56, 39.4, 48.1 and 34.9).

Three C-N coupling schemes induced different effects on C and N stoichiometric status for different pools (Figs. 5 and S2). All three schemes had significant limitation signs on woody, structural litter, fast and slow SOM pools but with different magnitudes (Fig. 5a).

SM2 had the highest C sizes for the roots (731.8 g C m$^{-2}$) and metabolic litter (1252.1 g C m$^{-}$

$^{2}$), while SM1 had the highest C size for passive SOM pool (4249.5 g C m$^{-2}$). SM2 had the constant C:N ratios for all the displaying pools (Fig. 5b), while the C:N ratios for three displaying pools (leaf, root and structural litter) had no significant change in both SM1 and

SM3. As for both woody and metabolic litter pools, SM1 and SM3 had higher C:N ratios (357.2 and 357.9, respectively) compared with SM2 (354). SM1 had the lowest C:N ratio (4.6) for soil passive SOM pool among the three schemes.

The divergent effects of three C-N schemes on plant N uptake (Fig. 3), autotrophic respiration, and NPP (Fig. 4) lead to different N use efficiency (NUE) and carbon use efficiency (CUE) (Fig. 6). SM1 had the highest NUE (159.1 g C g$^{-1}$ N), mainly resulting from its lowest plant N uptake. In contrast, SM3 had the lowest NUE (67.3 g C g$^{-1}$ N) as a result of its smallest NPP. Because of the hypothesis of N uptake for free, SM2 had the highest CUE

(0.54) among three C-N schemes, which was close to that in the C-only version (0.57).

However, SM3 had the lowest CUE (0.35) due to both C cost for plant actively uptake N and the assumption that increase respiration to remove the excess C.

**3.2 Simulation of C storage capacity**

The ecosystem C storage capacity also differed greatly among the three C-N coupling schemes and the C-only version of TECO model (Fig. 7). The C-only version had the largest

C storage capacity (19.5 Kg C m$^{-2}$) among the four simulations due to its highest NPP (879.9

g C m$^{-2}$ yr$^{-1}$). The C storage capacity in SM1 (15.1 Kg C m$^{-2}$) was close to that in SM2 (13.7

Kg C m$^{-2}$). The SM3 had the lowest C storage capacity (8.9 Kg C m$^{-2}$) among the four simulations as a result of its smallest NPP (483.9 g C m$^{-2}$ yr$^{-1}$) and relative short MRT (18.6

years). By comparison with the C-only version, the three C-N schemes all induced different reductions on NPP (-29%, -15% and -45% for SM1, SM2, SM3, respectively) and further reduced their ecosystem C storage capacity. For the MRT, SM1 exhibited positive effects (+9%) relative to that in the C-only version, while another two schemes induced negative ones (i.e., -16.9% in SM2 and -16.7% in SM3).

**3.3. Ecosystem C residence time**

Ecosystem C residence time ($\tau_E$) is collectively determined by baseline residence time, N

scalar, and environmental scalars as shown in Eqn. (31). Specifically, differences in $\tau_E$ among three C-N coupling schemes and C-only TECO model are determined by baseline residence time and the effects of N scalar on eight plant C pools (Fig. 8). For example, SM1 had the longest $\tau_E$ because the N scalar had very strong control on passive SOM. The baseline residence time was further determined by the C allocation (Fig. 9). Overall, compared with

C-only version, the additional N processes enhanced the partitioning coefficient of NPP to roots (33%, 82% and 53% for SM1, SM2 and SM3, respectively) but decreased the partitioning coefficient to wood (-25%, -45% and -34%, respectively). Furthermore, the decreased partitioning coefficient to wood (b2) regulated the variations of the baseline residence time of wood, structural litter, slow and passive SOM. However, the increased partitioning coefficient to roots (b3) determined the variations of the baseline residence time of roots and metabolic litter.

**3.4. Sensitivity of N processes to NPP and MRT**

For either NPP or MRT, the N processes had different sensitivities among the three C-N

schemes of TECO model (Fig. 10). For NPP, plant C:N ratio had the highest sensitivities in both SM1 (0.32) and SM2 (0.53). However, plant N uptake in SM3 had the highest sensitivity (0.87) for NPP. For MRT, competition between plants and microbes, down- regulation of photosynthesis and plant C:N had the highest sensitivities in SM1 (0.27), SM2

(0.19) and SM3 (0.56), respectively. As the NPP and MRT jointly determined the ecosystem

C storage capacity, the plant tissue C:N ratio, down-regulation of photosynthesis, and plant N

uptake had the highest sensitivities for the ecosystem C storage capacity in SM1 (0.06), SM2

(0.09) and SM3 (0.26), respectively.

**4. Discussions**

**4.1 Underlying N processes and plant production**

Gross or net primary production (i.e., GPP or NPP) is regulated by the amount of N

availability for plant growth through the N demand, which is set by the relative proportion of biomass growth in the different plant components and their C:N stoichiometry (Zaehle et al.,

2014; Thomas et al., 2015). The limitation of equilibrium N on plant production reflects the effects from multiple processes of the C-N interaction, mainly including down-regulation of photosynthetic capacity by N availability, the ecosystem's balance of N inputs and losses (i.e., net ecosystem N exchange), plant N uptake, soil N mineralization, and the C:N

stoichiometry of vegetation and soils. However, due to a lack of consensus on the nature of the mechanisms, the representation of these processes varies greatly among diverse models (Zaehle et al., 2014).

There are two common alternative assumptions for the down-regulation of photosynthesis that have been implemented in models: (1) the change in photosynthetic capacity is directly associated with the magnitude of plant available N (e.g., SM2), and (2) N limitation is associated with foliage N, which feeds back to limit photosynthetic capacity (e.g., SM1 and SM3). Our results showed that both assumptions had significant limitations with similar effects on GPP (Figs. 3a 4a and 3g4g). The probable reason is that the TECO model calculates photosynthesis by light availability and carboxylation rate based on the Farquhar model (Farquhar et al., 1980). The effects of N stress under the TECO framework, either associated with plant available N or associated with foliage N concentration, are estimated according to limiting factors of photosynthetic biochemistry (the maximum rate of carboxylation, $V_{cmax}$, and the maximum rate of electron transport at saturating irradiance, $VJ_{jmax}$). The two assumptions of down-regulation of photosynthesis may have different time-dependent effects on GPP in nonsteady-state systems (Xu et al., 2012; Walker et al., 2017).

At or near the steady state, net ecosystem N exchange is driven by the processes of N input via deposition and fixation and N loss via leaching and volatilization (Zaehle et al., 2014; Thomas et al., 2015). Previous studies have stated that analyzing the steady-state condition is useful to understand N effects because the balance between external N sources and N losses determine whether an ecosystem is N limited (Rastetter et al., 1997; Menge et al., 2009; Thomas et al., 2015). In this study, divergent NPP responses among the three schemes might partly result from their different representations of BNF (Figs. 3 and 10). Specifically, SM2 and SM3 simulated BNF explicitly, which used modified empirical relationships of BNF with NPP and evapotranspiration (ET), respectively (Cleveland et al., 1999). These phenomenological relationships generally captured biogeographical observations of higher rates of BNF in humid environments with high solar radiation (Wieder et al., 2015a). However, the highest response of NPP in only ET-driven BNF (i.e., SM3) may illustrate that not only energetic but also C costs of 'fixing' atmospheric di-N ($N_2$) into a biologically usable form ($NH_3$) broadly affect NPP (Gutschick 1981, Rastetter et al., 2001). This was because SM3 considered C investments in BNF while SM2 did not. By contrast, for the nonsteady state, the NPP-driven BNF creates a positive feedback between BNF and NPP, possibly causing large impact on C dynamic and terrestrial C storage (Wieder et al., 2015a). On the other hand, SM1 applied a different strategy, which set BNF as an option when the plant N uptake is not enough for growth in terms of C investment, leading to the highest plant

NUE (Fig. 6a) but a lower response of BNF to NPP (Fig. 10a). Another driving factor of the net ecosystem N exchange is the N loss, which depends on the rate of leaching and volatilization. In this study, using the same formulation as proportion to the size of soil mineral N pool among the three schemes, the different annual mean magnitude of N leaching was more correlated to soil mineral N. In the original CLM4.5 and O-CN (Oleson et al.,

2013; Zaehle et al., 2010), the soil mineral N pool is divided into two pools (ammonium and nitrate). The N leaching is only valid on the nitrate pool, while the ammonium pool is assumed to be unaffected by leaching. This hypothesis may reduce the correlation between leaching and total soil mineral N.

The processes of plant N uptake and net N mineralization determine how N moves through the plant-soil system, thereby triggering N limitation on plant growth and C storage capacity (Fig. 10). However, to our knowledge, exploring those processes exactly in models is limited by inadequate representation of above- and below-ground interactions that control the patterns of N allocation and whole-plant stoichiometry (Zaehle et al., 2014; Thomas et al.,

2015). Plant tissue, litter, and SOM are the primary sinks of N in terrestrial ecosystems, while

N in these forms is not directly available for plant uptake, leading to an increase in N demand for plant growth. These N must turn over to become available for plant uptake. Therefore, the time for N to stay in these unavailable pools controls the transactional delay between the incorporation of N into plant unavailable pool and becomes available for plant uptake. In this way, the residence time of N in SOM appears to be an important factor for governing plant growth. This N limitation mainly occurs in nonsteady state, because accumulation of N in slow turnover rate SOM pools reduces N available for plant uptake (Thomas et al., 2015). At or near steady state, however, the sequestration of N in SOM mainly affects the C residence time (Fig. 8 and 10b). In this study, the different NUE among three C-N schemes induced by different mechanisms. SM1 had the highest NUE due to the combined effects of plant N

uptake based on C investment strategy (as described above) and flexible tissue C:N ratio.

Nitrogen stress increased tissue C:N ratio (Fig. 5b), leading to a high microbial N

immobilization and then a lower net N mineralization (Fig. 3), which allowed plant cell construction with a lower N requirement. However, this was not the case for the SM3 since both hypotheses of increasing respiration to remove the excess C under N stress and the higher C investment for the BNF lead to the decrease in C input and then limits the microbial immobilization for the passive SOM pool. The inclusion of flexible C:N stoichiometry appeared to be an important feature allowing models to capture responses of the ecosystem C

storage capacity to climate variability through adjusting the C:N ratio of nonphotosynthetic tissues or the whole-plant allocation among tissues (Figs. 9 and 10) with different C:N ratios (Zaehle and& Friend, 2010).

**4.2 Ecosystem N status and C residence time**

Ecosystem N status in models, including plant-available and unavailable N forms, is set by N

inputs from N fixation and N deposition, N losses from leaching and denitrification, and N

gain from the turnover of litter and SOM through tissue senescence and decomposition. As noted above, external N cycle (i.e., N inputs and N losses) couples the N processes within the plant-litter-SOM system, being mainly associated with the limitation of plant production (Vitousek et al., 2004; Vicca et al., 2012; Craine et al., 2015). The effects of ecosystem N

status on C mean residence time (MRT), however, has been much less studied than N

limitation on productivity of plants and soil organisms, because these effects involve various impacts on C transfer among pools and C release from each pool via decomposition and respiration (Thompson & Randerson, 1999; Xia et al., 2013). Therefore, the different impacts of ecosystem N status induce oscillating N limitation on MRT (Figs. 8 and 10) due to the inherently different assumptions of C-N interactions among three C-N coupling schemes (Zhou et al., 2012; Shi et al., 2018).

At the steady state, the different effects of N status on changes in modelled MRT can be attributed to: the different rate of soil N mineralization dependent on the total amount of N in

SOM and its turnover time, immobilization based on the competition strategy between plants and microbes and their stoichiometry, and different deployment of reabsorbed N. The traceability framework in this study can trace those different effects into three components (i.e., climate forcing, N scalar $\xi_N$, and baseline MRT) based on three alternative C-N

coupling schemes under the TECO model framework. Since the forcing data are identical, we assumed the same effects for this component in all four experiments.

In our study, the N scalar $(\xi_N)$ was based on the dynamics of C:N ratios (Eqn. 34).

Therefore, N scalar had no effect on MRT in SM2, resulting from the assumption of fixed

C:N ratio in all C pools (Figs. 5b and 8c). In both SM1 and SM3, however, the N scalar had large effects on the SOM pool, which is probably related to different mechanisms.

Specifically, N scalar in the SM1 had the contrasting effects on MRT of fast and passive

SOM pools (i.e., negative vs. positive, respectively), which may largely be attributed to the plant and microbe competition strategy combining with a much larger passive SOM pool in

TECO-CN model (Du et al., 2017; Zhu et al., 2017). Under N stress, the competition between plants and microbes is expected to be intensified, resulting in increasing C:N ratio of nonphotosynthetic tissues (e.g., wood and root) and the vegetation C:N ratio. This effectively prevents N limitation of cell construction and corresponds to an increase in whole- plant NUE (Thomas et al., 2015). In this case, higher C:N ratio in those tissues lowers structural litter quality, leading to soil microbes to immobilize more N to maintain their stoichiometric balance (Hu et al., 2001; Manzoni et al., 2010). However, in the SM3, increased respiration acted as a mechanism to remove the excess C, which is a stoichiometry- based implementation to prevent the accumulation of labile C to prevent the accumulation of

C beyond the storage capacity under N stress (Zaehle and Friend, 2010; Thomas et al.,

2015). This mechanism promotes respiration of the faster turnover pools (fast and slow SOM

pools, Fig. 5a), leading to increased C:N ratio and decreased MRT in these two pools (Fig. 8).

In the traceability framework, the baseline MRT is determined by the potential decomposition rates of C pools (*C* matrix), coefficients of C partitioning of NPP (*B* vector), and transfer coefficients between C pools (*A* matrix, Eqn. [30]. Xia et al., 2013). The matrices

*A* and *C* are preset in the TECO model according to vegetation characteristics and soil texture (Weng and Luo., 2008). Therefore, the notable spread in baseline MRT across the C-N

schemes was induced by the *B* vector, which was modified by different N-limitation assumptions (Eqns. 1-6). Conceptually, in order to meet the N demand, plants adjust NPP

allocation to N absorption tissues (e.g., roots). In this study, three schemes all had similar trends of adjusting allocation C from wood to roots (Fig. 9), but with different mechanisms.

For both SM1 and SM3, increased root C allocation was mainly driven by N uptake capacity, which is associated with plant competitiveness in SM1 (Fig. 10b) and the respiration of excess labile C in SM3 (Fig. 4f, 4l and 10b), respectively. However, for SM2, increasing root

C allocation may occur in spin-up stage from plant adjustment to whole-plant allocation among tissues to fit fixed C:N ratio.

**5. Conclusions**

The C-N coupling has been represented in ecosystem and land surface models with different schemes, generating great uncertainties in model predictions. The most difference among terrestrial C-N coupling models occurs with the degree of flexibility of C:N ratio in vegetation and soils, plant N uptake strategies, down-regulation of photosynthesis, and the representations of the pathways of N import. In this study, we evaluated alternative representations of C-N interactions and their impacts on C cycle using the TECO model framework. Our traceability analysis showed that the different representations of C-N

coupling processes lead to divergent  simulations of both plant production and C

residence time, and thus the ecosystem C storage capacity. The plant production are mainly affected by the different assumptions on net ecosystem N exchange, plant N uptake, net N

mineralization, and the C:N ratio of vegetation and soil. In comparison, the alternative representations of the plant and microbe competition strategy and plant N uptake, combining with the flexible C:N ratio in vegetation and soils, led to a notable spread effects on C

residence time. Overall, the down-regulation of photosynthesis, plant tissue C:N ratio, plant

N uptake and N re-translocation  are the dominant processes of the ecosystem C

storage capacity. Identifying the representations of main C-N processes under different schemes can help us improve the N-limitation assumptions employed in terrestrial ecosystem models and forecast future C sink in response to climate change.

*Code availability*. The code for TECO-CN and the three C-N coupling schemes is available at https://github.com/zgdu/TECO-CN-2.0-new.

*Data availability*. The data for this paper are available upon request to the corresponding authors.

*Competing interests*. The authors declare that they have no conflict of interest.

**Acknowledgements**

This work was financially supported by the National Key R&D Program of China
(2017YFA0604600), the National Natural Science Foundation of China (31770559,
31722009, 41630528), National 1000 Young Talents Program of China, and the Fundamental
Research Funds for Central Universities. Zhenggang Du also thanks the China Scholarship
Council (201606140130) for scholarship support.

 **Figure legends**

**Figure 1.** Schematic diagram of the terrestrial ecosystem carbon (C) and nitrogen (N)
coupling model (TECO-CN). (A) Canopy module, (B) Plant growth module, (C) Soil water
dynamics module, (D) Soil carbon-nitrogen coupling module. Rectangles represent the
carbon and nitrogen pools. $R_a$, autotrophic respiration. $R_h$, heterotrophic respiration. Retr., re-
translocation. NSC, nonstructural carbohydrate. MNP, mineral N in plant tissues. SOM, soil
organic matter. * set N fixation as an option when the plant N uptake is not enough for
growth in terms of C investment.

**Figure 2.** Schematic diagram illustrating the major carbon (C) and nitrogen (N) flows and
stores in a terrestrial ecosystem, enclosing with alternative assumptions of N processes
represent in SM1, SM2 and SM3, respectively. Light-blue arrows indicate C-cycle processes
and red arrows show N-cycle processes.. Met./Str. Litter, metabolic and/or structural litters;
SOM, soil organic matter. * set N fixation as an option when the plant N uptake is not enough
for growth in terms of C investment in SM1, but go directly to soil mineral N pool in SM2
and SM3.

**Figure 3.** Simulated nitrogen fluxes and soil mineral nitrogen from three carbon-nitrogen
coupling schemes (SM1, SM2 and SM3) in TECO-CN model for 1996 to 2007 at Duke
Forest. Mineral., mineralization; BNF, biological N fixation; Imm., immobilization.

**Figure 4.** Simulated annual (a-f) and mean (g-l) carbon fluxes from carbon-only version and
carbon-nitrogen coupled with three schemes (SM1, SM2 and SM3) of TECO model for 1996
to 2007 at Duke Forest. GPP, gross primary productivity; NPP, net primary productivity;
NEE, net ecosystem exchange of $CO_2$; R-eco, ecosystem respiration; R-heter, heterotrophic
respiration; R-auto, autotrophic respiration.

**Figure 5.** The annual average sizes of carbon pools (panel a) at the steady-state among 1996-
2007 for C-only version and the three C-N schemes (SM1, SM2 and SM3) and the C:N ratio
(panel b) of each carbon pools for the three C-N schemes (SM1, SM2 and SM3) in TECO-
CN model.

**Figure 6.** The nitrogen use efficiency (NUE, panel a) in three C-N schemes of TECO model
(SM1, SM2 and SM3) and the carbon use efficiency (CUE, panel b) at the steady-state
among C-only version and the three C-N schemes of TECO model (SM1, SM2 and SM3).

**Figure 7.** Simulation of annual ecosystem carbon storage capacity for 1996 to 2006 at Duke
Forest by carbon in flux (NPP, x axis) and ecosystem residence time ($\tau_E$, y axis) in TECO
model framework with three carbon-nitrogen coupling schemes (SM1, SM2 and SM3) and in
TECO C-only model (C). Inset (a), ecosystem carbon residence time ($\tau_E$) in SM1, SM2, SM3
and C-only model; inset (b), mean ecosystem carbon storage simulated among SM1, SM2,
SM3 and C-only model; inset (c), relative change of NPP and ecosystem residence time
simulated among three schemes compared with in C-only model.

**Figure 8.** Determination of carbon-pool residence times based on traceability framework in
TECO C-N model with three C-N coupling schemes (SM1, SM2 and SM3) and TECO C-
only model (C). Panel (a), baseline residence time; panel (b), mean residence time, and panel
(c), nitrogen scalar.

**Figure 9.** Coefficients of partitioning of NPP to nonstructural C (NSC), root, woody and leaf
in C-only model (C) and C-N coupling model with three schemes (SM1, SM2 and SM3).

**Figure 10.** The sensitivity of nitrogen processes to NPP (panel a), ecosystem residence time
($\tau_E$, panel b), and ecosystem C storage capacity (panel c) among three carbon-nitrogen
coupling schemes (SM1, SM2 and SM3). DRP, down-regulation of photosynthesis; PS, plant
tissue C:N ratio; PNU, plant N uptake; PMC: plant and microbe competition; BNF, biological
N fixation; RtrN, re-tranlocation N; SS, soil pool C:N ratio.

**Table1.** Summary of the nitrogen-carbon coupling schemes used and the representation of
key processes in the carbon-nitrogen cycle.

| | SM1 (TECO-CN)[a] | SM2 (CLM4.5)[b,c] | SM3 (O-CN)[d,e] |
|---|---|---|---|
| **Down-regulation of photosynthesis by N availability (DRP)** | Based on the comparison between plant N demand and actual supply | Based on the available soil mineral N relative to the N demanded to allocate photosynthate to tissue | Based on foliage N concentration, which varies with N deficiency |
| **Plant tissue stoichiometry (PS)** | Flexible plant C:N ratio | Fixed plant C:N ratio | Flexible plant C:N ratio |
| **Plant N uptake (PNU)** | Based on fine root biomass, soil mineral N and N demand of plant.

Plants itself choose the strategy between uptake from soil mineral N and fix $N_2$ by comparing C investment | Based on N required to allocate NPP to tissue.

Plants uptake N for free | Combining active and passive uptake of mineral N based on fine root C, soil mineral N, plant transpiration flux, increases with increased plant N demand |
| **N competition between plants and microbes (PMC)** | Microbes have first access to soil mineral N | Based on demand by both microbial immobilization and plant N uptake | Microbes have first access to soil mineral N, the competitive strength of plants increases under nutrient stress |
| **Biological N fixation (BNF)** | Based on the nitrogen demand of plants and maximum N fixing ratio considering nutrient concentration | $f(NPP)$ | $f(ET)$ |
| **Deployment of re-translocated N (RtrN)** | Fixed fraction of litter | Based on available N in the tissue and the previous year's annual sum of plant N demand | Fixed fraction of dying leaf and root tissue |
| **Soil organic matter stoichiometry (SS)** | Flexible soil C:N ratio | Fixed soil C:N ratio | Flexible soil C:N ratio |
| **N leaching** | Function of soil mineral N pool and runoff | Function of soil mineral N pool and runoff | Function of soil mineral N and runoff |
| **\*Gaseous N loss** | Based on function of soil mineral N pool and soil temperature, and N deficit | Based on function of soil mineral N pool and soil temperature, and N deficit | Based on function of soil mineral N pool and soil temperature, and N deficit |

[a]See this study; [b]Koven et al. (2013), [c]Oleson et al. (2013); [d] Zaehle and&Friend (2010),
[e]Zaehle et al. (2011). *, use the same representation as in TECO-CN model among three
schemes.

C, carbon; N, nitrogen; NPP, net primary productivity; ET, evapotranspiration.

**Figure 1. TECO-CN**

[Figure]

**Figure 1.** Schematic diagram of the terrestrial ecosystem carbon (C) and nitrogen (N)
coupling model (TECO-CN). (A) Canopy module, (B) Plant growth module, (C) Soil water
dynamics module, (D) Soil carbon-nitrogen coupling module. Rectangles represent the
carbon and nitrogen pools. $R_a$, autotrophic respiration. $R_h$, heterotrophic respiration. Retr., re-
translocation. NSC, nonstructural carbohydrate. MNP, mineral N in plant tissues. SOM, soil
organic matter. * set N fixation as an option when the plant N uptake is not enough for
growth in terms of C investment.

**Figure 2**

[Figure]

**Figure 2.** Schematic diagram illustrating the major carbon (C) and nitrogen (N) flows and
stores in a terrestrial ecosystem, enclosing with alternative assumptions of N processes
represent in SM1, SM2 and SM3, respectively. Light-blue arrows indicate C-cycle processes
and red arrows show N-cycle processes. Met./Str. Litter, metabolic and/or structural litters;
SOM, soil organic matter. * set N fixation as an option when the plant N uptake is not enough
for growth in terms of C investment in SM1, but go directly to soil mineral N pool in SM2
and SM3.

**Figure 3**

[Figure]

**Figure 3.** Simulated nitrogen fluxes and soil mineral nitrogen from three carbon-nitrogen coupling schemes (SM1, SM2 and SM3) in TECO-CN model for 1996 to 2007 at Duke Forest. Mineral., mineralization; BNF, biological N fixation; Imm., immobilization.

**Figure 4**

[Figure]

**Figure 4.** Simulated annual (a-f) and mean (g-l) carbon fluxes from carbon-only version and
carbon-nitrogen coupled with three schemes (SM1, SM2 and SM3) of TECO model for 1996
to 2007 at Duke Forest. GPP, gross primary productivity; NPP, net primary productivity;

NEE, net ecosystem exchange of $CO_2$; R-eco, ecosystem respiration; R-heter, heterotrophic
respiration; R-auto, autotrophic respiration.

**Figure 5**

[Figure]

[Figure]

**Figure 5.** The annual average sizes of carbon pools (panel a) at the steady state among 1996-
2007 for C-only version and the three C-N schemes (SM1, SM2 and SM3) and the C:N ratio
(panel b) of each carbon pools for the three C-N schemes (SM1, SM2 and SM3) in TECO-
CN model.

**Figure 6**

[Figure]

**Figure 6.** The nitrogen use efficiency (NUE, the ratio of NPP:PNU, panel a) in three C-N
schemes of TECO model (SM1, SM2 and SM3) and the carbon use efficiency (CUE, the
ratio of NPP:GPP, panel b) at the steady-state among C-only version and the three C-N
schemes of TECO model (SM1, SM2 and SM3).

**Figure 7**

[Figure]

**Figure 7.** Simulation of annual ecosystem carbon storage capacity for 1996 to 2006 at Duke
Forest by carbon in flux (NPP, x axis) and ecosystem residence time ($\tau_E$, y axis) in TECO
model framework with three carbon-nitrogen coupling schemes (SM1, SM2 and SM3) and in
TECO C-only model (C). The hyperbolic curves represent constant values (shown across the
curves) of ecosystem carbon storage capacity. Inset (a), ecosystem carbon residence time ($\tau_E$)
in SM1, SM2, SM3 and C-only model; inset (b), mean ecosystem carbon storage simulated
among SM1, SM2, SM3 and C-only model; inset (c), relative change of NPP and ecosystem
residence time simulated among three schemes compared with in C-only model.

**Figure 8**

[Figure]

**Figure 8.** Determination of carbon-pool residence times based on traceability analysis
framework in TECO C-N model with three C-N coupling schemes (SM1, SM2 and SM3) and
TECO C-only model (C). Panel (a), baseline residence time; panel (b), mean residence time,
and panel (c), nitrogen scalar.

**Figure 9**

[Figure]

**Figure 9.** Coefficients of partitioning of NPP to nonstructural C (NSC), root, woody and leaf
in C-only model (C) and C-N coupling model with three schemes (SM1, SM2 and SM3).

**Figure 10**

[Figure]

**Figure 10.** The sensitivity of nitrogen processes to NPP (panel a), ecosystem residence time ($\tau_E$, panel b), and ecosystem C storage capacity (panel c) among three carbon-nitrogen coupling schemes (SM1, SM2 and SM3). DRP, down-regulation of photosynthesis; PS, plant tissue C:N ratio; PNU, plant N uptake; PMC: plant and microbe competition; BNF, biological N fixation; RtrN, re-tranlocation N; SS, soil pool C:N ratio.